# VeSX: A Framework Featured by Verification, Self-Correction and In-context Learning for Web Automation Tasks

## Abstract

While large language models have achieved remarkable success in tasks such as reasoning and question answering, applying LLMs to interactive tasks like web automation remains challenging. In web automation, existing planning-execution workflow often faces limitations due to the infeasible subtasks. We propose VeSX, a framework designed to enhance subtask feasibility through verification, self-correction, and in-context learning. VeSX introduces three key improvements: (1) subgoal-guided verification, which verifies the execution results of subtasks based on the preset subgoals; (2) hierarchical self-correction, which combines reflection and replanning, targeting to self-correct mistakes in both planning and execution phases; (3) exemplar bank, which improves in-context learning by partitioning execution trajectories and heuristically generating metadata for exemplars. We evaluate VeSX on WebArena benchmark and achieve the state-of-the-art average success rate of 0.34, which significantly outperforms existing methods without human guidance on all five scenarios.

## 1 Introduction

Large language models (LLMs) have achieved notable success in tasks such as logical reasoning, commonsense understanding, and question answering (Achiam et al., 2023; Touvron et al., 2023; Brown et al., 2020; Wei et al., 2022; Yao et al., 2023). Building agents with LLM as backbones to handle tasks in interactive scenarios has been identified as a feasible and promising approach but applying these agents in the real world remains highly challenging (Wang et al., 2023b; Kim et al., 2024; Wang et al., 2024). Web automation is one such scenario where LLM agents autonomously control web browsers to perform online activities such as shopping and posting by executing actions like clicking or typing.

When handling web automation tasks, two predominant agent workflow paradigms are empirically employed by current LLM agents. The first involves sequentially generating a series of actions (Koh et al., 2024; Deng et al., 2024; Zheng et al., 2023), as illustrated in Figure 1(a). The second commences with a planning phase, breaking down the task into a series of subtasks which are to be tackled one by one(Kim et al., 2024; Sodhi et al., 2024). For complex tasks that require multi-step reasoning, the second workflow is beneficial for that it breaks actions down into easier subtasks like chain-of-thought instead of a hard final task. However, such workflows are often constrained by the lack of guarantees regarding subtask feasibility (Wang et al., 2023b). An effective and commonly-used remedy is to rely on expertly designed subtask candidates with manually crafted prompts to decompose the entire task as illustrated in Figure 1(b). However, this method requires human annotation for new tasks, which limits the ability of LLM agents to autonomously adapt to more advanced and diverse web environments Sodhi et al. (2024); Zhou et al. (2022). Additionally, there is a significant shortage of data available for learning within these workflows, which collectively restricts the capabilities of LLMs during the planning and execution phases Lai et al. (2024). To overcome these limitations, we propose VeSX, a framework for LLM agents in web automation that seeks to improve subtask feasibility and empower LLM both in planning and execution through **Ve**rification, **S**elf-correction, and in-conte**X**t learning.

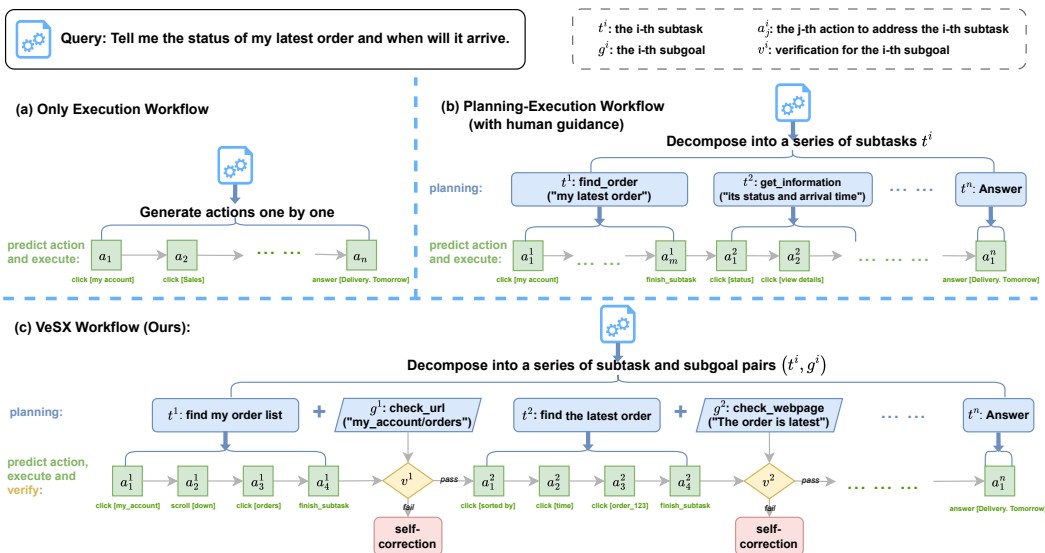

Figure 1: Comparison among different agent workflows for web automation. (a) is the workflow only engages the execution phase that predicts actions one by one. (b) is the planning-execution workflow with human guidance. (c) is the proposed VeSX workflow, featured with verification and self-correction.

VeSX introduces two significant modules to the planning-execution workflow for web automation tasks, as illustrated in Figure 1(c). The first involves adopting **subgoal-guided verification** to ascertain the successful completion of subtasks. VeSX generates subgoals in the planning phase and conducts verification to check whether the execution result satisfies the subgoal once execution is finished. We devised two modes for verification: self-verification which makes LLMs do verification by themselves, and external verification which calls for external functions to conduct verification. Verification ensures the completion of the subtasks and avoids the issue of subsequent subtasks becoming infeasible due to errors in the execution of previous subtasks.

The second improvement is **hierarchical self-correction**, which aims to self-correct both execution and planning. When verification fails, VeSX first tries to identify and correct errors in the current subtask's execution trajectory, a process referred to as "reflection". If reflection is unsuccessful, the subtask is likely infeasible, prompting VeSX to regenerate the remaining incomplete subtasks in a process called "replanning". This hierarchical self-correction mechanism facilitates error correction during both the execution and planning phases, enabling the framework to effectively assess the feasibility of subtasks and generate new plans as replacements.

Furthermore, VeSX develops an **exemplar bank** for in-context learning within the planning and execution phase. Existing research on in-context learning related to web automation primarily focuses on the execution-only workflow and is limited to simple simulated tasks, leading to a dependence on complete execution trajectories as exemplars (Kim et al., 2024; Zheng et al., 2023). However, complex web automation tasks in the real world involve long action sequences, and using entire trajectories as exemplars includes a lot of irrelevant information, which will consume excessive content space. To address these issues, VeSX samples actions and planning strategy from the entire trajectory as exemplars for execution and planning, and heuristically generates metadata for retrieval. This method creates a larger and more diverse dataset for the LLM to learn from, and therefore improve its performance in both the planning and execution phases.

We evaluate VeSX on 5 scenarios under the web automation benchmark, WebArena(Zhou et al., 2023b). VeSX outperforms all publicly reported methods without human guidance by a huge margin and pushes the state-of-the-art success rate to 34.0, which even exceeds the performance of the method using human guidance.

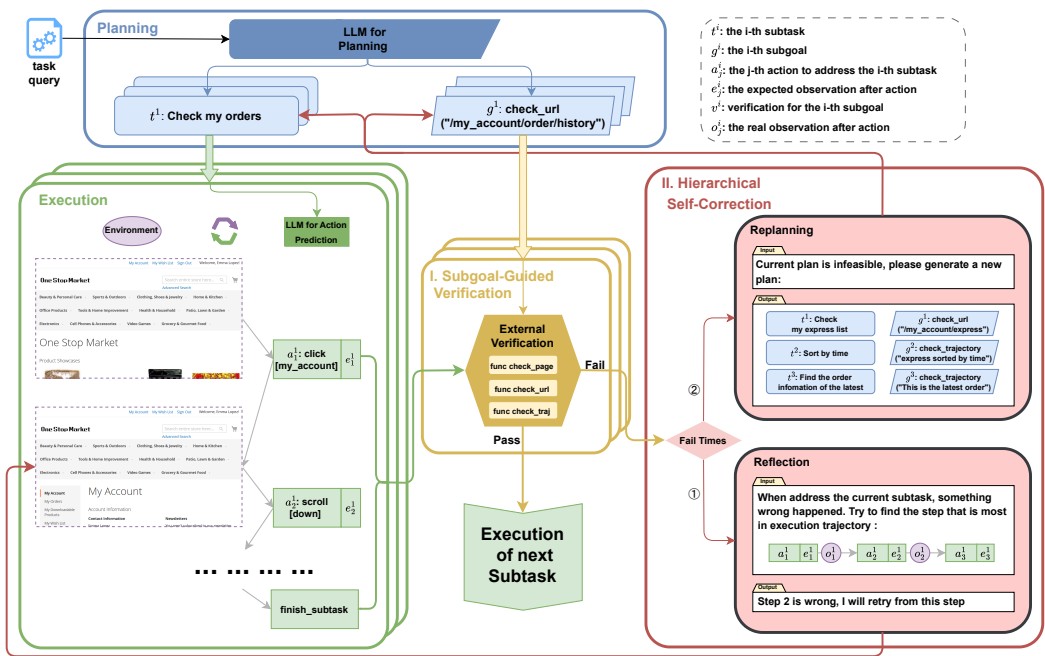

Figure 2: The framework of VeSX, which is made up of four phases and featured with subgoal-guided verification and hierarchical self-correction.

## 2 VeSX

### 2.1 Overview

The VeSX involves innovative design in both process and data aspects. In terms of process, VeSX implements a highly accurate verification scheme based on subgoals generated in the planning phase, which we refer to as subgoal-guided verification (Section 2.2). Additionally, when verification fails, we design a hierarchical self-correction scheme that involves local reflection to tackle the problems within one certain subtask and global replanning to address the errors happening in the planning (Section 2.3). In terms of data, to address the low efficiency of in-context-learning in the previous web automation methods, VeSX uses a single step instead of the entire trajectory for in-context-learning, and improves the metadata construction method to efficiently and accurately retrieve effective exemplars (Section 2.4).

### 2.2 Subgoal-Guided Verification

**Generate subgoals alongside with subtasks** In order to verify whether execution results align with the intentions, VeSX instructs the LLM to generate not only subtasks but also corresponding subgoal pairs during the planning phase, as illustrated in Figure 2. The requirements for tasks in real-world scenarios are often highly diverse, leading to a wide array of corresponding solutions. However, the methods for assessing whether these requirements can be met are typically simplistic and fixed. For example, in web automation, although there may be hundreds of distinct task intents to handle, the evaluation methods are relatively limited. For navigation tasks, the evaluation may be based on whether a specific webpage has been reached. For information-gathering tasks, the presence of the required information on the final webpage is often the criterion. We observe that evaluating the execution result of a subtask is much simpler and more standardized compared to addressing the subtask itself. From this perspective, we propose the subgoal-guided verification, which evaluates the execution results of the subtasks according to the subgoals. Specifically, in VeSX, we generate pairs of subtasks and subgoals during the planning phase. Subtasks are articulated in natural language to capture the diverse task requirements, which in turn guide the generation of actions during the execution phase. Meanwhile, subgoals, which are formulated in either natural

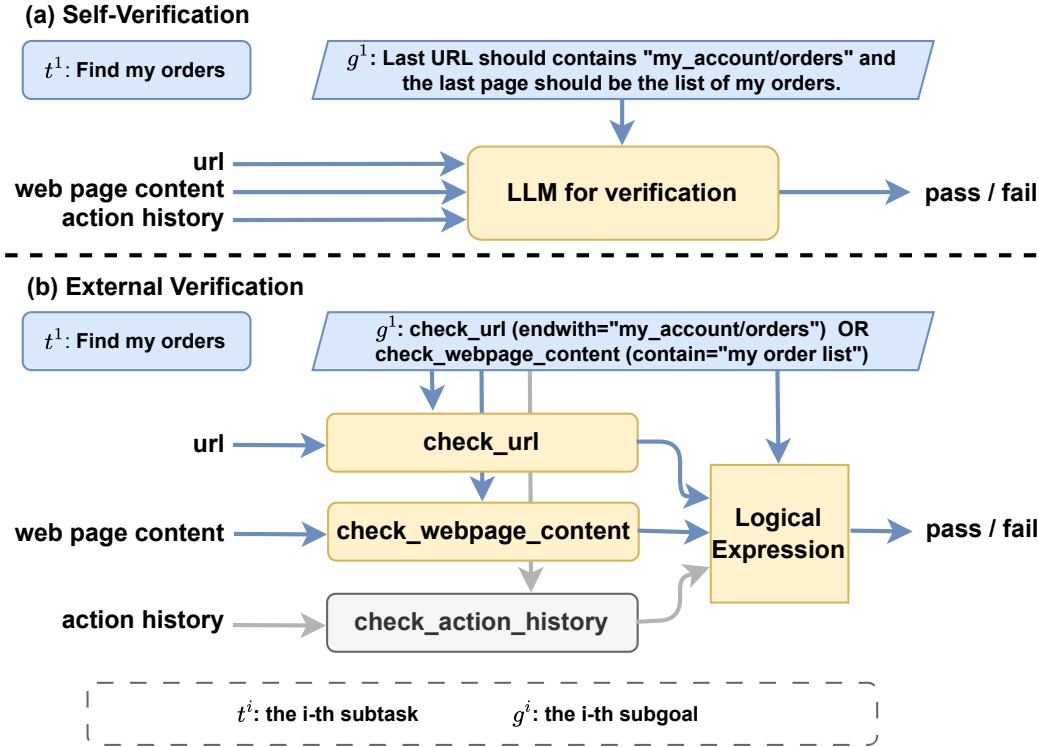

Figure 3: Two methods for verification. (a) is the self-verification using the LLM itself to check the subgoals are accomplished or not. (b) is the external verification using generally predefined verification functions to check the accomplishment, which is more stable and reliable.

or structured language—corresponding to the two verification methods we discuss later—are used to invoke external models or pre-designed tools to verify whether the subtasks have been successfully completed.

**Verification Method** We employed two methods to implement verification as illustrated in Figure 3. The first method is self-verification, which involves using natural language for subgoals. Self-verification incorporates all external environmental information and execution processes, making the LLM itself verify whether the execution results align with the previous intent. Self-verification has the advantage of flexibility, which means it can give the LLM to make decisions according to the current environment rather than totally depending on the predefined scheme.

The second method is external verification, which involves presenting subgoals in a structured language akin to function calls. In this approach, subgoals parameterize a set of predefined verification functions, creating a logical expression that defines the pass/fail criteria. During the verification process, external functions will be invoked according to the specifications of the subgoals. The results of these invocations will undergo logical operations to produce the final verification outcome, as illustrated in Figure 3(b). The advantage of this method is that it relies on external functions for verification, which is more stable, controllable, allows for swift adjustments, and reduces dependence on the LLM's capabilities.

## 2.3 HIERARCHICAL SELF-CORRECTION

Due to the inherent unpredictability in web environments, errors during an agent's execution are often unavoidable, making self-correction essential. However, since planning can lead to infeasible subtasks, merely reflecting on and retrying execution actions is insufficient for effective error correction. From this perspective, we propose a hierarchical self-correction approach. Specifically, the hierarchical self-correction phase will be engaged when the verification fails. First, VeSX will em-

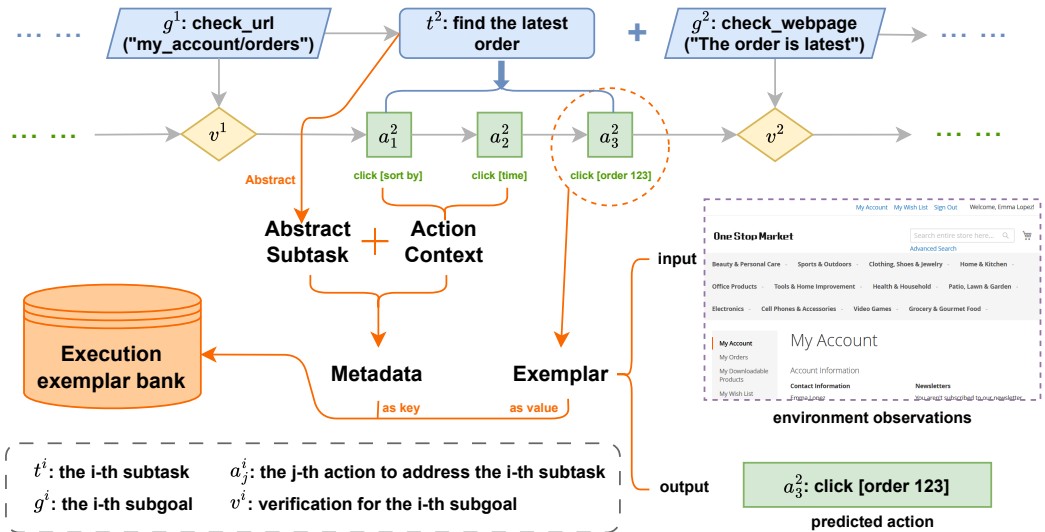

Figure 4: Construction of execution exemplar bank. We use the abstract subtask and the action context as the metadata for the exemplar. The key-value pairs of metadata's embedding are stored to construct exemplar bank.

ploy reflection to identify mistakes in the execution trajectories associated with the current subtask. If reflection fails, it indicates that the generated subtasks may be infeasible or too challenging to solve. In that case, replanning will adjust the subtasks and subgoals dynamically.

**Reflection** In reflection, the LLM takes the execution trajectory and verification results as inputs to identify the step where the error occurred and subsequently re-execute from that point. Previous work on making LLMs reflect primarily focused on errors that occur when actions are generated by the environment and task queries. However, in real interactive scenarios, errors are likely to occur when changes in the environment after action execution do not align with the expected outcomes of those actions. Adjustments to previous actions are also needed in such cases. To address this issue, we enhanced the outputs during the execution phase by having the LLM additionally output the expected results $e_j^i$ following an action. This allows the model to reflect on whether there is a discrepancy between expectations and environment changes during reflection thereby adjusting the action generation accordingly.

**Replanning** To enable dynamic adjustment capabilities in planning, VeSX employs replanning when reflection fails to correct errors. Specifically, when execution can not pass verification and reflection cannot rectify the issue, we consider the current subtask to potentially be infeasible. The previous error information and environment feedback are then fed into the LLM, prompting the LLM to regenerate subsequent subtasks and subgoals. Compared to the common framework that relies solely on reflection, the hybrid self-correction approach allows for adaptive adjustment, avoiding infeasible subtasks and thereby enhancing the long-term planning capabilities within the framework.

## 2.4 EXEMPLAR BANK

Previous research has typically relied on recalling the entire correct execution trajectories to perform in-context learning for execution. However, real-world tasks often require multi-step reasoning and operations, with the underlying logic of these steps varying significantly. This makes it challenging to use the full trajectory as an exemplar to guide the prediction of single-step actions, as it incorporates a significant amount of extraneous information and consumes substantial context. We decompose trajectories into individual actions as exemplars and have meticulously designed a heuristic approach to generate their metadata, which will serve as the key for recalling these exemplars later there is a lack of research on collecting and recalling planning exemplars.

**Metadata of execution exemplars**  VeSX utilizes single-step action instead of the entire trajectory as an exemplar to avoid the redundancy of the whole trajectory and enables LLM to learn from more diverse action examples. This involves designing metadata to describe the exemplars effectively. Specifically, for each action, VeSX will abstract the tackling subtask and concatenate with the action context, which means the sequence of prior actions within that subtask to get the metadata for the exemplar, as shown in Figure 4. For instance, in the subtask "Find the latest order" the correct execution trajectory constitutes three actions `click [sort by]`, `click [time]`, and `click [order 123]`. We treat each action as an individual exemplar and generate its metadata. The metadata for the third action would be "Abstract Subtask: find conditional order; Action Context: click sort by, click time". For each piece of exemplar, we store its embedding of metadata as the key and the exemplar as the value to construct the execution exemplar bank. Subsequently, during execution, VeSX recalls the most relevant examples from the exemplar bank through similarity-matching embeddings for in-context learning.

**Collecting planning exemplars**  Due to the planning phase playing a significant role in the planning-execution framework, VeSX also collects the planning exemplars from the correct trajectory except for the execution exemplars. The metadata of planning exemplars is generated from its corresponding task query, abstracting the specific concepts in it, and then the planning exemplar bank is constructed with the key-value pairs of embedding of metadata and exemplars. Construct planning exemplar bank enhance models' reasoning ability in the planning phase and facilitate the generation of more feasible subtasks.

## 3  EXPERIMENTS

### 3.1  EXPERIMENTAL SETUP

We evaluate VeSX on 5 different scenarios within the benchmark WebArena. WebArena is a benchmark featured with complex web tasks across multiple scenarios including Shop (Amazon Shopping site), CMS (Amazon Commercial Management System), Red (Reddit), Git (Gitlab), and Map (Open Street Map). It builds several virtual websites from the snapshots of real-world websites like Amazon Shopping, GitLab, and Reddit, and allows agents to take actions freely like humans do in browsers, making it well-suited for evaluating what LLM web agents will do in the real world.

To ensure fair comparisons, we choose `gpt-4o` the same as previous works without finetuning the LLMs. We use `text-embedding-3-small` as the embedding model and retrieve the top 3 relevant exemplars for in-context learning. We follow the browser system and action space built by Zhou et al. (2023b). We also set the temperature of the LLM to zero to minimize the uncertainty to generate reproducible results. For the data collection for the exemplar, we sampled 60 tasks randomly for each scenario and let the LLM try to solve them but not evaluate them with the correct answer. We heuristically filter out the bad trajectories like repeating actions or reaching the max actions, then make LLM self-evaluate the result by itself. If LLM considers the whole execution trajectory to be reasonable and the final result may be correct, the actions in this trajectory will be collected as exemplars. We use external verification in default unless a specific verification method is explicitly specified.

### 3.2  MAIN RESULTS

**Baselines**  We selected five works from different perspectives as the baseline for the experiment. Zhou et al. (2023b) and Lai et al. (2024) focus on improving the single model's reasoning ability by prompt or reinforcement learning: Zhou et al. (2023b) tries to improve the model performance using various combinations of prompt strategies. Lai et al. (2024) uses reinforcement learning to fine-tune an LLM with more knowledge of the web environment. Drouin et al. (2024) tries to refine this problem from the perspective of the environment, which builds a new browser system with more versatile actions and tries to resolve the tasks from the perspective of the environment. Koh et al. (2024) and Sodhi et al. (2024) improve from the perspective of the LLM agnet workflow, using the strategy like reflection and planning: Koh et al. (2024) enhances the workflow of generating actions sequentially by tree search method. And Sodhi et al. (2024) improves the planning-execution workflow with the planning in a stack of subtasks and expert-designed examples.

| Method | Model | Human-Feedback | Shop. | CMS | Red. | Git. | Map | Avg |
|--------|-------|----------------|-------|-----|------|------|-----|-----|
| Zhou et al. (2023b) | gpt-4o | × | - | - | - | - | - | 0.131 |
| Lai et al. (2024) | AutoWebGLM | × | - | - | - | - | - | 0.182 |
| Drouin et al. (2024) | gpt-4o | × | 0.172 | 0.148 | 0.202 | 0.190 | 0.255 | 0.235 |
| Koh et al. (2024) | gpt-4o | × | 0.281 | 0.165 | 0.105 | 0.133 | 0.255 | 0.192 |
| Sodhi et al. (2024) | gpt-4o | ✓ | *0.369* | *0.242* | **0.594** | **0.317** | **0.303** | *0.335* |
| VeSX | gpt-4o | × | **0.412** | **0.249** | *0.545* | *0.283* | *0.291* | **0.340** |

Table 1: Task success rate on five scenarios of WebArena. The best performance is displayed in bold, and the second performance is displayed in italics. It constitutes of five scenarios: Shop (Amazon Shopping site), CMS (Amazon Commercial Management System), Red (Reddit), Git (Gitlab), and Map (Open Street Map)

| Do Verification? | Methods | SR | ΔSR |
|:----------------:|:-------:|:----:|:-----:|
| × | - | 0.288 | 0.0 |
| ✓ | self | 0.346 | +0.058 |
| ✓ | external | 0.412 | +0.134 |

Table 2: Ablation study on subgoal-guided verification. SR represents the success rate.

| Reflection | Replan | SR | ΔSR |
|:----------:|:------:|:----:|:-----:|
| × | ✓ | 0.261 | 0.0 |
| ✓ (w/o expect) | × | 0.314 | +0.052 |
| ✓ | × | 0.340 | +0.078 |
| ✓ | ✓ | 0.412 | +0.150 |

Table 3: Ablation study on hierarchical self-correction. SR represents the success rate.

| Methods | SR | ΔSR | #P | #E |
|:-------:|:----:|:-----:|:--:|:--:|
| only execution exemplars | 0.340 | 0.0 | 0 | 171 |
| only planning exemplars | 0.373 | +0.033 | 35 | 0 |
| VeSX | 0.412 | +0.072 | 35 | 171 |

Table 4: Ablation study on exemplar bank. SR represents the success rate. #P refers to the number of planning exemplars and #E refers to the number of planning exemplars.

| Methods | SR | ΔSR |
|:-------:|:----:|:-----:|
| w/o abstraction | 0.275 | 0.0 |
| w/o action context | 0.333 | +0.058 |
| abstraction & action context | 0.412 | +0.137 |

Table 5: Ablation study on building metadata. SR represents the success rate.

As shown in Table 1, VeSX consistently outperforms the publicly known best-reported methods without human guidance in all five different scenarios. The improvements are particularly Significant in the Shop, CMS, and Red scenarios, with relative enhancements of 47%, 51%, and 169%, respectively, benefiting from our robust error detection and correction capabilities. Additionally, compared to the method in Sodhi et al. (2024), which uses expert-written examples for each scenario, our framework—entirely reliant on autonomous planning and execution by the LLM—demonstrates competitive performance, achieving superior results in two out of five scenarios, which are Shop and CMS. The hard tasks in Shop and CMS often require multi-step reasoning, creating a great opportunity for VeSX to showcase its error correction capabilities.

## 3.3 ABLATION STUDIES

To evaluate the impact of each part of VeSX, we conduct a comprehensive ablation study in this section. Our experiments are mainly conducted in the Shop scenario, whose tasks often involves multi-step reasoning and operations, well-suited for evaluating the effectiveness of our various methods. SR means the success rate of all tasks in this scenario, which corresponds to the column of Shop in Table 1. These findings validate the effectiveness of the subgoal-guided verification, hierarchical self-correction, and exemplar bank.

### 3.3.1 ABLATING SUBGOAL-GUIDED VERIFICATION

We conducted a comparative experiment to assess the implementation and methodologies of verification. verification will not be performed for the implementation of no verification because it does not have the ability to detect the error. As shown in Table 2, with the implementation of both verification schemes, the success rate has significantly improved. It is because the two verification methods can both enable the framework the error-detection ability. Moreover, using external verification provides additional support compared to self-verification, resulting in even more pronounced enhancements.

### 3.3.2 ABLATING HIERARCHICAL SELF-CORRECTION

We conducted ablation experiments to assess the impact of various components of self-correction on overall performance. We tested three implementations: using only reflection, using only replanning, and employing reflection without generating expectations during execution.

As illustrated in Table 3, it is evident that removing any component significantly affects the final performance. In comparison to replanning, the removal of reflection has a more substantial impact, indicating that errors generated during the execution phase are more frequent or potentially easier to rectify. Nonetheless, the absence of replanning also leads to a rapid decline in the success rate, underscoring its necessity. Moreover, the success rate decreases when expectations are not utilized, demonstrating that expectations indeed enhance VeSu's ability to detect discrepancies between environmental changes and action predictions.

### 3.3.3 ABLATING EXEMPLAR BANK

We performed ablation experiments on the components of the exemplar bank and evaluated various strategies for generating metadata to determine each element's contribution to overall system performance. The construction of the exemplar bank follows the same format as the main results. We sampled 60 tasks from the shopping experiment and employed a heuristic approach, allowing the LLM to evaluate its own execution results. Ultimately, we extracted 35 trajectories that were deemed correctly completed, thereby forming the exemplar bank.

As demonstrated in Table 5, excluding either the planning or execution exemplars led to a decrease in the success rate. Notably, utilizing only planning exemplars, without the inclusion of execution exemplars, achieved a higher success rate despite relying on fewer data samples. Based on the observations found in Section 3.3.2, this phenomenon suggests that for each subtask, generating accurate execution sequences requires dynamic adjustments. Even with a limited number of samples, the LLM can accomplish tasks through self-correction, indicating a higher dependency on an effective error-correction mechanism rather than on the volume of data. Conversely, errors in the planning phase seem to pose a more complex challenge, requiring data as a reference for effective reasoning. This also implies that, compared to execution exemplars, planning exemplars are more efficient in improving performance.

We conducted an ablation study on two techniques: abstracting subtasks and using action context. Our results indicate that both techniques are essential for enhancing the effectiveness of in-context learning. Moreover, omitting abstraction results in a more significant loss in accuracy compared to omitting action context. This may be attributed to the system's challenges in retrieving the most relevant examples from the exemplar bank and learning from them without abstraction.

## 4 RELATED WORK

**Web automation tasks.** Web automation is a sequential decision process, where agents are given a task that can be carried out on a series of websites, such as buying products, booking a ticket, and summarizing opinions on forums. The earliest research in this area often formulated the task as a Markov Decision Process and employed reinforcement learning methods Liu et al. (2018); Gur et al. (2018); Jia et al. (2018); Humphreys et al. (2022). However, with the increase in the scale of websites and tasks, these methods suffer for they are hard to comprehend and process the long web documents and in lack of generalization to different tasks (Gur et al., 2023).

**Large language model for web automation.** Language models have been used as text decoders since the era reinforcement learning is the prevalent solution to web automation tasks (Humphreys et al., 2022). As the ability to reasoning and acting of language models soared, using language models to make decisions in a web environment shows better performance, particularly on the real-world scale in recent years. Most of these methods use in-context learning paradigm (Kim et al., 2024; Zheng et al., 2023) or deal with other difficulties like long document (Zheng et al., 2023; Deng et al., 2024). However, applying this paradigm to multi-step reasoning tasks often leads to suboptimal performance. Other approaches decompose these tasks into predefined subtasks (Sodhi et al., 2024) or sequences of actions (Kim et al., 2024), which require manual design and may introduce inaccuracies. There also be works using finetuning to improve language models' ability of reasoning or understanding of web pages (Gur et al., 2023; Lai et al., 2024), which is orthogonal to our method.

**Large language model agent workflow.** LLM agents are artificial entities that can sense their environment, make decisions, and take actions. They leverage the capabilities of large language models (LLMs) to perform tasks autonomously (Xi et al., 2023). The general conceptual framework for LLM-based agents includes three main components: brain (primarily composed of an LLM), perception (multimodal inputs), and action (expanding the action space of LLM-based agents). There are many development in the research of how to construct LLM agent to interact with wild outer world, such as (Li et al., 2023; Wang et al., 2023a;b) or construct LLM agent to follow human's instructions to complete a given task with external tools, such as (Lightman et al., 2023; Hong et al., 2023; Lai et al., 2024; Significant Gravitas; Agashe et al., 2024). Web automation is a task that necessitates LLM the ability to interact with the web environment with the target given by human.

**Planning with large language model.** How to augment language models with reasoning skills and ability to act is of great significance when using language models in real world. Recent researches find that providing a series of intermediate reasoning steps can help LLM perform complex reasoning (Wei et al., 2022). Previous works augment the reasoning steps with the observations of the external tools (Yao et al., 2023; 2024). These prompt strategies try to decomposing the task into a sequence of actions. However, when the tasks become complex and need more steps to solve, it will be hard for LLM to generate correct action each step and the reasoning steps are hard to learn through in-context learning. Some other methods try to decompose the tasks into predefined subtasks, which can be guided to finish through in-context learning (Khot et al., 2022; Zhou et al., 2022; Sodhi et al., 2024). These methods need to manually define and write prompts for subtasks, and when the tasks become more and more various, it will be time-consuming and costly. Researches in other areas demonstrated that decomposing the task into the subtasks and using the verification module to ensure the effectiveness of the subtasks perform well in complex reasoning tasks (Wang et al., 2023b; Gou et al., 2023; Zhou et al., 2023a). Due to the strong interactivity and complexity of web automation tasks, simply transplanting these solutions is not applicable.

## 5  LIMITATIONS AND DISCUSSION

VeSX is an innovative LLM workflow designed for handling web automation tasks, characterized by subgoal-guided verification, hierarchical self-correction, and an exemplar bank. During our research process, we identified the following limitations of VeSX. First, its performance depends on the accuracy of verification and the success rate of reflection and replanning corrections. Although VeSX has achieved significant improvements by relying on these modules, there is still considerable room for enhancement to fully solve web automation tasks. Second, data in the web automation field is relatively scarce. While VeSX has implemented sophisticated designs for the exemplar bank to increase data utilization, generating high-quality data automatically without human annotation remains a significant challenge.

Additionally, the workflow concepts in VeSX may lead to opportunities in other fields. The novel approaches to verification and self-correction could be beneficial for tasks that require long reasoning steps and facilitate the verification of intermediate results. For instance, this could apply to complex mathematical logic reasoning or coding tasks, as well as long-term tasks in simulation environments like Minecraft. VeSX also represents a technological solution for building LLM agents with error-correction capabilities. Our future work will first focus on making new improvements

and contributions in data generation for web automation tasks, and then explore how the VeSX framework can be implemented and utilized in other tasks.

## 6 CONCLUSION

In this paper, we introduce VeSX, a framework of LLM agent for web automation tasks, featured with subgoal-guided verification, hierarchical self-correction, and exemplar bank to deal with the problem about infeasible subtasks and lack of data. Our experiments on the five different scenarios show VeSX have achieved state-of-the-art success rate and significantly outperform existing methods without human guidance. Our work shows the great potential of the workflow paradigm that armed with the planning-execution workflow with verification and self-correction. Moreover, VeSX proposes a new method to generate exemplars' descriptive metadata and constructs a memory system named exemplar bank for in-context learning in planning and execution phase. The ablation studies on the three contributions validate their effectiveness.

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

# A VeSX

## A.1 Workflow

### A.1.1 Flowchart

To help readers better understand VeSX, we have provided a simplified algorithm flowchart of the VeSX workflow. We have emphasized only the most critical modules of the algorithm. Some variables and dependencies are not depicted in the flowchart, as they would make the overall presentation overly lengthy, such as the transmission of error messages after verification, and the control for the environment by reflection and replanning, among others. The construction of the Exemplar Bank is also not represented in this inference workflow.

---

**Algorithm 1:** Simplified VeSX Workflow

---

**Input:** Web Automation Task Query $Q$, initial state $s_0$, LLM agent for planning $LLM_{\text{PLANNING}}$, LLM agent for action prediction $LLM_{\text{ACTION}}$, LLM agent for reflection $LLM_{\text{REFLECTION}}$, LLM agent for replanning $LLM_{\text{REPLANNING}}$, verification module $f_{\text{VERIFICATION}}$, exemplar-bank retrieval module $R$

**Output:** Answer for the query $A$

1   $\mathcal{S} := [(t_1, g_1), (t_2, g_2), ..., (t_n, g_n)] \leftarrow f_{\text{PLANNING}}(Q, s_0)$     ▷ Initial Planning

2   $s \leftarrow s_0, U \leftarrow [\,]$

3   **foreach** $(t^i, g^i)$ *in* $\mathcal{S}$ **do**

4     $C \leftarrow [], m \leftarrow False, v \leftarrow False$

5     **while** *not* $v$ **do**

6       $r \leftarrow R(t, c), a \leftarrow LLM_{\text{ACTION}}(t, s, r)$     ▷ Retrieval and Action Prediction

7       $c \leftarrow A + a$     ▷ Action Chain ("+" indicates concatenate)

8       **if** $a = FINISH\_SUBTASK$ **then**

9         $v \leftarrow f_{\text{VERIFICATION}}(g, s, c)$     ▷ Verification

10         **if** *not* $v$ **then**

11           **if** *not* $m$ **then**

12             $c, s \leftarrow LLM_{\text{REFLECTION}}(t, c, s)$     ▷ Reflection

13             $m \leftarrow True$

14           **end**

15           **else**

16             $[(t^{i+1}, g^{i+1}), ...] \leftarrow LLM_{\text{REPLANNING}}(\mathcal{S})$     ▷ Replanning

17             $\mathcal{S}.update([(t^{i+1}, g^{i+1}), ...])$

18             break

19           **end**

20         **end**

21       **end**

22       **else**

23         $s \leftarrow ENV(s, a)$

24       **end**

25     **end**

26     $u \leftarrow LLM_{\text{SUMMARY}}(t, c, s)$     ▷ Subtask Summary

27     $U \leftarrow U + u$

28   **end**

29   $A \leftarrow LLM_{\text{ANSWER}}(U, s)$     ▷ Get Final Answer

30   **return** A

---

### A.1.2 One Example

Here is a complete execution example of the workflow to clearly illustrate the VeSX process. The section for input prompts is presented in a simplified format to save space and enhance readability. For detailed information on the inputs and outputs of each module, please refer to Appendix A.2.

**One Complete Example**

**Query Intent**:
What is the price range of Canon photo printer in the One Stop Market?
**Ground Truth**
Final Answer: 'must_include': ['2.56', '649.99']

- - - - - - - - - - - - - - - - - - - - - - - - - - - - - - - - - - - - - - - - -

**Planning**
1. Navigate to the "Search" function on the One Stop Market website.
- Objective: check_in_webpage("You have located the 'Search' function")
2. Enter "Canon photo printer" into the search bar and initiate the search.
- Objective: check_in_url("search") |OR|check_in_webpage("Search results for 'Canon photo printer' displayed")
3. Filter the search results to display only products related to "Canon photo printer".
- Objective: check_in_webpage("Search results filtered to display only 'Canon photo printer' products")
4. Identify the prices of all products listed under the "Canon photo printer" category.
- Objective: check_in_webpage("Prices of all 'Canon photo printer' products identified")
5. Determine and display the price range (minimum and maximum prices) for the "Canon photo printer" products.
- Objective: check_in_history("Price range for 'Canon photo printer' products determined and displayed")

- - - - - - - - - - - - - - - - - - - - - - - - - - - - - - - - - - - - - - - - -

**Subtask 1**
1. Navigate to the "Search" function on the One Stop Market website.

**Figure of Observation (only for visualization, not used in input prompt)**:

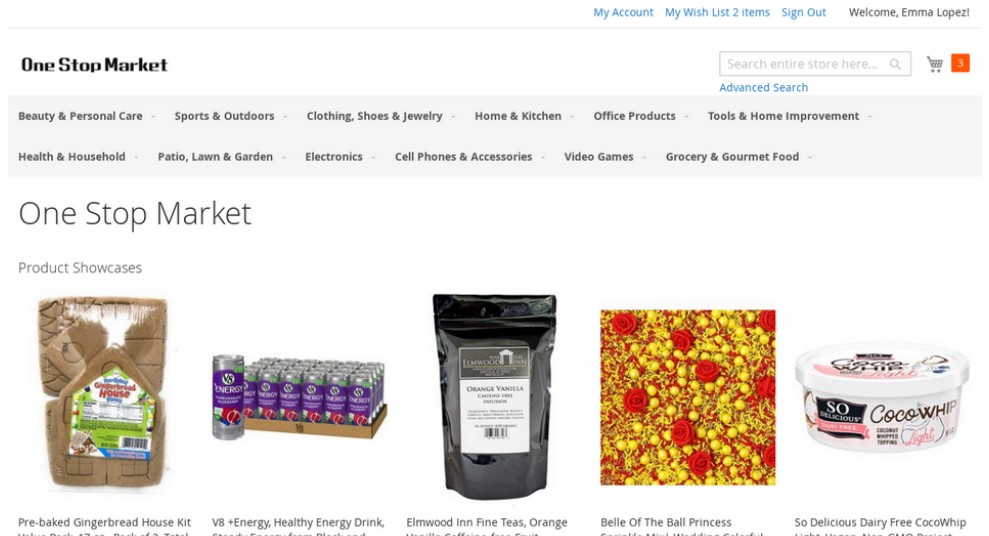

**Analyze and Predict Action**:
Thought: Let's think step-by-step. The current observation shows the One Stop Market homepage with a search combobox and a disabled search button. The first step is to type "white desk" into the search combobox to enable the search button.
Next Action: According to the thoughts, the next action I will perform is "'type [1697] [white desk]"' Target: After typing "white desk" into the search combobox, the search button should be enabled. Summary: The current observation shows the One Stop Market homepage with a search combobox and a disabled search button. The next step is to type "white desk" into the search combobox to enable the search button. The expected outcome is to enable the search button.

**Action: type [1697] [white desk ] where [1697] is combobox 'ue615 Search' autocomplete: both hasPopup: listbox required: False expanded: False**

**Figure of Observation (only for visualization, not used in input prompt)**:

**Analyze and Predict Action**:
Thought: Let's think step-by-step. The current observation shows the search results page for "white desk" on the One Stop Market website. This indicates that the search function has already been navigated to and the search has been performed. Therefore, the subtask to navigate to the "Search" function on the One Stop Market website has been completed.
Next Action: According to the thoughts, the next action I will perform is "'finish_subtask"'. Target: Confirm that the search function has been navigated to and the search results for "white desk" are displayed.
Summary: The current observation shows the search results page for "white desk" on the One Stop Market website. This indicates that the search function has already been navigated to and the search has been performed. Therefore, the subtask to navigate to the "Search" function on the One Stop Market website has been completed.
**Action: finish_subtask []**

**Verification**:
**check_in_webpage("You have located the 'Search' function")**
1.0 To verify if the 'Search' function has been located, I need to check if there is an element related to searching on the webpage. This could include a search bar, search button, or any other search-related elements.
**Verification Pass**

- - - - - - - - - - - - - - - - - - - - - - - - - - - - - - - - - - - - - - - - - -

**Subtask 2**
Enter "Canon photo printer" into the search bar and initiate the search.

**Figure of Observation (only for visualization, not used in input prompt)**:

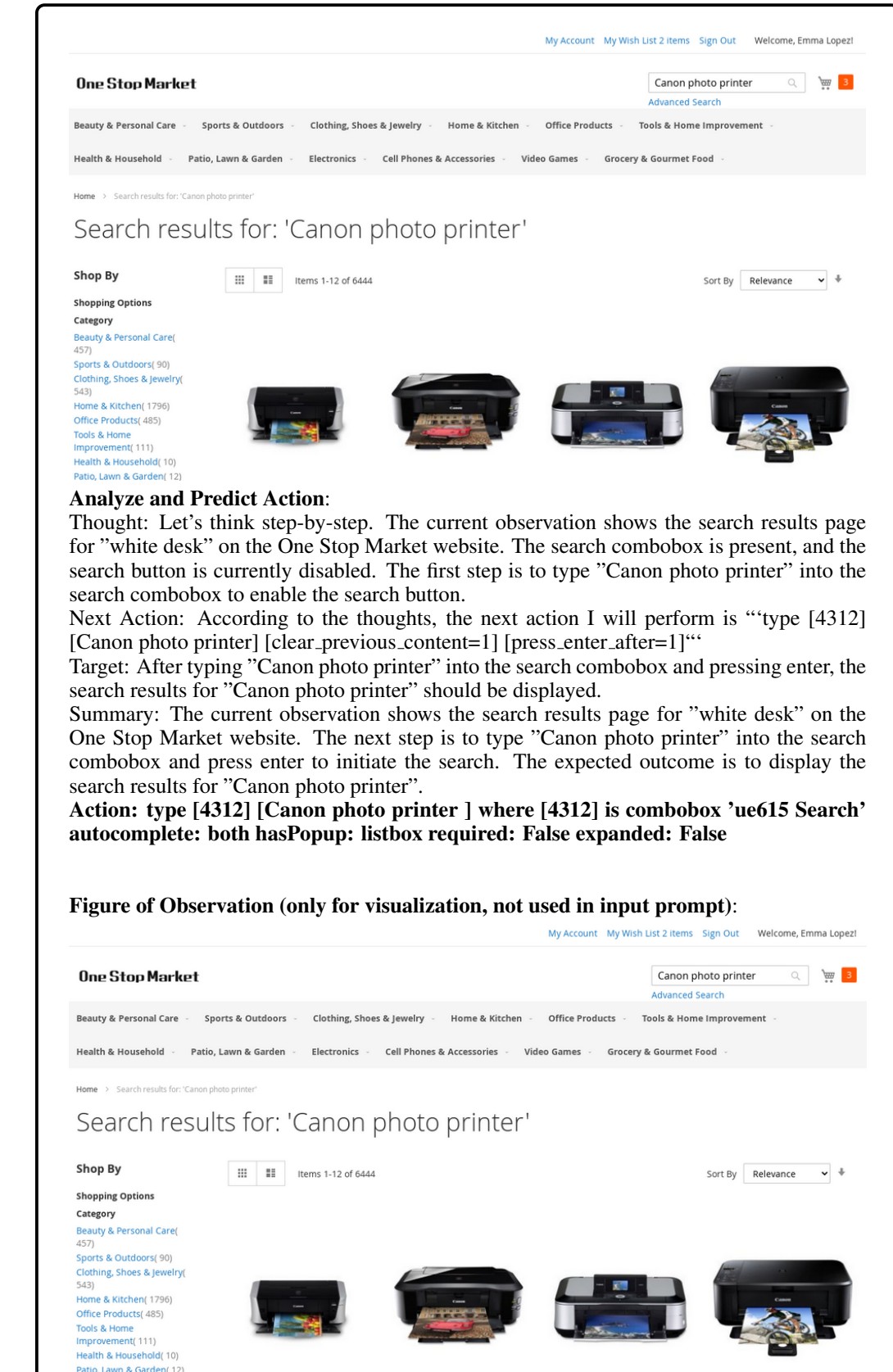

**Analyze and Predict Action**:

Thought: Let's think step-by-step. The current observation shows the search results page for "white desk" on the One Stop Market website. The search combobox is present, and the search button is currently disabled. The first step is to type "Canon photo printer" into the search combobox to enable the search button.

Next Action: According to the thoughts, the next action I will perform is "'type [4312] [Canon photo printer] [clear_previous_content=1] [press_enter_after=1]'"

Target: After typing "Canon photo printer" into the search combobox and pressing enter, the search results for "Canon photo printer" should be displayed.

Summary: The current observation shows the search results page for "white desk" on the One Stop Market website. The next step is to type "Canon photo printer" into the search combobox and press enter to initiate the search. The expected outcome is to display the search results for "Canon photo printer".

**Action: type [4312] [Canon photo printer ] where [4312] is combobox 'ue615 Search' autocomplete: both hasPopup: listbox required: False expanded: False**

**Figure of Observation (only for visualization, not used in input prompt)**:

**Analyze and Predict Action**:
Thought: Let's think step-by-step. The current observation shows the search results for "Canon photo printer" on the One Stop Market website. This indicates that the search function has already been navigated to and the search has been performed. Therefore, the subtask to enter "Canon photo printer" into the search bar and initiate the search has been completed. Next Action: According to the thoughts, the next action I will perform is "'finish_subtask"'.
Target: Confirm that the search results for "Canon photo printer" are displayed.
Summary: The current observation shows the search results for "Canon photo printer" on the One Stop Market website. This indicates that the search function has already been navigated to and the search has been performed. Therefore, the subtask to enter "Canon photo printer" into the search bar and initiate the search has been completed.
**Action: finish_subtask []**

**Verification**:
Objective: check_in_url("search") |OR|check_in_webpage("Search results for 'Canon photo printer' displayed"
**check_in_url("search")**
1.0 URL: http://127.0.0.1:7770/catalogsearch/result/?q=Canon+photo+printer, "search" is in URL
**check_in_webpage("Search results for 'Canon photo printer' displayed"**
1.0 The content of the webpage includes multiple instances of the phrase "Search results for: 'Canon photo printer'" and related elements such as headings and static text that confirm the search results are displayed
**Verification Pass**

- - - - - - - - - - - - - - - - - - - - - - - - - - - - - - - - - - - - - - - - - - - - - - - -

**Subtask 3**
Filter the search results to display only products related to "Canon photo printer".

**Figure of Observation (only for visualization, not used in input prompt)**:

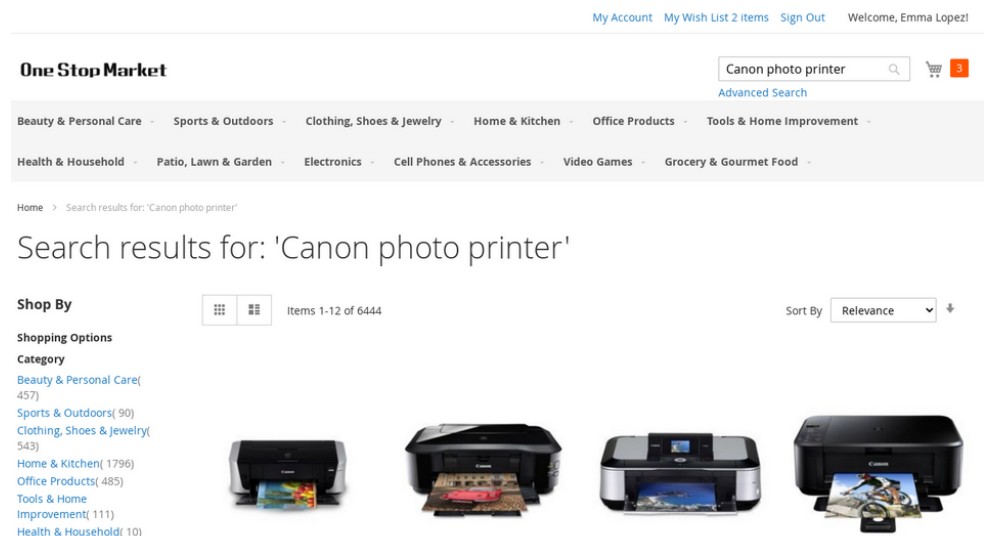

**Analyze and Predict Action**:
Thought: Let's think step-by-step. The current observation shows that I am on the search results page for the query "Canon photo printer." To filter the search results to display only products related to "Canon photo printer," I need to ensure that the search results are already filtered correctly. The page title and the search results indicate that the products displayed are related to "Canon photo printer."
Next Action: According to the thoughts, the next action I will perform is "'finish_subtask"'

Target: Confirm that the search results are already filtered to display only products related to "Canon photo printer." Summary: The current observation shows that the search results page is already displaying products related to "Canon photo printer." Therefore, the subtask has been completed.
**Action: finish_subtask []**

**Verification**:
**check_in_webpage("Search results filtered to display only 'Canon photo printer' products")**
0.0 I need to verify if the search results are specifically filtered to display only 'Canon photo printer' products. This can be determined by checking the search results and ensuring that they are relevant to 'Canon photo printer'.
**Verification Fail**
**Reflection**
Reflection: The execution process correctly identified that the search results page is displaying products related to "Canon photo printer." However, the verification result indicates that there needs to be a check to ensure that the search results are specifically filtered to display only "Canon photo printer" products.
The mistake lies in not explicitly verifying the relevance of each product in the search results to "Canon photo printer." To solve the task, I should include a step to verify that all the products listed in the search results are indeed related to "Canon photo printer." This can be done by checking the titles and descriptions of the products to ensure they match the search criteria.
Since the mistake occurred after identifying the search results, I should reload from the point just before finishing the subtask to include the verification step.
Mistakes happened during verification. There is no need to retry.
**Verification (changed to) Pass**

- - - - - - - - - - - - - - - - - - - - - - - - - - - - - - - - - - - - - - - - - - - - - - - - - -

**Subtask 4**
Identify the prices of all products listed under the "Canon photo printer" category.

**Figure of Observation (only for visualization, not used in input prompt)**:

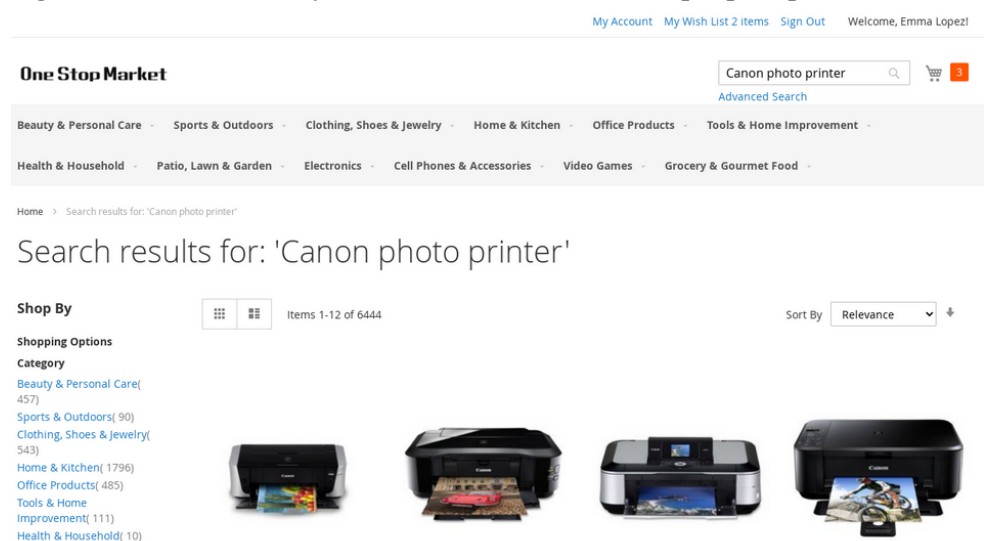

**Analyze and Predict Action**:
Thought: Let's think step-by-step. The current subtask is to identify the prices of all products listed under the "Canon photo printer" category. The current observation shows the search

results for "Canon photo printer" on the One Stop Market website. I need to locate the prices of the products listed under this category.

Next Action: According to the thoughts, the next action I will perform is "'scroll [direction=down]'"

Target: Scroll down the page to view more products and their prices.

Summary: I will scroll down the page to view more products listed under the "Canon photo printer" category and their prices.

**Action: scroll [down]**

**Figure of Observation (only for visualization, not used in input prompt)**:

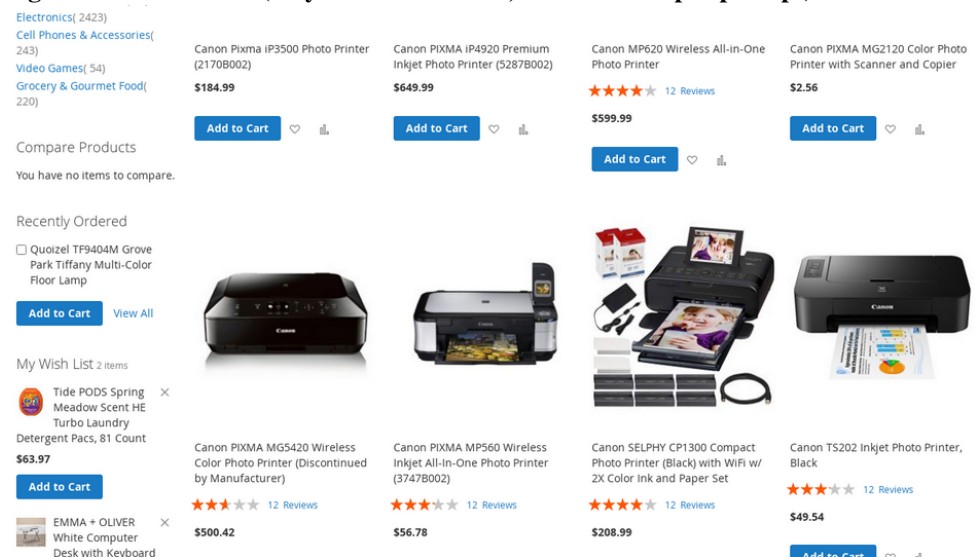

**Analyze and Predict Action**:

Thought: Let's think step-by-step. I have identified several products under the "Canon photo printer" category along with their prices. Here are the products and their prices: 1. Canon Pixma iP3500 Photo Printer - $184.99 2. Canon PIXMA iP4920 Premium Inkjet Photo Printer - $649.99 3. Canon MP620 Wireless All-in-One Photo Printer - $599.99 4. Canon PIXMA MG2120 Color Photo Printer with Scanner and Copier - $2.56 5. Canon PIXMA MG5420 Wireless Color Photo Printer - $500.42 6. Canon PIXMA MP560 Wireless Inkjet All-In-One Photo Printer - $56.78 7. Canon SELPHY CP1300 Compact Photo Printer - $208.99 8. Canon TS202 Inkjet Photo Printer - $49.54

I have identified the prices of all products listed under the "Canon photo printer" category.

Next Action: According to the thoughts, the next action I will perform is "'finish_subtask [I have identified the prices of all products listed under the "Canon photo printer" category: Canon Pixma iP3500 Photo Printer - $184.99, Canon PIXMA iP4920 Premium Inkjet Photo Printer - $649.99, Canon MP620 Wireless All-in-One Photo Printer - $599.99, Canon PIXMA MG2120 Color Photo Printer with Scanner and Copier - $2.56, Canon PIXMA MG5420 Wireless Color Photo Printer - $500.42, Canon PIXMA MP560 Wireless Inkjet All-In-One Photo Printer - $56.78, Canon SELPHY CP1300 Compact Photo Printer - $208.99, Canon TS202 Inkjet Photo Printer - $49.54]'"

Target: I expect to complete the subtask of identifying the prices of all products

**Action: finish_subtask [I have identified the prices of all products listed under the "Canon photo printer" category: Canon Pixma iP3500 Photo Printer - $184.99, Canon PIXMA iP4920 Premium Inkjet Photo Printer - $649.99, Canon MP620 Wireless All-in-One Photo Printer - $599.99, Canon PIXMA MG2120 Color Photo Printer with Scanner and Copier - $2.56, Canon PIXMA MG5420 Wireless Color Photo Printer - $500.42, Canon PIXMA MP560 Wireless Inkjet All-In-One Photo Printer - $56.78, Canon SELPHY CP1300 Compact Photo Printer - $208.99, Canon TS202 Inkjet Photo Printer - $49.54]**

**Verification**:
**check_in_webpage("Prices of all 'Canon photo printer' products identified")**
1.0 Prices of all 'Canon photo printer' products have been listed and identified.
**Verification Pass**

**Subtask 5**
Determine and display the price range (minimum and maximum prices) for the "Canon photo printer" products.

**Figure of Observation (only for visualization, not used in input prompt)**:

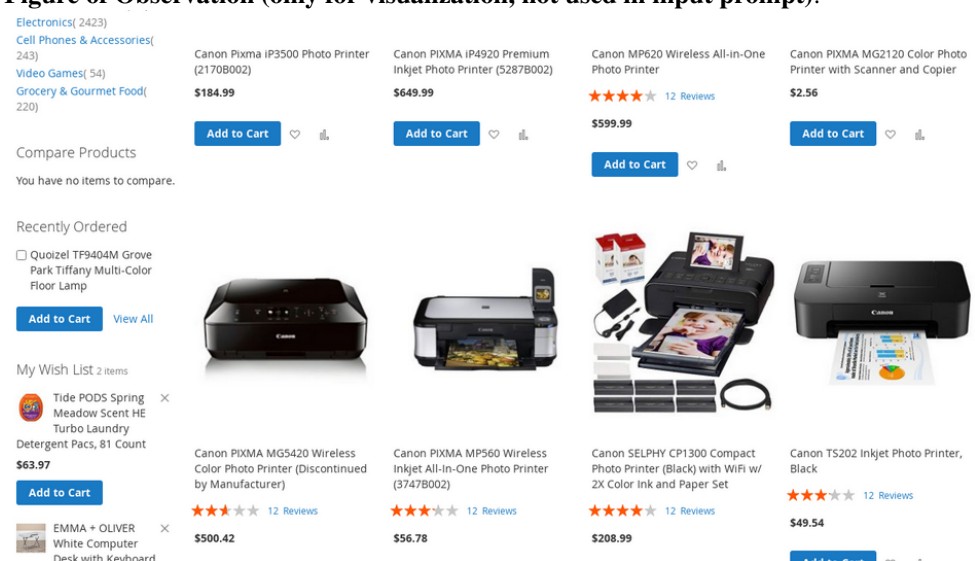

**Analyze and Predict Action**:
Thought: Let's think step-by-step. The current subtask is to determine and display the price range (minimum and maximum prices) for the "Canon photo printer" products. The current observation shows multiple Canon photo printers with their respective prices. I need to identify the minimum and maximum prices from the listed products.
Here are the prices of the listed Canon photo printers: - $184.99 - $649.99 - $599.99 - $2.56 - $500.42 - $56.78 - $208.99 - $49.54
The minimum price is $2.56, and the maximum price is $649.99.
Next Action: According to the thoughts, the next action I will perform is "'finish_subtask [2.56, 649.99]'"
Target: Confirm the price range (minimum and maximum prices) for the "Canon photo printer" products.
Summary: The prices of the listed Canon photo printers have been identified, with the minimum price being $2.56 and the maximum price being $649.99. Therefore, the subtask has been completed.
**Action: finish_subtask [2.56, 649.99]**

**Verification**:
**check_in_history("Price range for 'Canon photo printer' products determined and displayed")**
1.0 The subgoal target is to determine and display the price range for 'Canon photo printer' products. The summary indicates that the prices have been identified, with a minimum price of $2.56 and a maximum price of $649.99. This information aligns with the subgoal target.
**Verification Pass**

> **Answer**:
> The price range of Canon photo printers in the One Stop Market is from $2.56 to $649.99.

## A.2 DETAILS OF EACH MODULE

We provided a smooth example of the VeSX workflow for handling a web automation task to help readers understand it better. The query for this task is ****. From the workflow below, we can see how VeSX breaks down the task into individual subtasks, addresses each one separately, and verifies the results.

### A.2.1 EXAMPLE OF PLANNING PHASE

---

**Example of Planning Phase**

**Query Intent**
List out reviewers, if exist, who mention about complain of the customer service.

- - - - - - - - - - - - - - - - - - - - - - - - - - - - - - - - - - - - - - - - - - - - - - - - - -

**System Prompt**:
You are an autonomous intelligent agent tasked with navigating a web browser. Your responsibility entails performing specific web-based tasks, beginning from a defined starting point on a particular webpage. To ensure the effectiveness and tractability of your operations, you are to break down given task into manageable and coherent sub-tasks.
Each subtask you delineate should be precise, methodical, and feasible, though not necessarily to the extent of a single action or click. You should not use one action like click something as your subtask, but use navigate to some website or locate some button or function. Aim to limit the total number of sub-tasks to a maximum of five.
Additionally, for each sub-task stated, be sure to set clear and verifiable objectives. These objectives will act as benchmarks for assessing the successful completion of each step and written after each sub-task. The objective should be a form of function call, where function should be one of the following three:
1. check_in_url(part of url): You think you will check in the url. This often be used when you want to navigate to some website. Its parameter is the fragment you expect to be the part of the url.
2. check_in_webpage(objective): You think you will check the objective in the last webpage. This often be used when the last webpage has the information for your sub-task. Its parameter is the objective you want to achieve in the sub-task and can be check in the webpage.
3. check_in_history(objective): You think you should check in your action history. This often be used when your sub-task is to get some information. Its parameter is the objective you want to achieve in the sub-task and can be check in the history.
You can use '|OR|' to connect two objective if you want to use multiple methods to assess the result of subtasks. You should add objectives after each sub-task by starting with 'Objective'. In situations where authentication is necessary, account credentials will be provided to you subsequently.
Please proceed to methodically dissect the following task into its constituent sub-tasks, and articulate verifiable objectives for each to ensure their effective execution.

- - - - - - - - - - - - - - - - - - - - - - - - - - - - - - - - - - - - - - - - - - - - - - - - - -

**Example User**:
Start webpage: Current Observation: The webpage does not provide any details about reviewers mentioning the price being unfair. There is no information related to reviewers or their opinions on prices in the provided representation.
URL: http://onestopmarket.com/haflinger-men-s-wool-felt-open-back-slippers-beige-550-peat-us-7.html
Task: List out reviewers, if exist, who mention about price being unfair
Now, try to break this task down into several sub-tasks and generate related verifiable objectives:

---

**Example Assistant**:
1. **Navigate to the Reviews Section** - Locate and click on the "2 Reviews" link to access the reviews section.
- Objective: `check_in_url("reviews") |OR|check_in_webpage("Reviews")`
2. **Extract All Reviews** - Identify and extract the text of all reviews present on the page.
- Objective: `check_in_webpage("Review Text")`
3. **Search for Mentions of Price Being Unfair** - Analyze the extracted reviews to find any mentions of the price being unfair.
- Objective: `check_in_history("price being unfair")`
4. **List Out Reviewers** - Identify and list the reviewers who mentioned the price being unfair.
- Objective: `check_in_history("Reviewers mentioning price being unfair")`
5. **Compile and Present the Information** - Compile the information into a coherent list and present it.
- Objective: `check_in_history("Compiled list of reviewers")`

- - - - - - - - - - - - - - - - - - - - - - - - - - - - - - - - - - - -

**Example User**: Start webpage: Current Observation: The webpage mentions "Current Observation: One Stop" but does not provide a customer service number in the given text span.
URL: http://onestopmarket.com/
Task: Which number to call for the customer service? Now, try to break this task down into several sub-tasks and generate related verifiable objectives:

**Example Assistant**:
1. Navigate to the "Contact Us" or "Customer Service" section of the One Stop Market website.
Objective: check_in_url("contact") |OR|check_in_webpage("Customer Service")
2. Locate the customer service phone number on the "Contact Us" or "Customer Service" page.
Objective: check_in_webpage("Customer service phone number located")
3. Verify the customer service phone number by cross-referencing it with any additional information provided on the page (e.g., business hours, email support).
Objective: check_in_webpage("Customer service phone number verified with additional information")

- - - - - - - - - - - - - - - - - - - - - - - - - - - - - - - - - - - -

**Example User**: Start webpage: Current Observation: The actree representation includes customer reviews for the Amazon Echo Dot 3rd generation. Key points from the reviews highlight positive aspects such as improved sound quality, ease of setup, and useful integration with other smart devices. Customers appreciate the compact design and functionality, particularly the enhancement over previous versions. Some reviews also mention the device's good value for the price. Criticisms in the reviews include occasional issues with voice recognition and connectivity problems. Overall, the reviews provide a generally favorable outlook on the Amazon Echo Dot 3rd generation.
URL: http://onestopmarket.com/
Task: Summarize customer reviews for Amazon Echo Dot 3rd generation. Now, try to break this task down into several sub-tasks and generate related verifiable objectives:

**Example Assistant**:
1. Navigate to the product page for the Amazon Echo Dot 3rd generation on the One Stop Market website. Objective: check_in_url("amazon-echo-dot-3rd-generation") |OR|check_in_webpage("Amazon Echo Dot 3rd generation product page")
2. Locate the section containing customer reviews for the Amazon Echo Dot 3rd generation. Objective: check_in_webpage("Customer reviews for Amazon Echo Dot 3rd generation located")

3.Read through the customer reviews to identify key positive aspects mentioned. Objective: check_in_webpage("Positive aspects of Amazon Echo Dot reviews summarized")
4. Read through the customer reviews to identify key criticisms mentioned. Objective: check_in_webpage("Criticisms of Amazon Echo Dot reviews summarized")
5. Compile the summarized information into a coherent overview of customer reviews for the Amazon Echo Dot 3rd generation. Objective: check_in_webpage("Customer reviews summary for Amazon Echo Dot 3rd generation compiled")

- - - - - - - - - - - - - - - - - - - - - - - - - - - - - - - - - - - - - - - - - - - - - - - - - - - -

**User**:
Start webpage: Current Observation: Tab 0 (current): Epson WorkForce WF-3620 WiFi Direct All-in-One Color Inkjet Printer, Copier, Scanner, Amazon Dash Replenishment Ready
[1] RootWebArea 'Epson WorkForce WF-3620 WiFi Direct All-in-One Color Inkjet Printer, Copier, Scanner, Amazon Dash Replenishment Ready' focused: True
    [1428] link 'My Account'
    [1424] link 'My Wish List 1 item'
    [1430] link 'Sign Out'
    [1433] StaticText 'Welcome, Emma Lopez!'
    [1330] link 'Skip to Content'
    [1339] link 'store logo'
      [1436] img 'one_stop_market_logo'
    [1437] link 'ue611 My Cart 4 4items'
    [2496] StaticText 'Search'
    [1756] combobox 'ue615 Search' autocomplete: both hasPopup: listbox required: False expanded: False
    [2499] link 'Advanced Search'
    [1759] button 'Search' disabled: True
    [1135] tablist '' multiselectable: False orientation: horizontal
      [1137] tabpanel ''
        [79] menu '' orientation: vertical
          [830] menuitem 'ue622 Beauty & Personal Care' hasPopup: menu
          [895] menuitem 'ue622 Sports & Outdoors' hasPopup: menu
          [905] menuitem 'ue622 Clothing, Shoes & Jewelry' hasPopup: menu
          [919] menuitem 'ue622 Home & Kitchen' hasPopup: menu
          [956] menuitem 'ue622 Office Products' hasPopup: menu
          [964] menuitem 'ue622 Tools & Home Improvement' hasPopup: menu
          [969] menuitem 'ue622 Health & Household' hasPopup: menu
          [975] menuitem 'ue622 Patio, Lawn & Garden' hasPopup: menu
          [980] menuitem 'ue622 Electronics' hasPopup: menu
          [1041] menuitem 'ue622 Cell Phones & Accessories' hasPopup: menu
          [1056] menuitem 'ue622 Video Games' hasPopup: menu
          [1069] menuitem 'ue622 Grocery Gourmet Food' hasPopup: menu
    [1365] link 'Home'
    [1368] StaticText 'Epson WorkForce WF-3620 WiFi Direct All-in-One Color Inkjet Printer, Copier, Scanner, Amazon Dash Replenishment Ready'
    [5] main ''
      [1369] heading 'Epson WorkForce WF-3620 WiFi Direct All-in-One Color Inkjet Printer, Copier, Scanner, Amazon Dash Replenishment Ready'
      [11] generic 'Availability'
        [13] StaticText 'IN STOCK'
      [1445] StaticText 'SKU'
      [1715] StaticText 'B00JXLGF06'
      [1376] LayoutTable ''
        [1717] StaticText 'Rating:'
        [1448] generic '35%'
          [2469] StaticText '% of'
          [2472] StaticText '100'
          [1449] link '12xa0 Reviews '

```
          [1450] link 'Add Your Review'
      [1452] StaticText '$607.39'
      [1391] LayoutTable ''
          [1731] StaticText 'Qty'
          [1732] spinbutton 'Qty' required: False valuemin: 0 valuemax: 0 valuetext:
          [1733] button 'Add to Cart'
      [1393] link 'Add to Wish List'
      [1394] link 'Add to Compare'
      [1399] link 'Skip to the end of the images gallery'
      [1156] button 'Previous'
      [1158] generic 'Image'
          [2502] img 'Image'
      [1157] button 'Next'
```
URL:     http://onestopmarket.com/epson-workforce-wf-3620-wifi-direct-all-in-one-color-inkjet-printer-copier-scanner-amazon-dash-replenishment-ready.html
Task: List out reviewers, if exist, who mention about complain of the customer service
Now, try to break this task down into several sub-tasks and generate related verifiable objectives: "

- - - - - - - - - - - - - - - - - - - - - - - - - - - - - - - - - - - - - - - - - - - - -

**Output**:
1. **Navigate to the Reviews Section** - Locate and click on the "12 Reviews" link to access the reviews section.
- Objective: `check_in_url("reviews") |OR|check_in_webpage("Reviews")`
2. **Extract All Reviews** - Identify and extract the text of all reviews present on the page.
- Objective: `check_in_webpage("Review Text")`
3. **Search for Mentions of Customer Service Complaints** - Analyze the extracted reviews to find any mentions of complaints about customer service.
- Objective: `check_in_history("customer service complaints")`
4. **List Out Reviewers** - Identify and list the reviewers who mentioned complaints about customer service.
- Objective: `check_in_history("Reviewers mentioning customer service complaints")`
5. **Compile and Present the Information** - Compile the information into a coherent list and present it.
- Objective: `check_in_history("Compiled list of reviewers")`

A.2.2  EXAMPLE OF EXECUTION PHASE

Example of Execution Phase

**System Prompt**:
You are an autonomous intelligent agent tasked with navigating a web browser. You will be a given web-based task. This task will be decomposed into some sub-tasks and you will achieve the final objective of this final task by solving these sub-tasks one by one. These sub-tasks will be accomplished through the use of specific actions you can issue.
Here's the information you'll have:
The user's objective: This is the task you're trying to complete.
The current web page's accessibility tree: This is a simplified representation of the webpage, providing key information.
The current web page's URL: This is the page you're currently navigating.
The open tabs: These are the tabs you have open.
The webpage-browsing and the action history: This part contains the history of webpage you browsed and action you performed on them. The contents of these webpages are refined to decrease the context length. It may be helpful to track your progress and check whether you encountered any onstacles.
The actions you can perform fall into several categories:
Page Operation Actions:

'click [id]': This action clicks on an element with a specific id on the webpage.

'type [id] [content] [clear_previous_content=0—1] [press_enter_after=0—1]': Use this to type the content into the field with id. By default, the web brower will clear the previous content. If you want to input something to replace the original content, for instance when you need to change a value, set clear_previous_content to 1; when you want to add something, set it to 0. By default, the "Enter" key is pressed after typing unless press_enter_after is set to 0.

'hover [id]': Hover over an element with id.

'press [key_comb]': Simulates the pressing of a key combination on the keyboard (e.g., Ctrl+v).

'scroll [direction=down—up]': Scroll the page up or down.

Tab Management Actions:

'new_tab': Open a new, empty browser tab.

'tab_focus [tab_index]': Switch the browsers focus to a specific tab using its index.

'close_tab': Close the currently active tab.

URL Navigation Actions:

'go_back': Navigate to the previously viewed page.

'go_forward': Navigate to the next page (if a previous go_back action was performed).

'reload': Reload this web page (if you encountered an error, e.g. 500, 502, in this web page).

Completion Action:

'finish_subtask [answer]': Issue this action when you believe this subtask is finished and then you will check the correction and comopletion of this subtask. If the objective is to find a text-based answer, provide the answer in the bracket.

'stop [answer]': Issue this action when you believe the task is complete. If the objective is to find a text-based answer, provide the answer in the bracket.

Homepage:

If you want to visit other websites, check out the homepage at http://homepage.com by using 'goto [http://homepage.com]'. It has a list of websites you can visit.

http://homepage.com/password.html lists all the account name and password for the websites. You can use them to log in to the websites.

To be successful, it is very important to follow the following rules:

1. You should only issue an action that is valid given the current observation and you should only issue one action at a time. Your action must be one of the actions listed before. When you want to propose an action that is the same as previous steps, consider whether it is still necessary, as it has already been attempted.

2. You should follow the examples to reason step by step and then issue the next action. PAY ATTENTION!!! : You should first consider whether the subtask has been completed by previous actions or it is unnecessary to tackle with. If it is completed, please promptly issue finish_subtask. When you have completed a series of actions and largely achieved the subtask, but feel uncertain or have doubts about the results, you can also trigger finish_subtask to perform verification first. Especially when you want to re-execute some actions you have done in the past, please be particularly cautious. Do not simply redo the actions because of doubts; you need to confirm that the previous actions had no effect before proceeding.

3. Issue 'finish_subtask' action when you think you have accomplished the sub-task. Issue stop action when you think you have achieved the final objective of the whole task. When you think you have accomplished the subtask and the whole task at the same time, issue step action to generate your final answer. Sometimes the current subtask has already been completed in a distant previous action, not necessarily in recent actions. Feel free to issue "finish subtask" when you deem it appropriate.

4. The history contains two part: global history is the summaries of the previous sub-tasks aimed to accomplish the final whole task, and local history is the summaries of the actions you issued to accomplish this sub-task. You should pay more attention to the last step of local history and you should check if there is a target in it.

5. You should solve the sub-task with these parts: Thought, Next Action, Target and Summary. 'Thought' can reason the current situation step by step and help you generate next action. And if there is a target in the last previous local history, you should check whether it is achieved according to the observation in this part 'Thought. 'Next Action' is the action

you choose to take, which are listed before. PAY ATTENTION!!! : You should generate the action in the correct format faithfully. Start with a "According to the thoughts, the next action I will perform is" phrase, followed by action inside "' "'. For example, "According to the thoughts, the next action I will perform is "'click [1234]"'". The action in wrong format will be rejected and cause the process failed, so PLEASE GENERATE IN THE CORRECT FORMAT. 'Target' is what you expect to do or what content you expect to see after the 'Next Action' be performed. 'Summary' is the summary of this step, and it should contains the observation of the web page, the thoughts, the predicted action and the target you generated.
6. When you need to publish certain content, especially when posting on Reddit or creating a forum, you must fill out all the required fields first. These fields often appear in the format of textbox: "xxx", required: True. You need to complete all these fields before making the submission. Relevant examples include: when creating a website, the required fields are Name, Title, Description, and Sidebar; when posting, the required fields are Title, Body, and Forum.
7. If you need to sign in, use these accounts:
[omit]

- - - - - - - - - - - - - - - - - - - - - - - - - - - - - - - - - - - - - - - - - - - - - - - - - - - - - - - -

**Example User**:
Current Subtask:
1. **Navigate to the "My Account" section** - Click on the "My Account" link.
Recap of Recent Actions:
There is no local history. This is the first step to solve the current subtask.
Current Observation:
The webpage titled "One Stop Market" showcases an e-commerce platform with various sections and features accessible via links and menu items. Key elements include:
- **Header Links**:
- "My Account" (Link to user account)
- "My Wish List 2 items"
- "Sign Out"
- "Welcome, Emma Lopez!" (Greeting text)
- **Navigational Links**:
- "Skip to Content"
- "store logo" (with an embedded store logo image)
- "My Cart 3 items"
- **Search Functionality**:
- "Search" field (with options for autocomplete and advanced search)
- "Search" button (currently disabled)
- **Tab List and Menu**: A horizontal tab panel with various categories such as Beauty & Personal Care, Sports & Outdoors, and more, each with a submenu.
**Main Content Area**:
- "One Stop Market" heading
- "Product Showcases" section featuring several product links and images.
To **navigate to the "My Account" section**, the user should click on the "My Account" link, which is represented as:
"'plaintext
[1296] link ́My Account
"'
URL: http://onestopmarket.com/'

**Example Assistant**
Thought: Let ́s think step-by-step. The current observation indicates that I am on the One Stop Market homepage. The page contains a ́My Account ́link with ID [1296]. To navigate to the "My Account" page, I need to click on this link.
Next Action: According to the thoughts, the next action I will perform is "'click [1296]"'
Target: Navigate to the "My Account" page.

Summary: The current observation shows that I am on the One Stop Market homepage, and there is a ́My Account ́link. I will click on this link to navigate to the "My Account" page.

- - - - - - - - - - - - - - - - - - - - - - - - - - - - - - - - - - - - - - - - - - - - - - -

**Example User**
Current Subtask:
1. **Navigate to the "My Account" section**
- Click on the "My Account" link.
Recap of Recent Actions:
Step: 1
The summary of this step is:
The current observation shows that I am on the One Stop Market homepage, and there is a ́My Account ́link. I will click on this link to navigate to the "My Account" page.
You have taken action 'click [button_id] where [button_id] is link ́My Account ́' in this step.
Current Observation:
The webpage appears to be structured as a user account management page titled "My Account," which is already focused. Key elements include various links and static text related to user account information and navigation.
1. **Navigate to the "My Account" section**
- To navigate to the "My Account" section, you would click on the link labeled "My Account" ([4153] link ́My Account ́).
Currently, the page already shows the "My Account" section with relevant account information for the user Emma Lopez, including contact details, billing and shipping addresses, and links to managing various aspects of the account, such as editing contact information, changing the password, and managing addresses ([4064] group for Default Billing Address and [4081] group for Default Shipping Address). The page navigation elements offer access to the wish list, newsletter subscriptions, stored payment methods, and product reviews.
URL: http://onestopmarket.com/customer/account/'

**Example Assistant**:
Thought: Let ́s think step-by-step. The current observation indicates that I have successfully navigated to the "My Account" page. The page contains various elements such as account information, contact information, address book, and links to other account-related sections. Since the objective was to navigate to the "My Account" page, I have achieved this subtask.
Next Action: According to the thoughts, the next action I will perform is "'finish_subtask"'
Target: Confirm that the subtask of navigating to the "My Account" page is complete.
Summary: The current observation shows that I am on the "My Account" page, which contains various account-related information and links. Therefore, I will finish the subtask as it has been successfully completed.

- - - - - - - - - - - - - - - - - - - - - - - - - - - - - - - - - - - - - - - - - - - - - - -

**Example User**:
Current Subtask:
1. **Navigate to the Reviews Section**
- Locate and click on the "2 Reviews" link to access the reviews section.
Recap of Recent Actions:
Step: 1
The summary of this step is:
The current observation shows that I am on the product page for "HAFLINGER Men ́s Wool Felt Open Back Slippers," and there is a link labeled "2 Reviews" with ID [1357]. Therefore, I will click on this link to navigate to the reviews section.
You have taken action 'click [button_id] where [button_id] is link ́2xa0 Reviews ́' in this step.
Current Observation:
Summary of Webpage for Task Requirements
**Page Title:**
- HAFLINGER Men ́s Wool Felt Open Back Slippers, Beige 550 Peat, US:7
**Content Structure:**

1. **Main Web Area**
- **Tab List Details:**
- There are two main tabs:
- **Details**
- Controlled by a link labeled D́etails
- Not expanded or selected
- **Reviews (2)**
- Controlled by a link labeled Ŕeviews (2)
- Currently expanded and selected
2. **Reviews Section**
- **Customer Reviews Panel:**
- Contains StaticText elements:
- "Customer Reviews"
- "Wonderful!"
- Specific review details like "Rating: 100%", "Nothing.....everyday use."
- Reviewers: J́ayánd J́osef Bürger
- Review dates: "4/18/23"
- Statements such as "Die Haflinger sind wie beschrieben und erwartet sehr gut"
- **Review Form:**
- Group titled: "Youŕe reviewing: HAFLINGER Menś Wool Felt Open Back Slippers, Beige 550 Peat, US:7"
- Rating options (1 to 5 stars) with radio buttons
- Required textbox for Ńicknameĺabeled "Emma"
**Task Compliance:**
To navigate to the Reviews Section:
- **Locate and click the "Reviews (2)" link (Index [1374])**, which controls the Reviews tabpanel. This will display the reviews section.
**Key Actions for Users:**
1. Identify the "Reviews (2)" tab under the tablist.
2. Ensure the "Reviews (2)" is expanded and selected to access customer reviews.
This structure ensures that the reviews, including the two customer feedback entries and required rating fields for submitting a new review, are reached effectively.
URL: http://onestopmarket.com/haflinger-men-s-wool-felt-open-back-slippers-beige-550-peat-us-7.html

**Example Assistant**:
Thought: Letś think step-by-step. The current observation shows that I am on the product page for "HAFLINGER Menś Wool Felt Open Back Slippers," and the "Reviews (2)" tab is expanded and selected. The reviews section is visible, showing customer reviews and a form for submitting a new review. This means that the "2 Reviews" link has been successfully clicked and I have navigated to the reviews section. Therefore, I have completed the subtask.
Next Action: According to the thoughts, the next action I will perform is "'finish_subtask"'
Target: Confirm that the subtask of navigating to the reviews section by clicking on the "2 Reviews" link is completed.
Summary: The current observation shows that I am on the product page for "HAFLINGER Menś Wool Felt Open Back Slippers," and the "Reviews (2)" tab is expanded and selected. The reviews section is visible, showing customer reviews and a form for submitting a new review. This confirms that the "2 Reviews" link has been successfully clicked and I have navigated to the reviews section. Therefore, I will finish the subtask.

- - - - - - - - - - - - - - - - - - - - - - - - - - - - - - - - - - - - - - - -

**User**:
Current Subtask:
1. **Navigate to the Reviews Section**
- Locate and click on the "12 Reviews" link to access the reviews section.
Recap of Recent Actions:

There is no local history. This is the first step to solve the current subtask.
Current Observation:
Tab 0 (current): Epson WorkForce WF-3620 WiFi Direct All-in-One Color Inkjet Printer, Copier, Scanner, Amazon Dash Replenishment Ready
[1] RootWebArea Épson WorkForce WF-3620 WiFi Direct All-in-One Color Inkjet Printer, Copier, Scanner, Amazon Dash Replenishment Readýfocused: True
  [1428] link My Account
  [1424] link My Wish List 1 item
  [1430] link Sign Out
  [1433] StaticText Welcome, Emma Lopez!
  [1330] link Skip to Content
  [1339] link store logo
    [1436] img one_stop_market_logo
  [1437] link úe611 My Cart 4 4items
  [2496] StaticText Search
  [1756] combobox úe615 Search́autocomplete: both hasPopup: listbox required: False expanded: False
  [2499] link Advanced Search
  [1759] button Search́disabled: True
  [1135] tablist ˊˊ multiselectable: False orientation: horizontal
    [1137] tabpanel ˊˊ
      [79] menu ˊˊ orientation: vertical
        [830] menuitem úe622 Beauty & Personal CaréhasPopup: menu
        [895] menuitem úe622 Sports & OutdoorśhasPopup: menu
        [905] menuitem úe622 Clothing, Shoes & JewelrýhasPopup: menu
        [919] menuitem úe622 Home & KitcheńhasPopup: menu
        [956] menuitem úe622 Office ProductśhasPopup: menu
        [964] menuitem úe622 Tools & Home Improvement́hasPopup: menu
        [969] menuitem úe622 Health & Household́hasPopup: menu
        [975] menuitem úe622 Patio, Lawn & GardeńhasPopup: menu
        [980] menuitem úe622 ElectronicśhasPopup: menu
        [1041] menuitem úe622 Cell Phones & AccessorieśhasPopup: menu
        [1056] menuitem úe622 Video GameśhasPopup: menu
        [1069] menuitem úe622 Grocery & Gourmet Food́hasPopup: menu
  [1365] link Home
  [1368] StaticText Épson WorkForce WF-3620 WiFi Direct All-in-One Color Inkjet Printer, Copier, Scanner, Amazon Dash Replenishment Ready
  [5] main ˊˊ
    [1369] heading Épson WorkForce WF-3620 WiFi Direct All-in-One Color Inkjet Printer, Copier, Scanner, Amazon Dash Replenishment Ready
    [11] generic Ávailability
      [13] StaticText ÍN STOCK
    [1445] StaticText ŚKU
    [1715] StaticText B́00JXLGF06
    [1376] LayoutTable ˊˊ
      [1717] StaticText Rating:
      [1448] generic 35%
        [2469] StaticText % of
        [2472] StaticText 100
      [1449] link 12
xa0 Reviews
      [1450] link Add Your Review
    [1452] StaticText $607.39
    [1391] LayoutTable ˊˊ

[1731] StaticText Q́ty

[1732] spinbutton Q́týrequired: False valuemin: 0 valuemax: 0 valuetext:

[1733] button Ádd to Cart

[1393] link Ádd to Wish List

[1394] link Ádd to Compare

[1399] link Śkip to the end of the images gallery´    [1156] button Ṕrevious

[1158] generic Ímage

[2502] img Ímage

[1157] button Ńext

URL:    http://onestopmarket.com/epson-workforce-wf-3620-wifi-direct-all-in-one-color-inkjet-printer-copier-scanner-amazon-dash-replenishment-ready.html

**Figure of Observation (only for visualization, not used in input prompt)**:

**Output**:

Thought: Let's think step-by-step. The current observation indicates that I am on the product page for "Epson WorkForce WF-3620 WiFi Direct All-in-One Color Inkjet Printer, Copier, Scanner, Amazon Dash Replenishment Ready." The page contains a link labeled "12 Reviews" with ID [1449]. To navigate to the reviews section, I need to click on this link.

Next Action: According to the thoughts, the next action I will perform is "'click [1449]'"

Target: Navigate to the reviews section by clicking on the "12 Reviews" link.

Summary: The current observation shows that I am on the product page for "Epson WorkForce WF-3620 WiFi Direct All-in-One Color Inkjet Printer, Copier, Scanner, Amazon Dash Replenishment Ready," and there is a link labeled "12 Reviews." I will click on this link to navigate to the reviews section.

### A.2.3    EXAMPLE OF REFLECTION PHASE

Example of Reflection Phase

**System Prompt**:

You are an autonomous intelligent agent tasked with navigating a web browser. You encountered an error while performing a task between web pages. You need to reflect on the mistakes made in your previous execution process and select a checkpoint to reload to retry. Remember that in the reflection, if there could be multiple errors or points that are incon-

sistent with your expectations, you should choose the earliest one to reload from. You will be provided with the following information: Task objective, execution process, verification target, and verification result. You should answer in the following format:

Reflection: (This is the reflection on why the previous execution process meets the mistake, what is your alternative way to solve the task and why you think your new method can solve the problem, which checkpoint you choose to restart from and why.)

Reload checkpoint: (The checkpoint you choose to reload. It should be an integer indicates the step. If you feel that you have actually achieved the task objectives, but the verification method is unreasonable, you should set the value to 0.)

---

**Example User**:
Task Objective:
4. Find the most recent out of delivery order from the filtered list.
Execution Process:
Step: 1
The summary of this step is:
The current observation shows a table listing various orders with their statuses. There are no orders with the status "Out of Delivery" in the current list. I will now finish this subtask by indicating that there are no such orders.
You have taken action 'finish_subtask [No out of delivery orders found]' in this step.
Verification:
Target: check_in_webpage("You have found the most recent out of delivery order")
Result: [T́o verify if the content satisfies the subgoal of finding the most recent out-of-delivery order, I need to check the status of the orders listed and identify the most recent one that is not marked as "Complete" or "Canceled."]́

**Example Assistant**:
Reflection: The execution process involved checking the current list of orders and concluding that there are no orders with the status "Out of Delivery." However, the verification result indicates that the content needs to be checked to ensure that the most recent order with a status other than "Complete" or "Canceled" is identified.
The mistake lies in not thoroughly verifying the status of each order to ensure that none of them are "Out of Delivery." Additionally, the verification should include checking for the most recent order that is not marked as "Complete" or "Canceled."
To solve the task, I should include a step to verify the status of each order on the webpage, ensuring that none of them are "Out of Delivery." If there are no such orders, I should then identify the most recent order that is not marked as "Complete" or "Canceled."
Since the mistake occurred after checking the current list of orders, I should reload from the point just before finishing the subtask to include the verification step.
Reload checkpoint: 0

---

**Example User**:
Task Objective:
4. Switch to the tab with the product that has the lowest per unit price.
Execution Process:
Step: 1
The summary of this step is:
I need to check the price and quantity of the Ciclon Energy Drink in the current tab (Tab 0) to determine the per unit price.
You have taken action 'finish_subtask [Ciclon Energy Drink]' in this step.
Verification:
Target: check_in_url("URL of the tab with the lowest per unit price") —OR— check_in_webpage("You have switched to the tab with the lowest per unit price")
Result: [śubgoal url url of the tab with the lowest per unit price is not in current url http://127.0.0.1:7770/ciclon-energy-drink-regular-24-cans-8-3oz.html; T́o determine if the

content satisfies the subgoal of switching to the tab with the lowest per unit price, I need to compare the prices of the items in both tabs. The first tab contains "Ciclon Energy Drink - Regular 24 cans (8.3oz)" priced at $36.99. The second tab contains "V8 +Energy, Healthy Energy Drink, Steady Energy from Black and Green Tea, Pomegranate Blueberry, 8 Ounce Can, Pack of 24". I need to find the price of the V8 +Energy drink to make the comparison.́

**Example Assistant**:
Reflection: The execution process involved checking the price and quantity of the Ciclon Energy Drink in the current tab and concluding that it has the lowest per unit price. However, the verification result indicates that the content needs to be checked to ensure that the tab with the lowest per unit price has been correctly identified. The mistake lies in not comparing the per unit prices of the items in both tabs before making a decision.
To solve the task, I should include a step to check the price and quantity of the V8 +Energy drink in the second tab to compare the per unit prices of both products. This comparison will ensure that I switch to the tab with the actual lowest per unit price.
Since the mistake occurred after checking the price of the Ciclon Energy Drink but before comparing it with the V8 +Energy drink, I should reload from the point just before finishing the subtask to include the comparison step.
Reload checkpoint: 0'

- - - - - - - - - - - - - - - - - - - - - - - - - - - - - - - - - - - - - - - - - - - - -

**Example User**:
Task Objective:
3. **Initiate the search**
- Click the search button or press Enter to start the search.
Execution Process:
Step: 1
The summary of this step is:
The current observation shows the search results page for "EYZUTAK" on the One Stop Market website. The search has already been initiated, and the results are displayed. Therefore, the subtask to initiate the search for "EYZUTAK" has been completed.
You have taken action 'finish_subtask []' in this step.
Verification:
Target: : check_in_url(śearch́ —OR— check_in_webpage(Śearch results for "usb wifi" displayed́
Result:          ["subgoal     url     śearch́ís     not     in     current     url http://127.0.0.1:7770/catalogsearch/result/?q=eyzutak", T́he content provided shows search results for the term "EYZUTAK" and not for "usb wifi". The subgoal target is to verify if the search results for "usb wifi" are displayed.́

**Example Assistant**:
Reflection: The execution process correctly identified that the search results for "EYZU-TAK" were displayed, but the task objective was to initiate the search for "usb wifi." The verification result indicates that the search results for "usb wifi" were not displayed, and the URL did not contain the expected search term.
The mistake lies in assuming that the search for "EYZUTAK" was relevant to the task objective. To solve the task, I should ensure that the search term "usb wifi" is entered and the search is initiated correctly.
Since the mistake occurred at the very beginning of the task, I should reload from the initial step to correct the search term and initiate the search for "usb wifi."
Reload checkpoint: 1

- - - - - - - - - - - - - - - - - - - - - - - - - - - - - - - - - - - - - - - - - - - - -

**User**
Task Objective:
4. **Identify the best storage option for 11 game cards**

- Review the search results to find a storage option that can fit at least 11 Nintendo Switch game cards.
- **
Execution Process:
Step: 1
The summary of this step is:
The current observation shows the search results page for "Nintendo Switch game card storage" on the One Stop Market website. I will scroll down to view more details about the storage options.
You have taken action 'scroll [down]' in this step.
Step: 2
The summary of this step is:
The current observation shows the search results for "Nintendo Switch game card storage." The best storage option for 11 game cards is the "HEIYING
You have taken action 'finish_subtask [The best storage option for 11 game cards is the HEIYING Game Card Case for Nintendo Switch & Switch OLED Game Card or Micro SD Memory Cards, Portable Switch Game Memory Card Storage with 24 Game Card Slots and 24 Micro SD Card Slots. (Black) priced at $14.69]' in this step.
Verification:
Target: **: 'check_in_webpage("Storage option for 11 Nintendo Switch game cards")'
Result: [́The search results provide various storage options for Nintendo Switch game cards. I need to verify if any of these options specifically mention the capacity to store 11 game cards.́]

- - - - - - - - - - - - - - - - - - - - - - - - - - - - - - - - - - - - - - -

**Output**:
Reflection: The execution process involved scrolling down to view more details about the storage options and then selecting the "HEIYING Game Card Case" as the best option for 11 game cards. However, the verification result indicates that the content needs to be checked to ensure that the selected storage option specifically mentions the capacity to store 11 game cards.
The mistake lies in not explicitly verifying that the selected storage option can fit at least 11 game cards. The selected option has 24 game card slots, which meets the requirement, but this detail was not clearly verified in the process.
To solve the task, I should include a step to explicitly verify that the selected storage option can fit at least 11 game cards by checking the product details.
Since the mistake occurred after selecting the storage option but before verifying its capacity, I should reload from the point just before finishing the subtask to include the verification step.
Reload checkpoint: 1

### A.2.4 EXAMPLE OF REPLANNING PHASE

**Example of Replanning Phase**

**System Prompt**:
You are an autonomous intelligent agent tasked with navigating a web browser. To achieve the final objective, you have broken it down into a series of subtasks to be tackled one by one. However, when you encounter errors while executing a particular subtask, and retrying within this subtask does not resolve the issue, you decide to re-decompose and create new subtasks. You will be provided with the following information:
Final Objective: Intent: The ultimate goal;
Old Scheme: The previously generated set of subtasks;
Global history: The execution history of the subtasks that have been successfully completed before the error occurred; these subtasks are done and do not need to be changed or regenerated;
Current subtask: The subtask that encountered the error;

Local history: The execution history of the subtask that encountered the error;

Verification target: The indicator for verification of the error;

Verification result: The error message.

Please note:

1. The subtasks executed before the error do not need to be modified, as they are already completed. You only need to generate replacements for the new subtasks following the one that encountered the error;

2. When creating new subtasks, you also need to establish methods of verification and verification targets. Please follow the following format for specifics.

Your verification method should be one of the three:

1. check_in_url (part of url): You think you will check in the url. This often be used when you want to navigate to some website. Its parameter is the fragment you expect to be the part of the url.

2. check_in_webpage (objective): You think you will check the objective in the last web-page. This often be used when the last webpage has the information for your sub-task. Its parameter is the objective you want to achieve in the sub-task and can be check in the webpage.

3. check_in_history (objective): You think you should check in your action history. This often be used when your sub-task is to get some information. Its parameter is the objective you want to achieve in the sub-task and can be check in the history.

You can use '|OR|' to connect two objective if you want to use multiple methods to assess the result of subtasks. You should add objectives by starting with Óbjectiveáfter each sub-task. For example, in F́ind the latest post. Objective: check_in_webpage ("This is the latest post"), F́ind the latest post.ís your verification method. After it is your verification method ćheck_in_webpageánd your objective of your method T́his is the latest post.

3. Your response should contain two parts. One is your thoughts of why the old subtasks encountered error and why your new subtasks can solve the problem. The other is your generated new subtasks, and this part should start with Ńew Subtasks.

4. You need to generate objectives for every subtasks. It is forbidden to generate a subtask without an objective.

- - - - - - - - - - - - - - - - - - - - - - - - - - - - - - - - - - - - - - - - - - - - -

**Eexample User**:

Final Objective: Intent: Buy the highest rated product from the Beauty & Personal Care category within a budget under 20.

To achieve the final objective, I decompose it into these subtasks:

Old Scheme:

1. Navigate to the "Beauty & Personal Care" category.

2. Sort the products by rating in descending order.

3. Filter the products to show only those priced under $20.

4. Identify the highest-rated product within the filtered results.

5. Add the highest-rated product to the shopping cart.

Previous Subtasks and the Summary of Their Execution History:

Global History:

1. Navigate to the "Beauty & Personal Care" category.

Summary: The navigation to the "Beauty & Personal Care" category was successful, as confirmed by the URL indicating the correct category page.

Current Subtask:

2. Sort the products by rating in descending order.

Local History:

Previous Steps:

Step: 1

The summary of this step is:

The current observation shows a "Sort By" combobox that is currently focused and expanded. I will click on this combobox to view the sorting options.

You have taken action 'click [13132] where [13132] is combobox Śort Byf́ocused: True hasPopup: menu expanded: True' in this step.

Step: 2
The summary of this step is:
The current observation shows that the "Sort By" combobox is focused but not expanded. The next step is to expand the combobox to view the sorting options.
You have taken action 'click [13132] where [13132] is combobox Śort Byfocused: True hasPopup: menu expanded: False' in this step.
Step: 3
The summary of this step is:
The current observation shows that the "Sort By" combobox is focused and expanded, displaying the sorting options. The next step is to select the option to sort the products by rating in descending order.
You have taken action 'click [13133] where [13133] is link úe613 Set Descending Direction' in this step.
Step: 4
The summary of this step is:
The current observation shows that the "Sort By" combobox is focused but not expanded, and the sorting direction is set to descending. I need to ensure that the products are sorted by rating in descending order. Therefore, I will expand the "Sort By" combobox to view the sorting options.
You have taken action 'click [16146] where [16146] is combobox Śort ByhasPopup: menu expanded: False' in this step.
Step: 5
The summary of this step is:
The current observation shows that the "Sort By" combobox is focused and expanded, displaying the sorting options. The next step is to select the option to sort the products by rating in descending order.
You have taken action 'click [16147] where [16147] is link úe614 Set Ascending Direction' in this step.
Step: 6
The summary of this step is:
The current observation shows that the "Sort By" combobox is focused but not expanded, and the sorting direction is set to descending. I need to ensure that the products are sorted by rating in descending order. Therefore, I will expand the "Sort By" combobox to view the sorting options.
You have taken action 'click [19160] where [19160] is combobox Śort ByhasPopup: menu expanded: False' in this step.
Verification:
Target: : check_in_webpage("Products sorted by rating")
Result: ['To determine if the products are sorted by rating, I need to look for any indication that the sorting criteria is based on ratings. This could be a label or a dropdown selection indicating "Sort by Rating" or similar wording.]
You should not modify the already completed subtasks, namely
1. Navigate to the "Beauty & Personal Care" category.
You should reflect on the original mistakes (i.e., the local history mentioned above) and generate subsequent new subtasks and verification plans based on the completed results (i.e., the global history mentioned above). Note that your new sub-tasks index should be started from the 'Current Subtask', that is 1, and your new subtasks should not exceed 4.

**Example Assistant**:
The error in the previous execution was due to the repeated attempts to expand the "Sort By" combobox and select the sorting option without successfully verifying that the products were sorted by rating in descending order. The verification method requires checking for any indication that the sorting criteria are based on ratings.
To resolve this, the new subtasks must ensure that the products are sorted by rating in descending order and that this sorting is verified before proceeding to the next steps.

New Subtasks:
2. Sort the products by rating in descending order.
Objective: check_in_webpage("Products sorted by rating in descending order")
3. Filter the products to show only those priced under $20.
Objective: check_in_webpage("Products filtered by price under $20")
4. Identify the highest-rated product within the filtered results.
Objective: check_in_webpage("Highest-rated product identified within filtered results")
5. Add the highest-rated product to the shopping cart.
Objective: check_in_webpage("Highest-rated product added to the shopping cart")
These new subtasks should ensure that the products are sorted by rating in descending order, filtered by price, and that the highest-rated product is identified and added to the shopping cart. This should resolve the previous issues and achieve the final objective.

- - - - - - - - - - - - - - - - - - - - - - - - - - - - - - - - - - - - - - - - - - - -

**Example User**:
Final Objective: Intent: Buy the highest rated product from the Beauty & Personal Care category within a budget under 20.
To achieve the final objective, I decompose it into these subtasks:
Old Scheme:
1. Navigate to the "Beauty & Personal Care" category.
2. Sort the products by rating in descending order.
3. Filter the products to show only those priced under $20.
4. Identify the highest-rated product within the filtered results.
5. Add the highest-rated product to the shopping cart.
Previous Subtasks and the Summary of Their Execution History:
Global History:
1. Navigate to the "Beauty & Personal Care" category.
Summary: The navigation to the "Beauty & Personal Care" category was successful, as confirmed by the URL indicating the correct category page.
Current Subtask:
2. Sort the products by rating in descending order.
Local History:
Previous Steps:
Step: 1
The summary of this step is:
The current observation shows a page listing beauty and personal care products. To sort the products by rating in descending order, I need to interact with the "Sort By" combobox. I will start by clicking on the combobox to open it and view the sorting options.
You have taken action 'click [19160] where [19160] is combobox Śort Byfocused: True hasPopup: menu expanded: True' in this step.
Step: 2
The summary of this step is:
The current observation shows that the "Sort By" combobox is focused but not expanded. The next step is to click on the "Sort By" combobox to expand it and view the sorting options.
You have taken action 'click [19160] where [19160] is combobox Śort Byfocused: True hasPopup: menu expanded: False' in this step.
Step: 3
The summary of this step is:
The current observation shows that the "Sort By" combobox is now expanded, and the sorting options are visible. The next step is to select the option to sort by rating in descending order by clicking on the "Set Descending Direction" link.
You have taken action 'click [19161] where [19161] is link úe613 Set Descending Direction' in this step.
Step: 4
The summary of this step is:

The current observation shows that the "Sort By" combobox is not expanded, and the sorting options are not visible. The next step is to click on the "Sort By" combobox to expand it and view the sorting options.

You have taken action 'click [22174] where [22174] is combobox Śort Byh́asPopup: menu expanded: False' in this step.

Step: 5

The summary of this step is:

The current observation shows that the "Sort By" combobox is expanded, and the sorting options are visible. The next step is to select the option to sort by rating in descending order by clicking on the appropriate option.

You have taken action 'click [22175] where [22175] is link úe614 Set Ascending Direction´ in this step.

Step: 6

The summary of this step is:

The current observation shows that the "Sort By" combobox is not expanded, and the sorting options are not visible. The next step is to click on the "Sort By" combobox to expand it and view the sorting options.

You have taken action 'click [25188] where [25188] is combobox Śort Byh́asPopup: menu expanded: False' in this step.

Verification:

Target: : check_in_webpage("Products sorted by rating in descending order")

Result: [T́o verify if the products are sorted by rating in descending order, I need to check for any indication that the sorting criteria is based on rating and that it is in descending order. The content should show a sorting option or label indicating "rating" and "descending".]́

You should not modify the already completed subtasks, namely

1. Navigate to the "Beauty & Personal Care" category.

You should reflect on the original mistakes (i.e., the local history mentioned above) and generate subsequent new subtasks and verification plans based on the completed results (i.e., the global history mentioned above). Note that your new sub-tasks index should be started from the 'Current Subtask', that is 1, and your new subtasks should not exceed 4.

**Example Assistant**:

The error in the previous execution was due to the repeated attempts to interact with the "Sort By" combobox without successfully selecting the option to sort by rating in descending order. The verification method requires checking for any indication that the sorting criteria is based on rating and that it is in descending order.

To resolve this, the new subtasks must ensure that the products are sorted by rating in descending order and that this sorting is verified.

New Subtasks:

2. Sort the products by rating in descending order.

Objective: check_in_webpage("Products sorted by rating in descending order")

3. Filter the products to show only those priced under $20.

Objective: check_in_webpage("Products filtered to show only those priced under $20")

4. Identify the highest-rated product within the filtered results.

Objective: check_in_webpage("Highest-rated product identified within the filtered results")

5. Add the highest-rated product to the shopping cart.

Objective: check_in_webpage("Highest-rated product added to the shopping cart")

These new subtasks should ensure that the products are sorted correctly, filtered by price, and that the highest-rated product is identified and added to the shopping cart. This should resolve the previous issues and achieve the final objective.

- - - - - - - - - - - - - - - - - - - - - - - - - - - - - - - - - - - - - - - - - -

**Example User**:

Final Objective: Intent: Tell me the status of my latest order and when will it arrive

To achieve the final objective, I decompose it into these subtasks:

Old Scheme:

1. Navigate to the "My Account" page.
2. Locate the "Orders" section within the "My Account" page.
3. Identify the list of orders and find the latest order.
4. Retrieve the status of the latest order.
5. Verify the presence or absence of the estimated arrival date for the latest order.
Previous Subtasks and the Summary of Their Execution History:
Global History:
1. Navigate to the "My Account" page.
Summary: The "My Account" link was clicked, successfully navigating to the "My Account" page, as confirmed by the URL and the presence of account-related information on the page.
2. Locate the "Orders" section within the "My Account" page.
Summary: The "My Orders" link was clicked, successfully navigating to the "Orders" section within the "My Account" page, as confirmed by the detailed order table present on the page.
3. Identify the list of orders and find the latest order.
Summary: The latest order, 000000170 dated 5/17/23, was identified, but the verification step to confirm it as the most recent order on the webpage was missed. The process needs to be reloaded to include this verification step.
4. Retrieve the status of the latest order.
Summary: The status of the latest order (000000170) was identified as "Canceled," but the verification step to check the webpage content for this status was missed. The process needs to include a step to verify the presence of the latest order's status on the webpage.
Current Subtask:
5. Verify the presence or absence of the estimated arrival date for the latest order.
Local History:
Previous Steps:
Step: 1
The summary of this step is:
The current observation shows the details of order 000000170. To verify the presence or absence of the estimated arrival date, I need to scroll down the page to check for any mention of it. The expected outcome of this action is to see if there is any information regarding the estimated arrival date for this order.
You have taken action 'scroll [down]' in this step.
Step: 2
The summary of this step is:
The current observation shows the details of order 000000170, including items ordered, order information, shipping address, shipping method, billing address, and payment method. There is no mention of the estimated arrival date in the visible details. Therefore, I will finish the subtask to confirm that the estimated arrival date is absent for the latest order.
You have taken action 'finish_subtask []' in this step.
You have encountered error in previous trial, this is your reflection:
Reflection: The execution process involved checking the details of order 000000170 to verify the presence or absence of the estimated arrival date. The steps included scrolling down the page and observing the order details, ultimately concluding that there was no mention of the estimated arrival date. The verification result indicates that the content does not include any information about the estimated arrival date and suggests checking for any mention of an estimated arrival date or an indication of order cancellation.
The mistake lies in not explicitly verifying the webpage content for any mention of an estimated arrival date or an indication of order cancellation. To solve the task, I should include a step to verify the webpage content for any mention of an estimated arrival date or an indication that the order has been canceled.
Since the mistake occurred after observing the order details and not verifying the absence of the estimated arrival date or order cancellation, I should reload from the point just before finishing the subtask to include the verification step.
And you choose to restart from this state.
Step: 3

The summary of this step is:

The current observation shows the details of order 000000170, including items ordered, order information, shipping address, shipping method, billing address, and payment method. There is no mention of the estimated arrival date in the visible details. The next step involves scrolling down the page to check for any mention of the estimated arrival date. The expected outcome of this action is to see if there is any information regarding the estimated arrival date for this order.

You have taken action 'scroll [down]' in this step.

Step: 4

The summary of this step is:

The current observation shows the details of order 000000170, including items ordered, order information, shipping address, shipping method, billing address, and payment method. There is no mention of the estimated arrival date in the visible details. The next step involves confirming the absence of the estimated arrival date for the latest order. The expected outcome of this action is to confirm that the estimated arrival date is absent for the latest order.

You have taken action 'finish_subtask [The estimated arrival date is absent for the latest order.]' in this step.

Verification:

Target: : check_in_webpage("Estimated arrival date for order 000000170") |OR|check_in_webpage("No estimated arrival date due to order cancellation")

Result: [T́he content provided does not include any information about the estimated arrival date for order 000000170. It contains details about the order information, shipping address, shipping method, billing address, payment method, and some additional elements like recently ordered items and newsletter sign-up, but no mention of the estimated arrival date.;

T́he content provided does not mention anything about an estimated arrival date or order cancellation. It primarily includes order information, shipping and billing addresses, shipping method, payment method, and some additional elements like a wish list and newsletter sign-up.]

You should not modify the already completed subtasks, namely

1. Navigate to the "My Account" page.
2. Locate the "Orders" section within the "My Account" page.
3. Identify the list of orders and find the latest order.
4. Retrieve the status of the latest order.

You should reflect on the original mistakes (i.e., the local history mentioned above) and generate subsequent new subtasks and verification plans based on the completed results (i.e., the global history mentioned above). Note that your new sub-tasks index should be started from the 'Current Subtask', that is 4, and your new subtasks should not exceed 1.

**Example Assistant**:

The error in the previous execution was due to the lack of explicit verification for the presence or absence of the estimated arrival date or an indication of order cancellation. The verification method requires checking the webpage content for any mention of the estimated arrival date or an indication that the order has been canceled. To resolve this, the new subtask must ensure that the webpage content is thoroughly checked and verified for this information.

New Subtasks:

5. Verify the presence or absence of the estimated arrival date for the latest order and check for any indication of order cancellation.

Objective: check_in_webpage("Estimated arrival date for order 000000170") |OR|check_in_webpage("No estimated arrival date due to order cancellation")

This new subtask should ensure that the webpage content is thoroughly checked and verified for the presence or absence of the estimated arrival date or any indication of order cancellation. This will ensure that the task objective is fully met and verified.

**User**:

Final Objective: Intent: I have a lot of Nintendo Switch game cards now, help me find the best storage option to fit all 23 cards

To achieve the final objective, I decompose it into these subtasks:

Old Scheme:

1. **Navigate to the "Electronics" section**
- Locate and click on the "Electronics" menu item to explore related products.
2. **Locate the "Video Games" subcategory**
- Within the "Electronics" section, find and click on the "Video Games" subcategory to narrow down the search.
3. **Search for "Nintendo Switch game card storage"**
- Use the search bar within the "Video Games" section to search for "Nintendo Switch game card storage".
4. **Identify the best storage option for 23 game cards**
- Review the search results and identify a storage option that can fit at least 23 Nintendo Switch game cards.
5. **Verify the product details**
- Click on the identified product to verify its details, ensuring it meets the requirement of fitting 23 game cards.

Previous Subtasks and the Summary of Their Execution History:

Global History:

1. **Navigate to the "Electronics" section**
- Locate and click on the "Electronics" menu item to explore related products.
Summary: The "Electronics" menu item was successfully clicked, leading to the "Electronics" section, as confirmed by the URL containing "electronics".
2. **Locate the "Video Games" subcategory**
- Within the "Electronics" section, find and click on the "Video Games" subcategory to narrow down the search.
Summary: The "Video Games" subcategory was successfully located and clicked within the "Electronics" section, leading to the correct "Video Games" page as confirmed by the URL.
3. **Search for "Nintendo Switch game card storage"**
- Use the search bar within the "Video Games" section to search for "Nintendo Switch game card storage".
Summary: The search query "Nintendo Switch game card storage" was successfully entered and the search button was clicked, resulting in the display of relevant search results. The search bar was verified to contain the correct query text.
4. **Identify the best storage option for 23 game cards**
- Review the search results and identify a storage option that can fit at least 23 Nintendo Switch game cards.
Summary:  Detailed Summary of Steps and Verification Result
**Step 1:**
- **Summary:** The current observation shows the search results page for "Nintendo Switch game card storage" on the One Stop Market website.  The next step is to scroll down to view more details about the search results, with the expected outcome of seeing more details about the storage options, including their capacities.
- **Action Taken:** Scrolled down to view more details.
**Reflection:**
- **Summary:** The execution process involved identifying the "HEIYING Game Card Case" as the best storage option for 23 game cards based on the search results. However, the verification result indicates that the content needs to be checked to ensure that the identified product meets the specific requirement of fitting at least 23 Nintendo Switch game cards. The mistake was not thoroughly verifying the details of the identified product and not checking other potential options.
- **Action Taken:** Decided to restart from the point just before finishing the subtask to include the verification step.
**Step 3:**

- **Summary:** The current observation shows the search results for "Nintendo Switch game card storage" on the One Stop Market website. The next step is to click on the "HEIYING Game Card Case" to navigate to its product page and verify its details.
- **Action Taken:** Clicked on the link for "HEIYING Game Card Case."
**Step 4:**
- **Summary:** The current observation confirms that the "HEIYING Game Card Case" has 24 game card slots, making it a suitable option for storing at least 23 Nintendo Switch game cards. Therefore, the subtask is finished.
- **Action Taken:** Finished the subtask.
**Verification Result:**
- **Target:** The webpage should describe a storage option for 23 Nintendo Switch game cards.
- **Result:** The content describes a game card
Current Subtask:
5. **Verify the product details**
- Click on the identified product to verify its details, ensuring it meets the requirement of fitting 23 game cards.
- **Local History:
Previous Steps:
Step: 1
The summary of this step is:
The current observation shows the product page for the "HEIYING Game Card Case for Nintendo Switch & Switch OLED Game Card or Micro SD Memory Cards, Portable Switch Game Memory Card Storage with 24 Game Card Slots and 24 Micro SD Card Slots." I will click on the product to verify its details and ensure it meets the requirement of fitting 23 game cards.
You have taken action 'click [16505] where [16505] is StaticText ́HEIYING Game Card Case for Nintendo Switch & Switch OLED Game Card or Micro SD Memory Cards,Portable Switch Game Memory Card Storage with 24 Game Card Slots and 24 Micro SD Card Slots. (Black) ́ in this step.
You have encountered error in previous trial, this is your reflection:
Reflection: The execution process involved clicking on the product and confirming that it meets the requirement of fitting 23 game cards. However, the verification target requires explicitly checking the product details to confirm the storage capacity for 23 Nintendo Switch game cards. The mistake lies in not explicitly verifying the product description for this specific detail.
To resolve this, I should reload from the point before finishing the subtask and explicitly check the product description on the webpage to confirm that it mentions storage for 23 Nintendo Switch game cards.
And you choose to restart from this state.
Step: 3
The summary of this step is:
The current observation confirms that the "HEIYING Game Card Case for Nintendo Switch & Switch OLED Game Card or Micro SD Memory Cards, Portable Switch Game Memory Card Storage with 24 Game Card Slots and 24 Micro SD Card Slots" has 24 game card slots, which meets the requirement of fitting 23 game cards. Therefore, I will finish the subtask.
You have taken action 'finish_subtask [The product has 24 game card slots, which meets the requirement of fitting 23 game cards] ́ in this step.
Verification:
Target: **: 'check_in_webpage("Product details confirm storage for 23 Nintendo Switch game cards")
Result: [ ́The subgoal target is to confirm whether the product details mention storage for 23 Nintendo Switch game cards. I need to verify if the product description includes this specific storage capacity. ́]
You should not modify the already completed subtasks, namely
1. **Navigate to the "Electronics" section**
- Locate and click on the "Electronics" menu item to explore related products.

2. **Locate the "Video Games" subcategory**
- Within the "Electronics" section, find and click on the "Video Games" subcategory to narrow down the search.
3. **Search for "Nintendo Switch game card storage"**
- Use the search bar within the "Video Games" section to search for "Nintendo Switch game card storage".
4. **Identify the best storage option for 23 game cards**
- Review the search results and identify a storage option that can fit at least 23 Nintendo Switch game cards.
You should reflect on the original mistakes (i.e., the local history mentioned above) and generate subsequent new subtasks and verification plans based on the completed results (i.e., the global history mentioned above). Note that your new sub-tasks index should be started from the 'Current Subtask', that is 4, and your new subtasks should not exceed 1.

- - - - - - - - - - - - - - - - - - - - - - - - - - - - - - - - - - - - - - - - - - - - - - - -

**Output**:
The error in the previous execution was due to not explicitly verifying the product description for the specific detail of fitting 23 Nintendo Switch game cards. The verification method requires checking the product details on the webpage to confirm this specific storage capacity.
To resolve this, the new subtask must ensure that the product description is thoroughly checked and verified for the storage capacity of 23 Nintendo Switch game cards.
New Subtasks: 5. Verify the product details to ensure it meets the requirement of fitting 23 game cards. Objective: check_in_webpage("Product details confirm storage for 23 Nintendo Switch game cards")
This new subtask should ensure that the product description is thoroughly checked and verified for the specific storage capacity of 23 Nintendo Switch game cards. This will ensure that the task objective is fully met and verified.

## B MORE EXPERIMENTS

### B.1 COMPUTATIONAL COST

The computational cost is an important criterion for evaluating the efficiency of an LLM workflow. In Table 6, we report the computational cost incurred by each module in VeSX and visualize the proportion of each module's contribution in Figure 5, Figure 6, Figure 7. Although we implemented some design changes in prompts for planning and execution, the count of tokens used is small. The additional cost resulting from process improvements can largely be attributed to verification, reflection, and replanning.

Considering that the computational cost is determined by the number of tokens, the verification, reflection and replanning modules account for approximately 10.7% of the input tokens (Figure 5). For output tokens, the three modules account for about 16.1% of the tokens (Figure 6). Moreover, the count of LLM calls in the three modules account for about 22.3%. This indicates that the three main modules in VeSX workflow, verification, reflection, and replanning, do not introduce significant more computational cost.

We also tested the token usage for examples retrieved from the exemplar bank. The examples for execution accounted for 33.9% of the total input tokens, while the examples for planning also accounted for 2.4% of the input tokens. This is a considerable proportion, but given the widespread adoption of ICL, most methodsSodhi et al. (2024)Koh et al. (2024)Drouin et al. (2024) will incur this overhead. And the computional cost for these examples can be further saved due to the fact found in Appendix B.2 that fewer examples can still maintain a high level of accuracy.

Due to the significant time and cost involved in reproducing other approaches, we did not fully reproduce all of them. We compared our method with the Sodhi et al. (2024) approach, which uses human-labeled subtasks to guide the LLM in breaking down and solving tasks. Their average

| Module | planning | execution | verification | reflection | replanning | summary&answer |
|---|---|---|---|---|---|---|
| Input Token Count | 7743 | 69045 | 5229 | 1034 | 4076 | 10311 |
| Output Token Count | 188 | 2095 | 247 | 98 | 184 | 475 |
| Module Call Count | 1 | 13.1 | 4.1 | 0.60 | 0.68 | 4.6 |

Table 6: Computational cost statistics of each module in VeSX workflow.

| Module | Instruction | Exemplar |
|---|---|---|
| Planning | 4147 | 3596 |
| Execution | 18583 | 50462 |

Table 7: Statistics of token counts for instructions and exemplars in planning and execution module inputs.

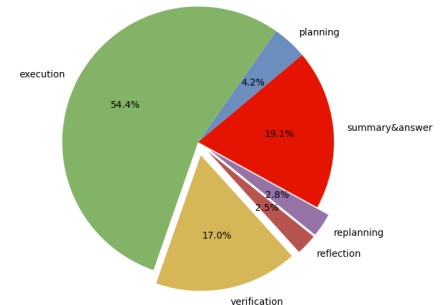

Figure 5: Proportions of input tokens for each module.

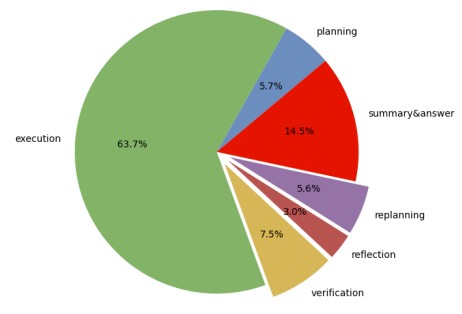

Figure 6: Proportions of output tokens for each module.

Figure 7: Proportions of module calls for each module.

action step is 9.1, while our average action step is 13.1. From this perspective, VeSX requires approximately 43.9% more actions compared to human-guided workflows to achieve competitive results.

### B.2 SCALING OF EXEMPLARS

We first analyzed the size of the exemplar bank in various scenarios, as shown in the Table **??**.

We also conducted experiments to test the scalability of the exemplars, primarily using shopping as the experimental scenario. In Table 9, we tested the impact of using different numbers of exemplars during ICL on the final end-to-end results. The SR improves when the count of ICL exemplars increases, underlying the potential of the scaling of exemplars during the inference process. In Table 10, we examined how the SR changes under different settings of the count of planning and execution exemplars. We set four settings: (1) sampling 70% of the whole exemplar bank, (2) using the whole execution exemplar bank but not the planning exemplar, (3) using the whole planning exemplar bank but not the execution exemplar, and (4) using the whole exemplar bank. The results shows that using only 70% of the whole bank causes a obvious decrease of SR comparing to even removing the whole planning exemplars or execution exemplars. It indicates the more diversity of the execution or the planning exemplars will bring more increase and the potential of the scaling of the exemplar bank.

| Scenario | Shop. | CMS | Red. | Git. | Map |
|----------|-------|-----|------|------|-----|
| Planning | 35 | 24 | 30 | 29 | 20 |
| Execution | 171 | 123 | 211 | 193 | 87 |

Table 8: Statistics of exemplar banks for the five scenarios.

| Num of ICL exemplars | SR |
|----------------------|-----|
| 3 | 0.412 |
| 5 | 0.458 |

Table 9: Success rates under different settings of the count of in-context-learning exemplars. SR represents the success rate.

| #P | #E | SR |
|-----|-----|-------|
| 26 | 120 | 0.334 |
| 0 | 171 | 0.340 |
| 35 | 0 | 0.373 |
| 35 | 171 | 0.412 |

Table 10: Success rates under different settings of the count of planning and execution exemplars. SR represents the success rate. #P refers to the number of planning exemplars and #E refers to the number of planning exemplars.

| Condition | SR | Proportion |
|-----------|-----|-----------|
| Reflection | 0.386 | 43.3% |
| Replanning | 0.229 | 53.6% |
| Reflection (w/o replanning) | 0.661 | 12.5% |
| Verification pass | 0.384 | 33.8% |
| Verification fail | 0.318 | 66.2% |

Table 11: Success rates and proportion of different modules works. SR represents the success rate. The "verification pass" means that all verifications in this process pass, while the "verification fail" means at least one verification does not pass.

| Condition | SR | Proportion |
|-----------|-----|-----------|
| Verification pass | 0.43 | 38% |
| Verification fail | 0.40 | 62% |

Table 12: Success rates and proportion of different modules works in scenario Shop. SR represents the success rate. The "verification pass" means that all verifications in this process pass, while the "verification fail" means at least one verification does not pass.

| | Label=Pos | Label=Neg |
|----------|-----------|-----------|
| Pred=Pos | 0.16 | 0.22 |
| Pred=Neg | 0.13 | 0.49 |

Table 13: The confusion matrix of verification in scene Shop. Pos represents positive samples and Neg represents negative samples.

### B.3 PERFORMANCE OF VERIFICATION, REFLECTION AND REPLANNING

We analyzed the performance of the success rate after verification, reflection, and replanning occurred, and we calculated the proportions of each situation to test the capabilities and performance of each module. Table 11 presents the performance across all five scenarios. Here, the "verification pass" is defined as the instance that all verifications in this process pass, while the "verification fail" is defined as the instance where at least one verification does not pass. It can be seen that the success rate of "verification pass" reaches 0.384, while that of "verification fail" achieve 0.318 after undergoing reflection and replanning. When reflection is triggered without the need for replanning, the success rate can reach 0.661, and this portion of examples accounts for 12.5%. This is because these examples are relatively simple, making errors easier to correct (C.1 provides such an example). The cases that trigger replanning account for 53.6%, with a corresponding success rate of only 0.229, indicating that these examples are difficult and highlighting the challenge of having the LLM replan correctly globally.

Additionally, we analyzed the success rate and its proportion in the shopping scenario when verification passes and fails, as shown in Table 12. We previously conducted an ablation study on the cases without verification. Combining these two parts of the experiment, we can roughly estimate the confusion matrix of verification, as shown in Table 13. The false positive rate can be estimated as 0.57. This result demonstrates that the verification may not be reliable when classified as a positive sample. The false negative rate needs to be estimated using the success rate when not verifying and the success rate when verification fails, and can be estimated as 0.20. This result demonstrates that

verification rarely classifies positive samples as negative, which does not incur much unnecessary self-correction overhead. The true negative rate can be estimated as 0.79. This result demonstrates that the verification has high rate to capture the error in the process.

## C  CASE STUDIES

### C.1  REFLECTION

---

**One Complete Example**

**Query Intent**:
I have a lot of Nintendo Switch game cards now, help me find the best storage option to fit all 11 cards.
**Ground Truth**
URL Match: 'reference_url': 'http://127.0.0.1:7770/heiying-game-card-case-for-nintendo-switch-switch-oled-game-card-or-micro-sd-memory-cards-portable-switch-game-memory-card-storage-with-24-game-card-slots-and-24-micro-sd-card-slots-black.html'

- - - - - - - - - - - - - - - - - - - - - - - - - - - - - - - - - - - - - - - - - - - -

**Planning**
1. **Navigate to the "Electronics" section** - Locate and click on the "Electronics" menu item to access the electronics category.
- **Objective**: 'check_in_url("electronics")'
2. **Locate the "Video Games" subcategory** - Within the "Electronics" section, find and click on the "Video Games" subcategory to narrow down the search to video game-related products.
- **Objective**: 'check_in_url("video-games")'
3. **Search for "Nintendo Switch game card storage"** - Use the search bar within the "Video Games" section to search for "Nintendo Switch game card storage".
- **Objective**: 'check_in_webpage("Search bar is populated with 'Nintendo Switch game card storage'")'
4. **Identify the best storage option for 11 game cards** - Review the search results to find a storage option that can fit at least 11 Nintendo Switch game cards.
- **Objective**: 'check_in_webpage("Storage option for 11 Nintendo Switch game cards")'
5. **Navigate to the product page of the identified storage option** - Click on the identified storage option to navigate to its product page for more details.
-        **Objective**:                'check_in_url("nintendo-switch-game-card-storage") |OR|check_in_webpage("You are on the product page of the identified storage option")'

- - - - - - - - - - - - - - - - - - - - - - - - - - - - - - - - - - - - - - - - - - - -

**Subtask 1**
1. **Navigate to the "Electronics" section** - Locate and click on the "Electronics" menu item to access the electronics category.

**Figure of Observation (only for visualization, not used in input prompt)**:

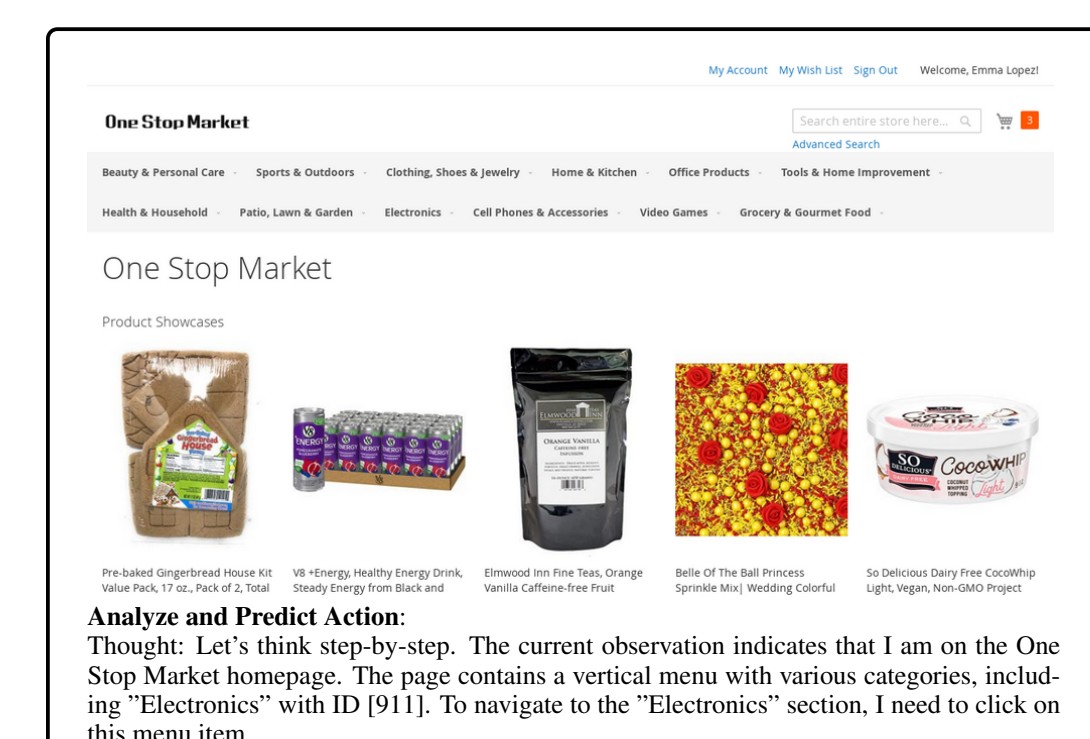

**Analyze and Predict Action**:

Thought: Let's think step-by-step. The current observation indicates that I am on the One Stop Market homepage. The page contains a vertical menu with various categories, including "Electronics" with ID [911]. To navigate to the "Electronics" section, I need to click on this menu item.

Next Action: According to the thoughts, the next action I will perform is "'click [911]'"

Target: Navigate to the "Electronics" section.

Summary: The current observation shows that I am on the One Stop Market homepage, and there is an "Electronics" menu item. I will click on this menu item to navigate to the "Electronics" section.

**Action: click [911] where [911] is menuitem 'ue622 Electronics' hasPopup: menu**

**Figure of Observation (only for visualization, not used in input prompt)**:

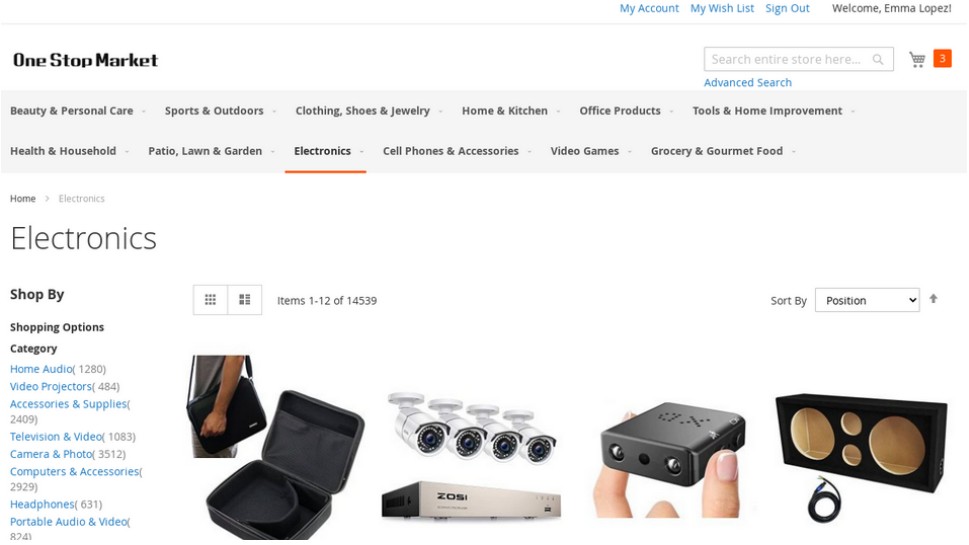

**Analyze and Predict Action**:

Thought: Let's think step-by-step. The current observation shows that I am on the "Electronics" page, which lists various items in this category. This means that the "Electronics" menu item has been successfully clicked and I have navigated to the correct page. Therefore, I have completed the subtask.

Next Action: According to the thoughts, the next action I will perform is "'finish_subtask"'
Target: Confirm that the subtask of locating and clicking on the "Electronics" menu item is completed.
Summary: The current observation shows that I am on the "Electronics" page, which lists various items in this category. This confirms that the "Electronics" menu item has been successfully clicked and I have navigated to the correct page. Therefore, I will finish the subtask.
**Action: finish_subtask []**

**Verification**:
**check_in_url("electronics")**
1.0 "electronics" is in the URL: http://127.0.0.1:7770/electronics.html
**Verification Pass**

**Subtask 2**
2. **Locate the "Video Games" subcategory** - Within the "Electronics" section, find and click on the "Video Games" subcategory to narrow down the search to video game-related products.

**Figure of Observation (only for visualization, not used in input prompt)**:

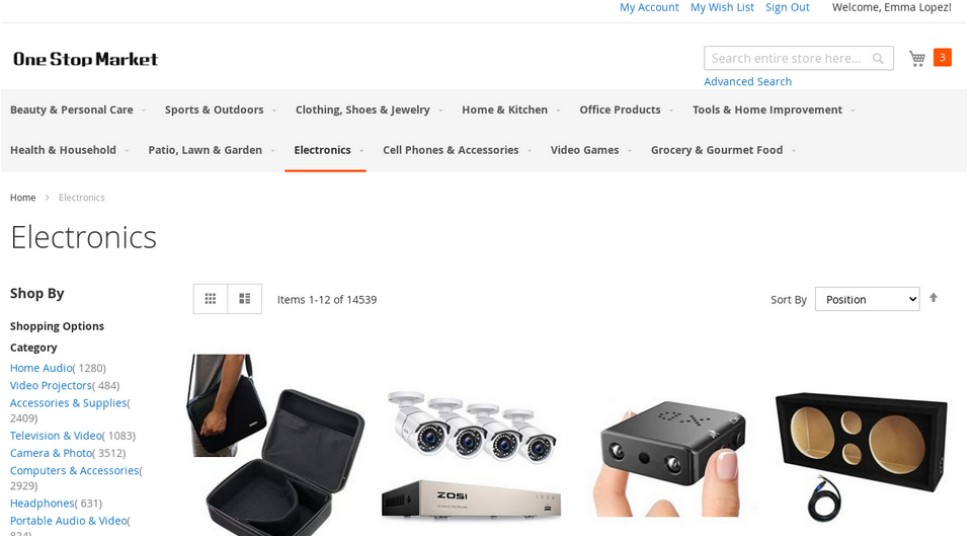

**Analyze and Predict Action**:
Thought: Let's think step-by-step. The current observation shows that I am on the "Electronics" page, and there is a menu item labeled "Video Games" with ID [3789]. Therefore, I will click on this menu item to navigate to the "Video Games" subcategory page.
Next Action: According to the thoughts, the next action I will perform is "'click [3789]"'
Target: Navigate to the "Video Games" subcategory page.
Summary: The current observation shows that I am on the "Electronics" page, and there is a menu item labeled "Video Games" with ID [3789]. Therefore, I will click on this menu item to navigate to the "Video Games" subcategory page.
**Action: click [3789] where [3789] is [3789] menuitem 'ue622 Video Games' hasPopup: menu**

**Figure of Observation (only for visualization, not used in input prompt)**:

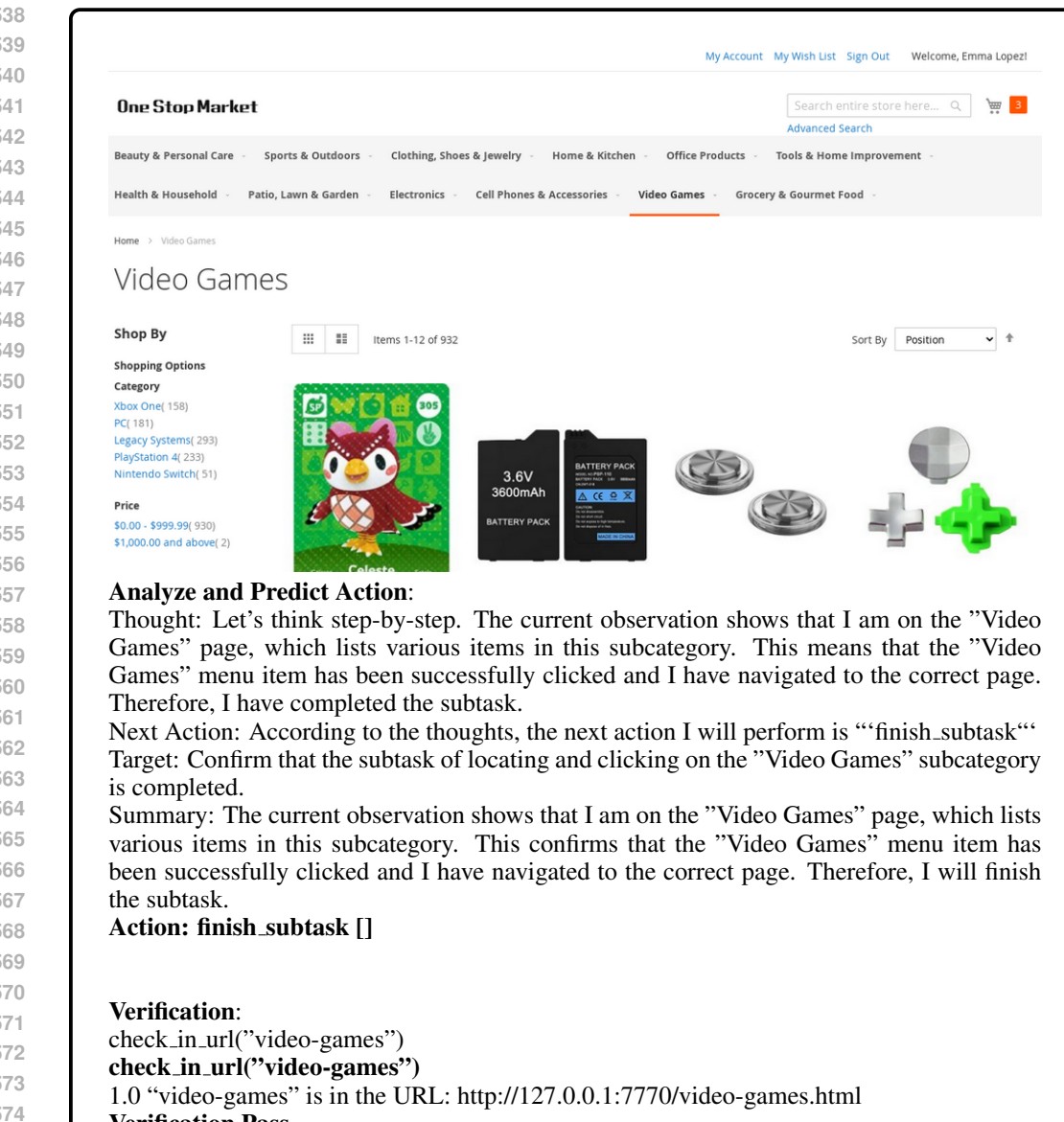

**Analyze and Predict Action**:

Thought: Let's think step-by-step. The current observation shows that I am on the "Video Games" page, which lists various items in this subcategory. This means that the "Video Games" menu item has been successfully clicked and I have navigated to the correct page. Therefore, I have completed the subtask.

Next Action: According to the thoughts, the next action I will perform is "'finish_subtask'"

Target: Confirm that the subtask of locating and clicking on the "Video Games" subcategory is completed.

Summary: The current observation shows that I am on the "Video Games" page, which lists various items in this subcategory. This confirms that the "Video Games" menu item has been successfully clicked and I have navigated to the correct page. Therefore, I will finish the subtask.

**Action: finish_subtask []**

**Verification**:

check_in_url("video-games")

**check_in_url("video-games")**

1.0 "video-games" is in the URL: http://127.0.0.1:7770/video-games.html

**Verification Pass**

- - - - - - - - - - - - - - - - - - - - - - - - - - - - - - - - - - - - - - - - - - -

**Subtask 3**

3. **Search for "Nintendo Switch game card storage"** - Use the search bar within the "Video Games" section to search for "Nintendo Switch game card storage".

**Figure of Observation (only for visualization, not used in input prompt)**:

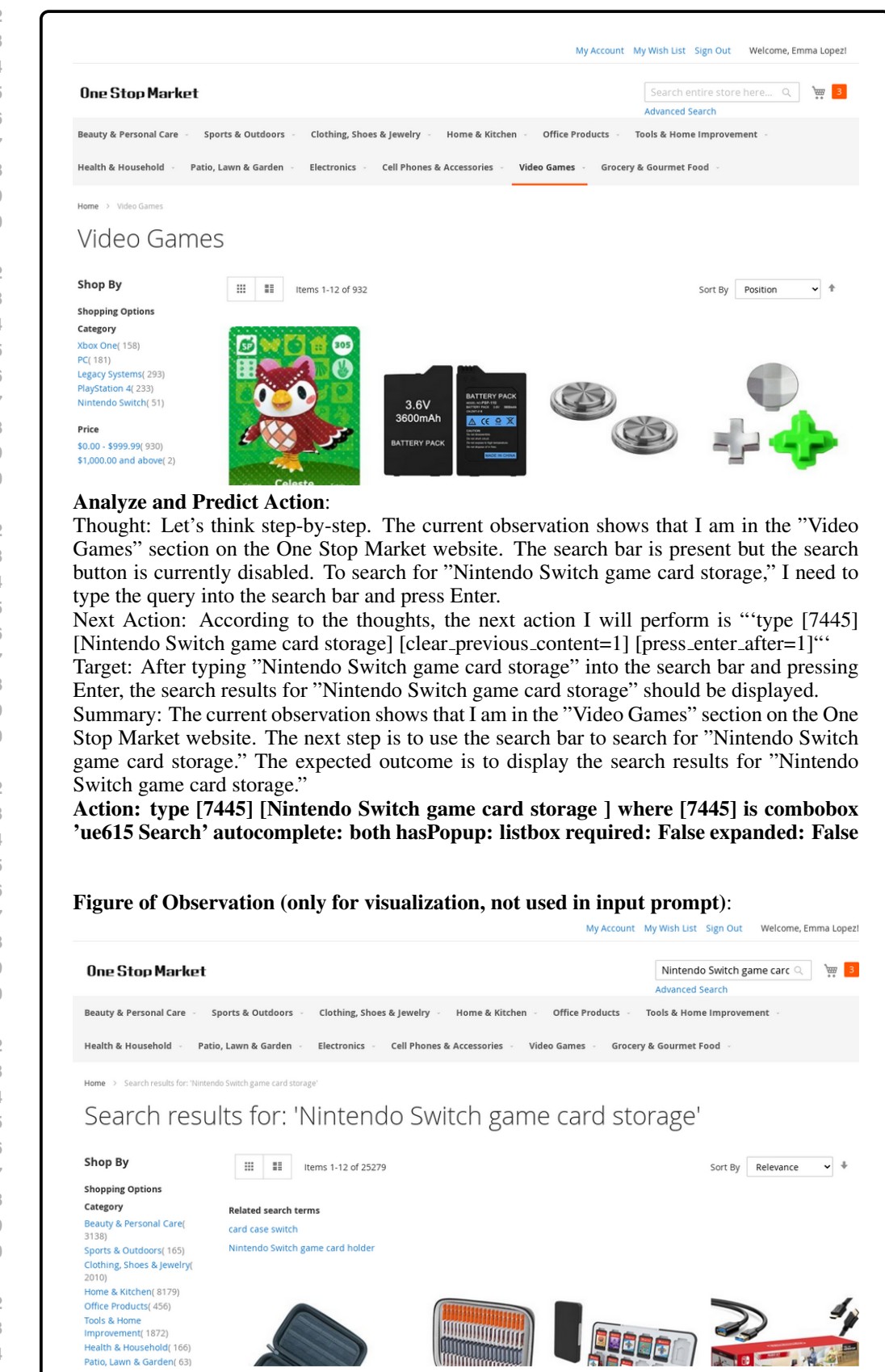

**Analyze and Predict Action**:

Thought: Let's think step-by-step. The current observation shows that I am in the "Video Games" section on the One Stop Market website. The search bar is present but the search button is currently disabled. To search for "Nintendo Switch game card storage," I need to type the query into the search bar and press Enter.

Next Action: According to the thoughts, the next action I will perform is "'type [7445] [Nintendo Switch game card storage] [clear_previous_content=1] [press_enter_after=1]'"

Target: After typing "Nintendo Switch game card storage" into the search bar and pressing Enter, the search results for "Nintendo Switch game card storage" should be displayed.

Summary: The current observation shows that I am in the "Video Games" section on the One Stop Market website. The next step is to use the search bar to search for "Nintendo Switch game card storage." The expected outcome is to display the search results for "Nintendo Switch game card storage."

**Action: type [7445] [Nintendo Switch game card storage ] where [7445] is combobox 'ue615 Search' autocomplete: both hasPopup: listbox required: False expanded: False**

**Figure of Observation (only for visualization, not used in input prompt)**:

**Analyze and Predict Action**:
Thought: Let's think step-by-step. The current observation shows the search results page for "Nintendo Switch game card storage" on the One Stop Market website. This indicates that the search function has already been navigated to and the search has been performed. Therefore, the subtask to search for "Nintendo Switch game card storage" has been completed.
Next Action: According to the thoughts, the next action I will perform is "'finish_subtask"'.
Target: Confirm that the search function has been navigated to and the search results for "Nintendo Switch game card storage" are displayed.
Summary: The current observation shows the search results page for "Nintendo Switch game card storage" on the One Stop Market website. This indicates that the search function has already been navigated to and the search has been performed. Therefore, the subtask to search for "Nintendo Switch game card storage" has been completed.
**Action: finish_subtask []**

**Verification**:
**check_in_webpage("Search bar is populated with 'Nintendo Switch game card storage'")**
1.0 I need to verify if the search bar contains the text 'Nintendo Switch game card storage'. The relevant information should be found in the description of the search bar or its associated elements.
**Verification Pass**

- - - - - - - - - - - - - - - - - - - - - - - - - - - - - - - - - - - - - - - - - - - -

**Subtask 4**
4. **Identify the best storage option for 11 game cards** - Review the search results to find a storage option that can fit at least 11 Nintendo Switch game cards.

**Figure of Observation (only for visualization, not used in input prompt)**:

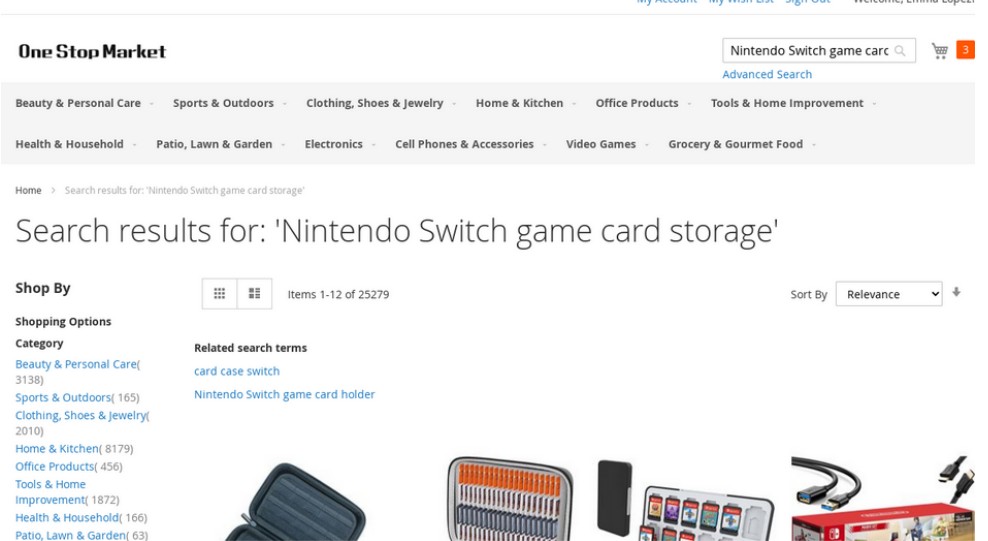

**Analyze and Predict Action**:
Thought: Let's think step-by-step. The current observation shows the search results page for "Nintendo Switch game card storage" on the One Stop Market website. To identify the best storage option for 11 game cards, I need to review the search results and find a storage option that can fit at least 11 Nintendo Switch game cards. The next step is to scroll down to view more details about the search results.
Next Action: According to the thoughts, the next action I will perform is "'scroll [down]"'
Target: View more details about the storage options, including their capacities.

Summary: The current observation shows the search results page for "Nintendo Switch game card storage" on the One Stop Market website. I will scroll down to view more details about the storage options.
**Action: scroll [down]**

**Figure of Observation (only for visualization, not used in input prompt)**:

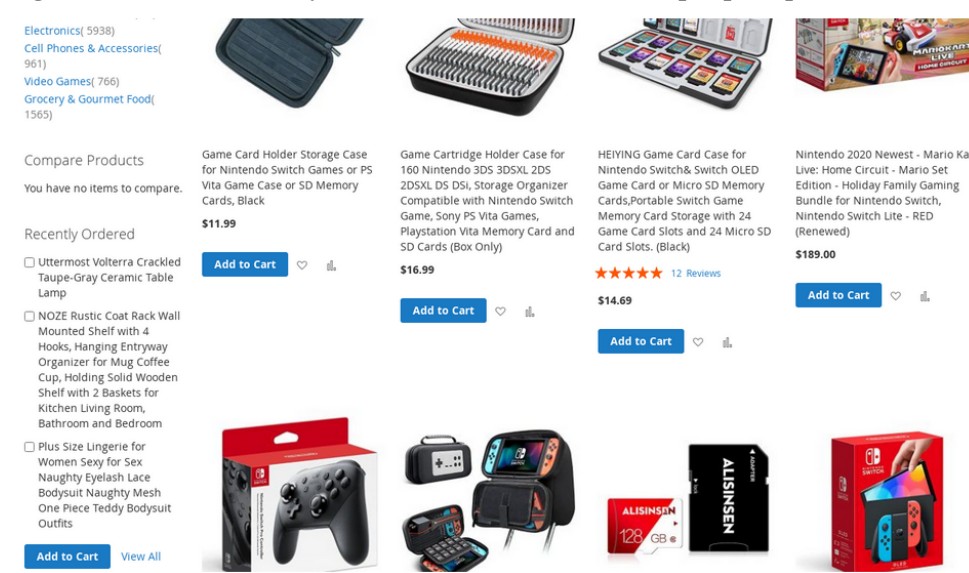

**Analyze and Predict Action**:

Thought: Let's think step-by-step. The current observation shows the search results for "Nintendo Switch game card storage" on the One Stop Market website. I need to identify a storage option that can fit at least 11 Nintendo Switch game cards. From the search results, I can see the following options:

1. **Game Card Holder Storage Case for Nintendo Switch Games or PS Vita Game Case or SD Memory Cards, Black** - $11.99 2. **Game Cartridge Holder Case for 160 Nintendo 3DS 3DSXL 2DS 2DSXL DS DSi, Storage Organizer Compatible with Nintendo Switch Game, Sony PS Vita Games, Playstation Vita Memory Card and SD Cards (Box Only)** - $16.99 3. **HEIYING Game Card Case for Nintendo Switch & Switch OLED Game Card or Micro SD Memory Cards, Portable Switch Game Memory Card Storage with 24 Game Card Slots and 24 Micro SD Card Slots. (Black)** - $14.69

The third option, "HEIYING Game Card Case," has 24 game card slots, which is sufficient for storing 11 game cards. This makes it the best storage option among the listed products.

Next Action: According to the thoughts, the next action I will perform is "'finish_subtask [The best storage option for 11 game cards is the HEIYING Game Card Case for Nintendo Switch & Switch OLED Game Card or Micro SD Memory Cards, Portable Switch Game Memory Card Storage with 24 Game Card Slots and 24 Micro SD Card Slots. (Black) priced at $14.69]"'

Target: Confirm that the subtask of identifying the best storage option for 11 game cards has been completed.

Summary: The current observation shows the search results for "Nintendo Switch game card storage." The best storage option for 11 game cards is the "HEIYING

**Action: finish_subtask [The best storage option for 11 game cards is the HEIYING Game Card Case for Nintendo Switch & Switch OLED Game Card or Micro SD Memory Cards, Portable Switch Game Memory Card Storage with 24 Game Card Slots and 24 Micro SD Card Slots. (Black) priced at $14.69]**

**Verification**:

**Objective: check_in_webpage("Storage option for 11 Nintendo Switch game cards")**
0.0 The search results provide various storage options for Nintendo Switch game cards. I need to verify if any of these options specifically mention the capacity to store 11 game cards.
**Verification Fail**

- - - - - - - - - - - - - - - - - - - - - - - - - - - - - - - - - - - - - - - - - - - - - - - - - - - -

**Reflection**
Reflection: The execution process involved scrolling down to view more details about the storage options and then selecting the "HEIYING Game Card Case" as the best option for 11 game cards. However, the verification result indicates that the content needs to be checked to ensure that the selected storage option specifically mentions the capacity to store 11 game cards.
The mistake lies in not explicitly verifying that the selected storage option can fit at least 11 game cards. The selected option has 24 game card slots, which meets the requirement, but this detail was not clearly verified in the process.
To solve the task, I should include a step to explicitly verify that the selected storage option can fit at least 11 game cards by checking the product details.
Since the mistake occurred after selecting the storage option but before verifying its capacity, I should reload from the point just before finishing the subtask to include the verification step.
**Reload checkpoint: 1**

- - - - - - - - - - - - - - - - - - - - - - - - - - - - - - - - - - - - - - - - - - - - - - - - - - - -

**Figure of Observation (only for visualization, not used in input prompt):**

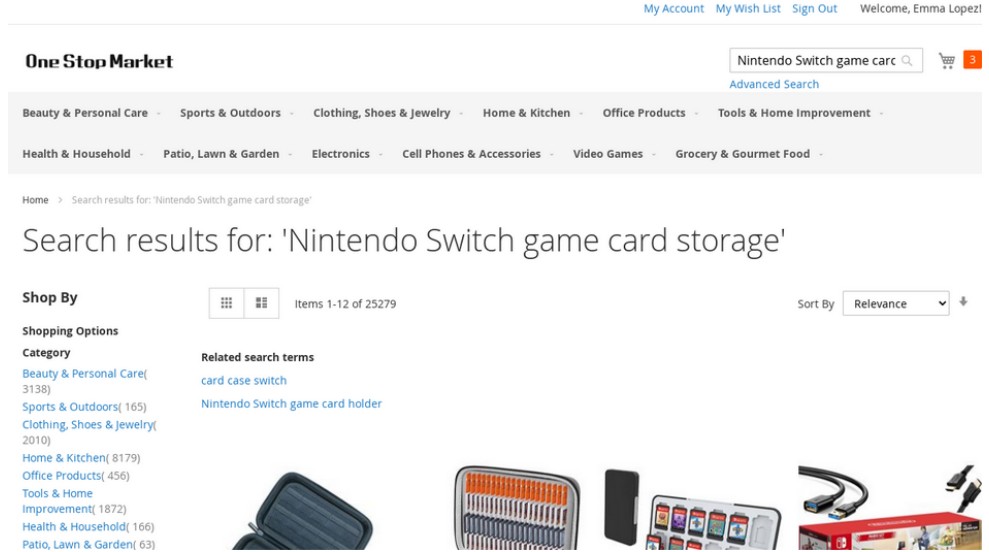

**Analyze and Predict Action:**
Thought: Let's think step-by-step. The current observation shows the search results for "Nintendo Switch game card storage" on the One Stop Market website. I need to identify a storage option that can fit at least 11 Nintendo Switch game cards. From the search results, I can see the following options:
1. **Game Card Holder Storage Case for Nintendo Switch Games or PS Vita Game Case or SD Memory Cards, Black** - $11.99 2. **Game Cartridge Holder Case for 160 Nintendo 3DS 3DSXL 2DS 2DSXL DS DSi, Storage Organizer Compatible with Nintendo Switch Game, Sony PS Vita Games, Playstation Vita Memory Card and SD Cards (Box Only)** - $16.99 3. **HEIYING Game Card Case for Nintendo Switch & Switch OLED Game Card or Micro SD Memory Cards, Portable Switch Game Memory Card Storage with 24 Game Card Slots and 24 Micro SD Card Slots. (Black)** - $14.69

The third option, "HEIYING Game Card Case," has 24 game card slots, which is sufficient for storing 11 game cards. This makes it the best storage option among the listed products.
Next Action: According to the thoughts, the next action I will perform is "'click [14241]"'
Target: Navigate to the product page of the "HEIYING Game Card Case for Nintendo Switch & Switch OLED Game Card or Micro SD Memory Cards,Portable Switch Game Memory Card Storage with 24 Game Card Slots and 24 Micro SD Card Slots. (Black)"
Summary: The current observation shows the search results for "Nintendo Switch game card storage" on the One Stop Market website. I will click on the link for the "HEIYING Game Card Case" to navigate to its product page and verify its details.
**Action: click [14241] where [14241] is link 'HEIYING Game Card Case for Nintendo Switch& Switch OLED Game Card or Micro SD Memory Cards,Portable Switch Game Memory Card Storage with 24 Game Card Slots and 24 Micro SD Card Slots. (Black)'**

**Figure of Observation (only for visualization, not used in input prompt)**:

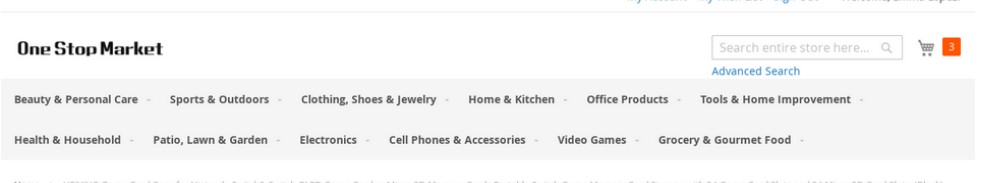

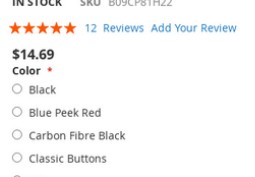

**Analyze and Predict Action**:
Thought: Let's think step-by-step. The current page provides detailed information about the "HEIYING Game Card Case for Nintendo Switch& Switch OLED Game Card or Micro SD Memory Cards, Portable Switch Game Memory Card Storage with 24 Game Card Slots and 24 Micro SD Card Slots." This confirms that the product meets the requirement of fitting at least 11 game cards, as it has 24 game card slots. I will now finish the subtask.
Next Action: According to the thoughts, the next action I will perform is "'finish_subtask [The best storage option for 11 game cards is the HEIYING Game Card Case for Nintendo Switch & Switch OLED Game Card or Micro SD Memory Cards, Portable Switch Game Memory Card Storage with 24 Game Card Slots and 24 Micro SD Card Slots. (Black) priced at $14.69]"'
Target: Confirm that the subtask of identifying the best storage option for 11 game cards has been completed.
Summary: The current observation confirms that the "HEIYING Game Card Case" has 24 game card slots, making it a suitable option for storing at least 11 Nintendo Switch game cards. Therefore, I will finish the subtask.
**Action: finish_subtask [The best storage option for 11 game cards is the HEIYING Game Card Case for Nintendo Switch & Switch OLED Game Card or Micro SD Memory Cards, Portable Switch Game Memory Card Storage with 24 Game Card Slots and 24 Micro SD Card Slots. (Black) priced at $14.69]**

**Verification**:
**Objective: check_in_webpage("Storage option for 11 Nintendo Switch game cards")**
1.0 The content describes a game card case for the Nintendo Switch that has 24 game card slots and 24 micro SD card slots. The subgoal target is to find a storage option for 11 Nintendo Switch game cards.
**Verification Pass**

- - - - - - - - - - - - - - - - - - - - - - - - - - - - - - - - - - - - - - - -

**Subtask 5**
5. **Navigate to the product page of the identified storage option** - Click on the identified storage option to navigate to its product page for more details.
**Figure of Observation (only for visualization, not used in input prompt)**:

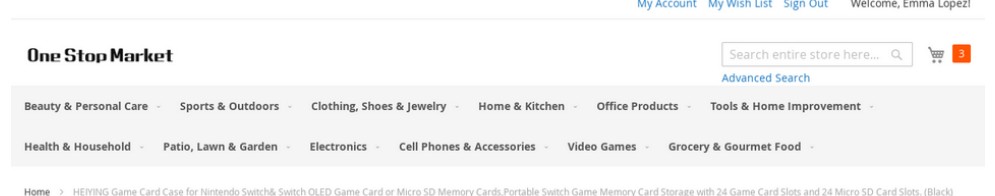

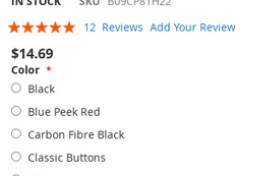

**Analyze and Predict Action**:
Thought: Let's think step-by-step. The current observation indicates that I am already on the product page for the "HEIYING Game Card Case for Nintendo Switch& Switch OLED Game Card or Micro SD Memory Cards,Portable Switch Game Memory Card Storage with 24 Game Card Slots and 24 Micro SD Card Slots. (Black)". Therefore, the subtask to navigate to the product page has already been completed.
Next Action: According to the thoughts, the next action I will perform is "'finish_subtask'".
Target: Confirm that I have navigated to the product page of the identified storage option.
Summary: I am already on the product page for the "HEIYING Game Card Case for Nintendo Switch& Switch OLED Game Card or Micro SD Memory Cards,Portable Switch Game Memory Card Storage with 24 Game Card Slots and 24 Micro SD Card Slots. (Black)". I will now confirm that this subtask is complete.
**Action: finish_subtask []**

**Verification**:
check_in_url("nintendo-switch-game-card-storage") |OR|check_in_webpage("You are on the product page of the identified storage option")
**check_in_url("nintendo-switch-game-card-storage")**
1.0 "nintendo-switch-game-card-storage" is in the URL: http://127.0.0.1:7770/heiying-game-card-case-for-nintendo-switch-switch-oled-game-card-or-micro-sd-memory-cards-portable-switch-game-memory-card-storage-with-24-game-card-slots-and-24-micro-sd-card-slots-black.html
**check_in_webpage("You are on the product page of the identified storage option")**
1.0 The content provided includes detailed information about a product, specifically the "HEIYING Game Card Case for Nintendo Switch & Switch OLED Game Card or Micro SD Memory Cards." This indicates that I am indeed on a product page for a storage option.

**Verification Pass**

- - - - - - - - - - - - - - - - - - - - - - - - - - - - - - - - - - - - - - - - -

**Answer**:
The best storage option for your 11 Nintendo Switch game cards is the **HEIYING Game Card Case for Nintendo Switch & Switch OLED Game Card or Micro SD Memory Cards**. This case offers 24 game card slots, which is more than sufficient for your needs. It is priced at $14.69 and is available in multiple colors. You can find it on the One Stop Market website.
**Last URL**: http://127.0.0.1:7770/heiying-game-card-case-for-nintendo-switch-switch-oled-game-card-or-micro-sd-memory-cards-portable-switch-game-memory-card-storage-with-24-game-card-slots-and-24-micro-sd-card-slots-black.html

## C.2 REPLANNING

One Complete Example

**Query Intent**:
Show me the billing address for order number 00178.
**Ground Truth**
Final Answer: 'must_include': ['101 S San Mateo Dr', 'San Mateo', 'California', '94010', 'United States']

- - - - - - - - - - - - - - - - - - - - - - - - - - - - - - - - - - - - - - - - -

**Planning**
1. **Navigate to the "My Account" section** - Click on the "My Account" link.
- Objective: 'check_in_url("account") |OR|check_in_webpage("You have arrived at the 'My Account' page")'
2. **Locate and access the "Order History" or "Orders" section** - Find and click on the link or tab that leads to order history or orders.
- Objective: 'check_in_url("order-history") |OR|check_in_webpage("Order History")'
3. **Search for order number 00178 in the list of orders** - Look for order number 00178 in the order history list.
- Objective: 'check_in_webpage("Order number 00178")'
4. **Access the details of order number 00178** - Click on the order number 00178 to view its details.
- Objective: 'check_in_webpage("Order details for 00178")'
5. **Identify and display the billing address for order number 00178** - Locate the billing address section within the order details and display it.
- Objective: 'check_in_webpage("Billing address for order number 00178")'

- - - - - - - - - - - - - - - - - - - - - - - - - - - - - - - - - - - - - - - - -

**Subtask 1**
1. **Navigate to the "My Account" section** - Click on the "My Account" link.

**Figure of Observation (only for visualization, not used in input prompt)**:

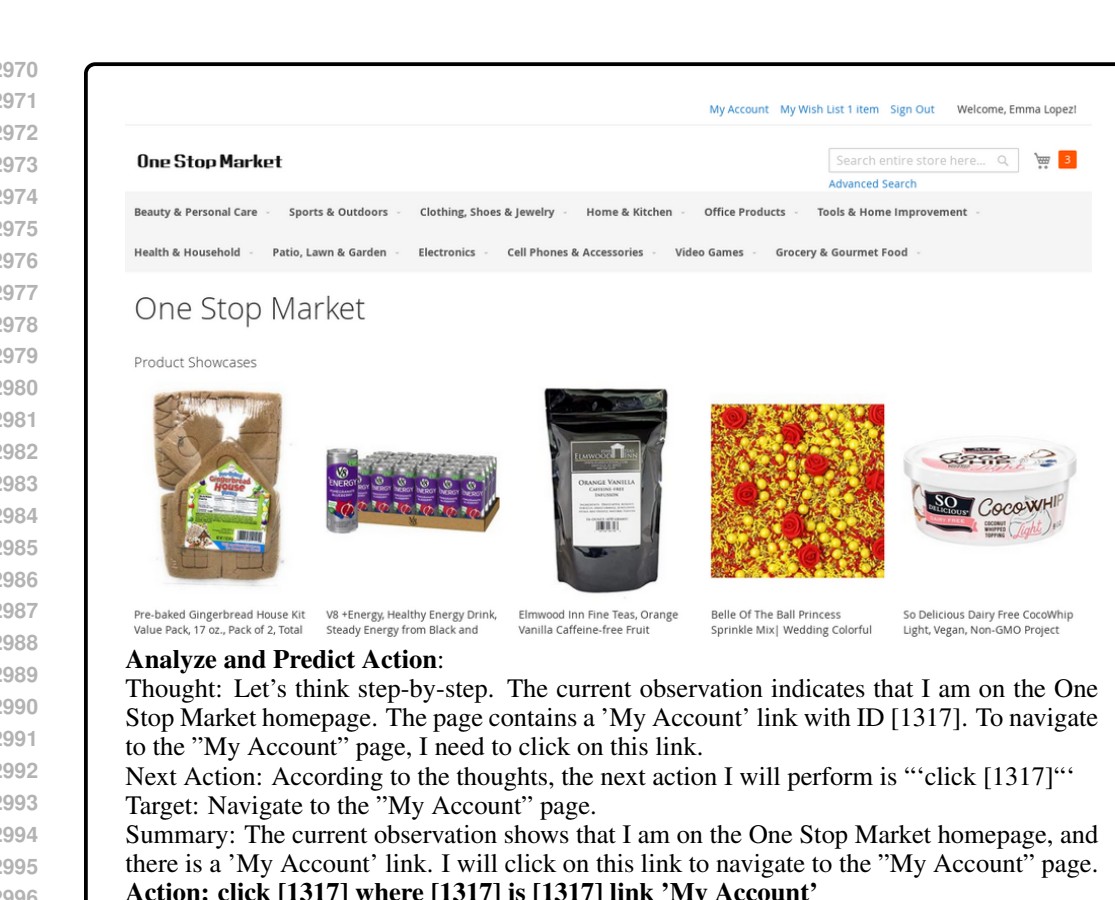

**Analyze and Predict Action**:

Thought: Let's think step-by-step. The current observation indicates that I am on the One Stop Market homepage. The page contains a 'My Account' link with ID [1317]. To navigate to the "My Account" page, I need to click on this link.

Next Action: According to the thoughts, the next action I will perform is "'click [1317]'"

Target: Navigate to the "My Account" page.

Summary: The current observation shows that I am on the One Stop Market homepage, and there is a 'My Account' link. I will click on this link to navigate to the "My Account" page.

**Action: click [1317] where [1317] is [1317] link 'My Account'**

**Figure of Observation (only for visualization, not used in input prompt)**:

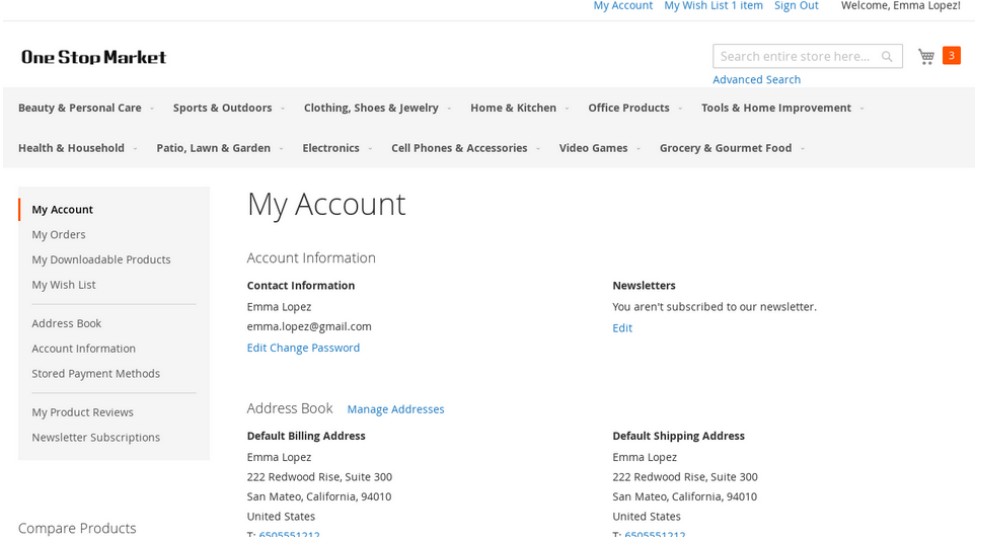

**Analyze and Predict Action**:

Thought: The current observation indicates that I have successfully navigated to the "My Account" page. The page contains various elements such as account information, contact information, address book, and links to other account-related sections. Since the objective was to navigate to the "My Account" page, I have achieved this subtask.

Next Action: According to the thoughts, the next action I will perform is "'finish_subtask'"

Target: Confirm that the subtask of navigating to the "My Account" page is complete.
Summary: The current observation shows that I am on the "My Account" page, which contains various account-related information and links. Therefore, I will finish the subtask as it has been successfully completed.
**Action: finish_subtask []**

**Verification**:
**'check_in_url("account")**
1.0 "account" is in the URL: http://127.0.0.1:7770/customer/account/
**check_in_webpage("You have arrived at the 'My Account' page")'**
1.0 The content of the webpage includes multiple elements that are typically found on a 'My Account' page, such as account information, contact information, address book, and links to manage various account-related settings. Additionally, the heading 'My Account' is present, which confirms the page's identity.
**Verification Pass**

- - - - - - - - - - - - - - - - - - - - - - - - - - - - - - - - - - - - - - - -

**Subtask 2**
**Locate and access the "Order History" or "Orders" section** - Find and click on the link or tab that leads to order history or orders.

**Figure of Observation (only for visualization, not used in input prompt)**:

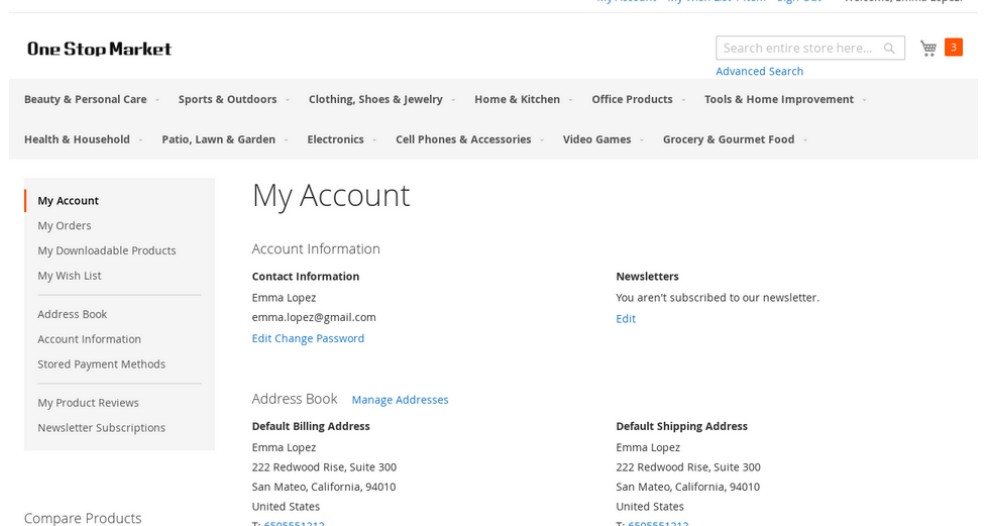

**Analyze and Predict Action**:
Thought: The current observation shows that I am on the "My Account" page. There is a link labeled "My Orders" with ID [4527]. Clicking on this link should navigate to the "Order History" or "Orders" section.
Next Action: According to the thoughts, the next action I will perform is "'click [4527]"'
Target: Navigate to the "Order History" or "Orders" section.
Summary: The current observation shows that I am on the "My Account" page, and there is a link labeled "My Orders" with ID [4527]. Clicking on this link should navigate to the "Order History" or "Orders" section.
**Action: click [4527] where [4527] is link 'My Orders'**

**Figure of Observation (only for visualization, not used in input prompt)**:

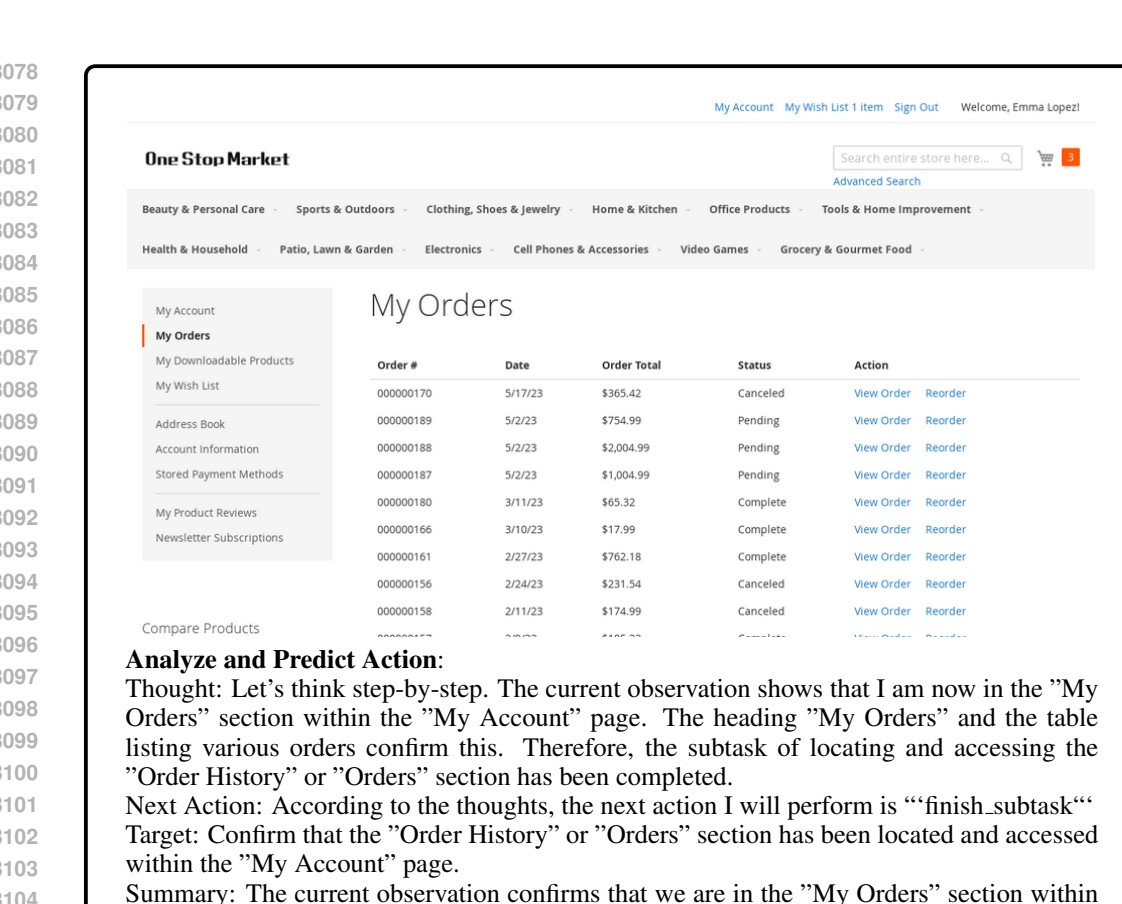

**Analyze and Predict Action**:

Thought: Let's think step-by-step. The current observation shows that I am now in the "My Orders" section within the "My Account" page. The heading "My Orders" and the table listing various orders confirm this. Therefore, the subtask of locating and accessing the "Order History" or "Orders" section has been completed.

Next Action: According to the thoughts, the next action I will perform is "'finish_subtask'"

Target: Confirm that the "Order History" or "Orders" section has been located and accessed within the "My Account" page.

Summary: The current observation confirms that we are in the "My Orders" section within the "My Account" page, as indicated by the heading "My Orders" and the table listing various orders. Therefore, the subtask of locating and accessing the "Order History" or "Orders" section has been completed.

**Action: finish_subtask []**

**Verification**:

Objective: check_in_url("search") |OR|check_in_webpage("Search results for 'Canon photo printer' displayed"

**check_in_url("order-history")**

1.0 "order-history" is in the URL: http://127.0.0.1:7770/sales/order/history/

**check_in_webpage("Order History")'**

1.0 The content provided is a detailed view of the "My Orders" page, which includes a table listing various orders with their respective order numbers, dates, totals, statuses, and actions. This matches the expected content for an "Order History" page

**Verification Pass**

- - - - - - - - - - - - - - - - - - - - - - - - - - - - - - - - - - - - - - - - - -

**Subtask 3**

3. **Search for order number 00178 in the list of orders** - Look for order number 00178 in the order history list.

**Figure of Observation (only for visualization, not used in input prompt)**:

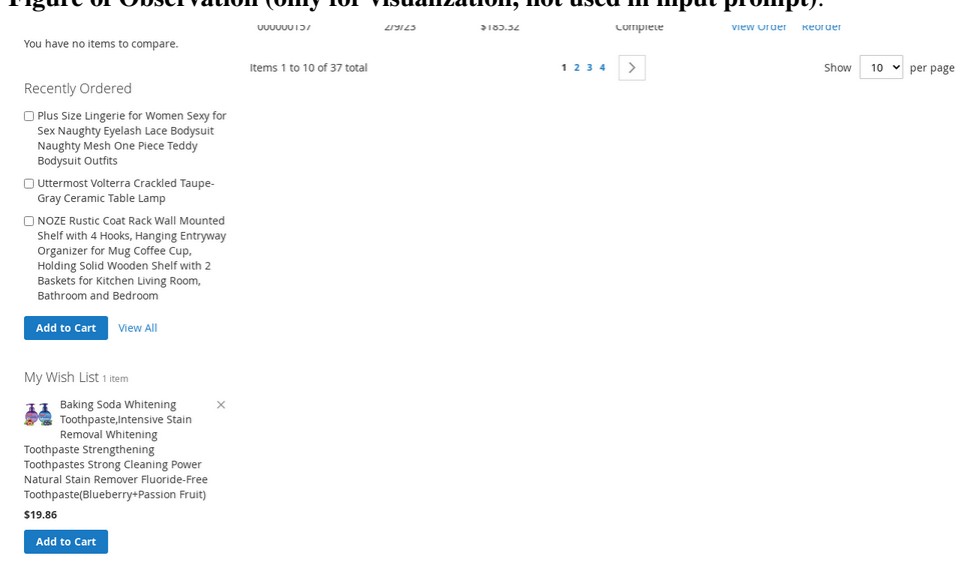

**Analyze and Predict Action**:

Thought: Let's think step-by-step. The current observation shows that I am on the "My Orders" page, and the list of orders is displayed. However, order number "00178" is not visible in the current view. There are multiple pages of orders, and I need to scroll down to check if it is listed further down or navigate to subsequent pages if necessary.

Next Action: According to the thoughts, the next action I will perform is "'scroll [down]'"

Target: Scroll down the "My Orders" page to check if order number "00178" is listed further down.

Summary: The current observation shows that order number "00178" is not visible in the current view of the "My Orders" page. I will scroll down to check if it is listed further down. The expected outcome of this action is to see more orders listed on the page.

**Action: scroll [down]**

**Figure of Observation (only for visualization, not used in input prompt)**:

**Analyze and Predict Action**:

Thought: Let's think step-by-step. The current observation shows that order number "00178" is not visible in the current view of the "My Orders" page. There are multiple

pages of orders, and I need to navigate to the next page to check if order number "00178" is listed there. I will click on the link for "Page 2" to navigate to the next page of orders.
Next Action: According to the thoughts, the next action I will perform is "'click [7801]"'
Target: Navigate to the next page of orders to locate order number "00178".
Summary: The current observation shows that order number "00178" is not visible on the current page of "My Orders". There are multiple pages of orders, and I will click on the link for "Page 2" to navigate to the next page and check if order number "00178" is listed there. The expected outcome of this action is to navigate to the next page of orders.
**Action: click [7801] where [7801] is link 'Page 2'**

**Figure of Observation (only for visualization, not used in input prompt)**:

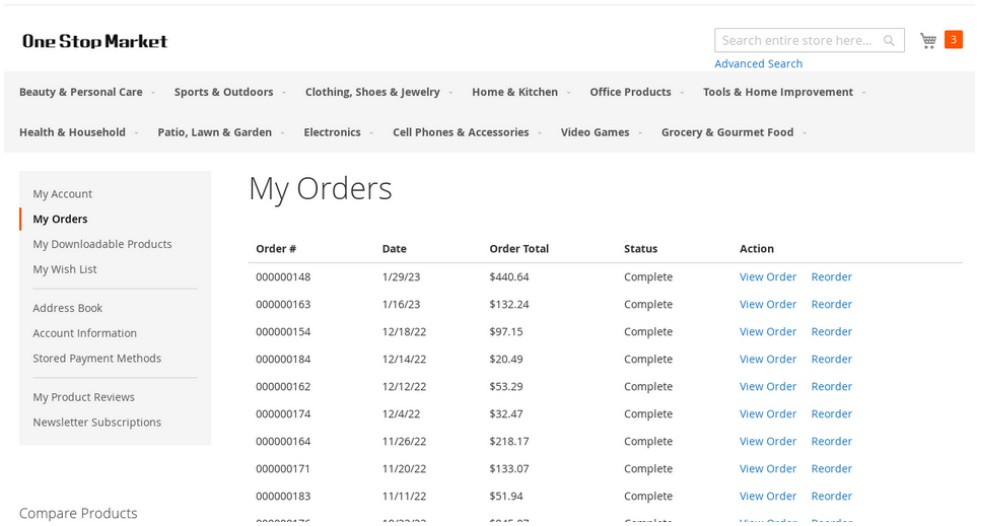

**Analyze and Predict Action**:
Thought: Let's think step-by-step. The current observation shows that I am on the "My Orders" page, specifically on Page 2. The table lists various orders, but order number "00178" is not visible on this page. There are multiple pages of orders, and I need to navigate to the next page to check if order number "00178" is listed there. I will click on the link for "Page 3" to navigate to the next page of orders.
Next Action: According to the thoughts, the next action I will perform is "'click [7802]"'
Target: Navigate to the next page of orders to locate order number "00178".
Summary: The current observation shows that order number "00178" is not visible on the current page of "My Orders". There are multiple pages of orders, and I will click on the link for "Page 3" to navigate to the next page and check if order number "00178" is listed there. The expected outcome of this action is to navigate to the next page of orders.
**Action: click [7802] where [7802] is No match found**

**Figure of Observation (only for visualization, not used in input prompt)**:

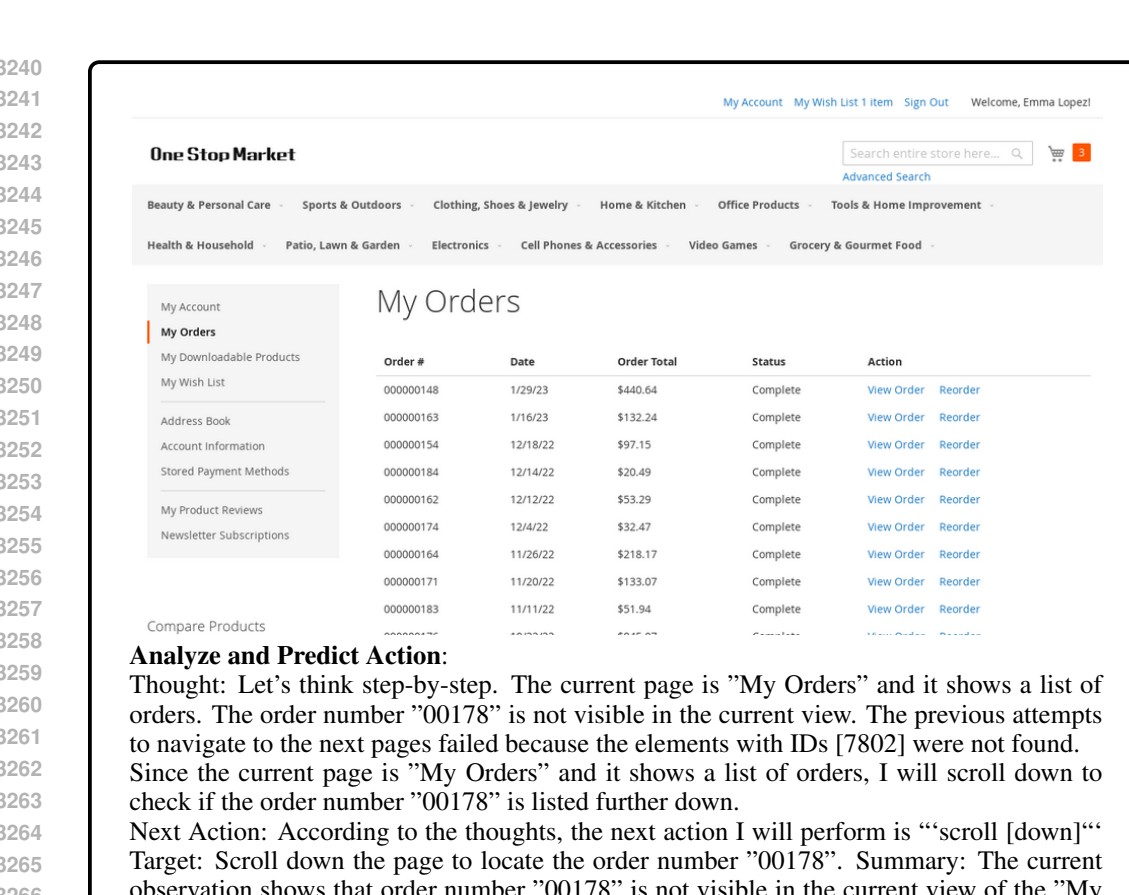

**Analyze and Predict Action**:

Thought: Let's think step-by-step. The current page is "My Orders" and it shows a list of orders. The order number "00178" is not visible in the current view. The previous attempts to navigate to the next pages failed because the elements with IDs [7802] were not found. Since the current page is "My Orders" and it shows a list of orders, I will scroll down to check if the order number "00178" is listed further down.

Next Action: According to the thoughts, the next action I will perform is "'scroll [down]'" Target: Scroll down the page to locate the order number "00178". Summary: The current observation shows that order number "00178" is not visible in the current view of the "My Orders" page. I will scroll down to check if it is listed further down. The expected outcome of this action is to see more orders listed on the page.

**Action: scroll [down]**

**Figure of Observation (only for visualization, not used in input prompt)**:

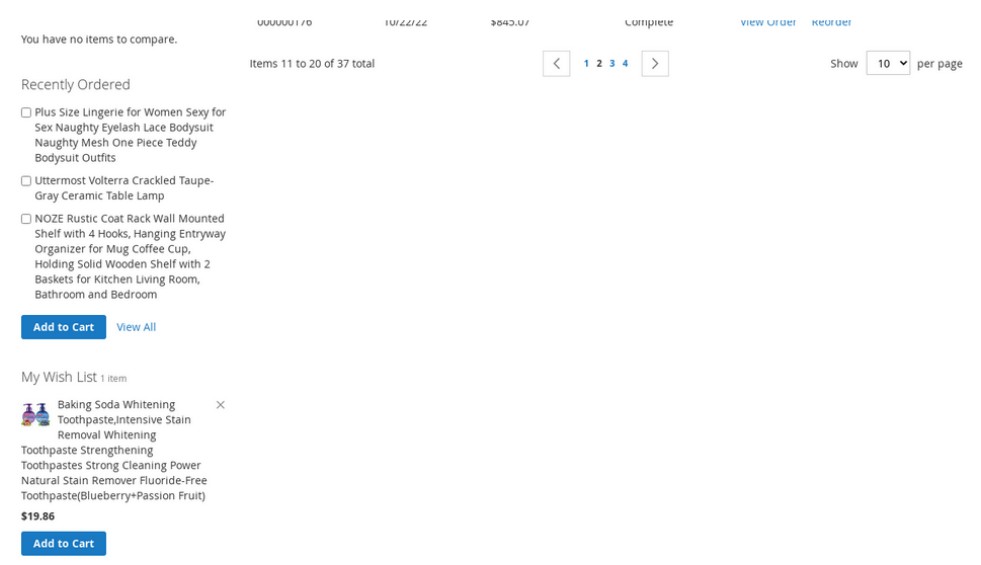

**Analyze and Predict Action**:

Thought: Let's think step-by-step. The current observation shows that order number "00178" is not visible on the current page of "My Orders". There are multiple pages of

orders, and I will click on the link for "Page 3" to navigate to the next page and check if order number "00178" is listed there. The expected outcome of this action is to navigate to the next page of orders.

Next Action: According to the thoughts, the next action I will perform is "'click [10365]'"

Target: Navigate to Page 3 of the order history to check for order number "00178".

Summary: The current observation shows that order number "00178" is not visible on the current page of "My Orders". There are multiple pages of orders, and I will click on the link for "Page 3" to navigate to the next page and check if order number "00178" is listed there. The expected outcome of this action is to navigate to the next page of orders.

**Action: click [10365] where [10365] is [10365] link 'Page 3'**

Reached step limit within one subtask. Enter verification.

**Verification**:

**check_in_webpage("Order number 00178")**

0.0 I need to verify if the order number "00178" is present in the content provided. I will look for the specific order number in the table of orders.

**Verification Fail**

- - - - - - - - - - - - - - - - - - - - - - - - - - - - - - - - - - - - - - - - - - - - - - - - - -

**Replanning**

The error in the previous execution was due to the discrepancy between the order number formats "00178" and "000000178." The verification result indicates that the order number "000000178" should be confirmed as the target order number "00178" despite the difference in format. To resolve this, the new subtasks must include steps to verify the order number and then proceed to identify and display the billing address.

New Subtasks: 3. Verify that order number "000000178" corresponds to the target order number "00178" despite the difference in format. Objective: check_in_webpage("Order number 00178")

4. Access the details of order number "000000178." Objective: check_in_webpage("Order 000000178")

5. Identify and display the billing address for order number "000000178." Objective: check_in_webpage("Billing Address")

These new subtasks should ensure that the order number is verified correctly and that the billing address for the order is identified and displayed. This should resolve the previous issues and achieve the final objective.

- - - - - - - - - - - - - - - - - - - - - - - - - - - - - - - - - - - - - - - - - - - - - - - - - -

**New Subtask 3**

3. Verify that order number "000000178" corresponds to the target order number "00178" despite the difference in format.

**Figure of Observation (only for visualization, not used in input prompt)**:

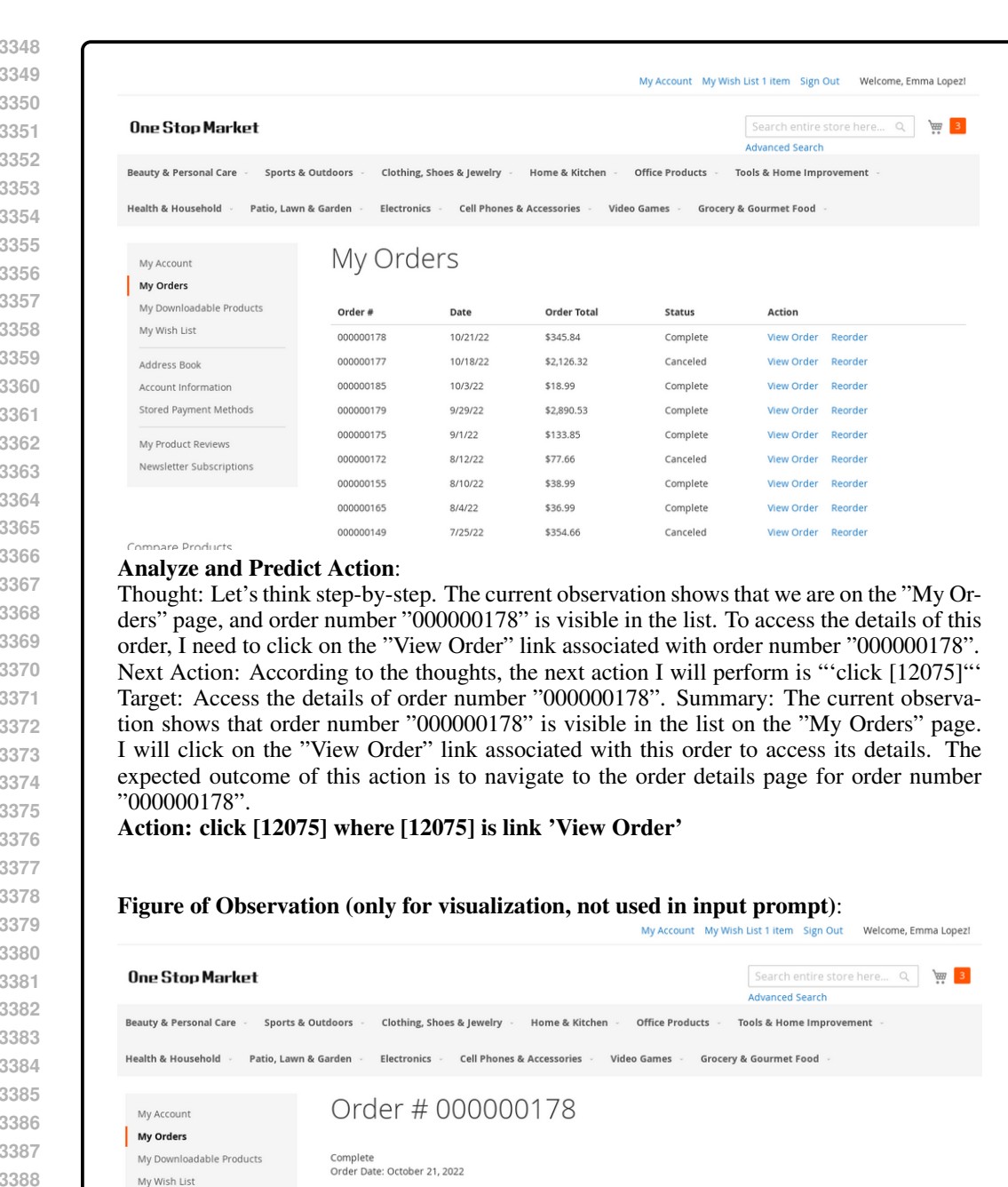

**Analyze and Predict Action**:
Thought: Let's think step-by-step. The current observation shows that we are on the "My Orders" page, and order number "000000178" is visible in the list. To access the details of this order, I need to click on the "View Order" link associated with order number "000000178". Next Action: According to the thoughts, the next action I will perform is "'click [12075]'" Target: Access the details of order number "000000178". Summary: The current observation shows that order number "000000178" is visible in the list on the "My Orders" page. I will click on the "View Order" link associated with this order to access its details. The expected outcome of this action is to navigate to the order details page for order number "000000178".

**Action: click [12075] where [12075] is link 'View Order'**

**Figure of Observation (only for visualization, not used in input prompt)**:

**Analyze and Predict Action**:
Thought: Let's think step-by-step. The current observation shows the details of order number "000000178". To verify that this order corresponds to the target order number "00178" despite the difference in format, I need to compare the significant digits of both order num-

bers. The significant digits of "000000178" are "178", which matches the target order number "00178".

Next Action: According to the thoughts, the next action I will perform is "'finish_subtask [The order number "000000178" corresponds to the target order number "00178" despite the difference in format.]"'

Target: Confirm that the order number "000000178" corresponds to the target order number "00178".

Summary: The current observation shows the details of order number "000000178". The significant digits of "000000178" are "178", which matches the target order number "00178". The next step involves confirming that the order number "000000178" corresponds to the target order number "00178" despite the difference in format. The expected outcome of this action is to confirm that the order number "000000178" corresponds to the target order number "00178".

**Action: finish_subtask [The order number "000000178" corresponds to the target order number "00178" despite the difference in format.]**

**Verification**:
**Objective: check_in_webpage("Order number 00178")**
1.0 The subgoal target is to verify if the order number is 00178. The content provided shows the order number as 000000178.
**Verification Pass**

- - - - - - - - - - - - - - - - - - - - - - - - - - - - - - - - - - - - - - - - - - - - - - - - - - - - -

**New Subtask 4**
4. Access the details of order number "000000178."

**Figure of Observation (only for visualization, not used in input prompt)**:

**Analyze and Predict Action**:
Thought: Let's think step-by-step. The current observation shows that I am already on the "Order 000000178" page, which displays the details of order number 178. The details include the order date, status, items ordered, and the total cost. I have achieved the objective of accessing the details of order number 178. I will issue the 'finish_subtask' action to confirm this.

Next Action: According to the thoughts, the next action I will perform is "'finish_subtask"'

Target: Confirm that the details of order number 178 have been accessed.

Summary: The current observation section specifies elements present on the 'Order 000000178' page, including the order date, status, items ordered, and the total cost. The next step involves confirming that the details of order number 178 have been accessed. The

expected outcome of this action is to confirm that the details of order number 178 have been accessed.

**Action: finish_subtask []**

**Verification**:

**Objective: check_in_webpage("Order number 00178")**

1.0 The content provided is a detailed webpage structure for Order  000000178. It includes various elements such as links, static texts, headings, and a table with order details.  The presence of the heading "Order  000000178" and the detailed order information confirms that the content is indeed related to Order  000000178.

**Verification Pass**

- - - - - - - - - - - - - - - - - - - - - - - - - - - - - - - - - - - - - - - - - - -

**New Subtask 5**

Identify and display the billing address for order number "000000178."

**Figure of Observation (only for visualization, not used in input prompt)**:

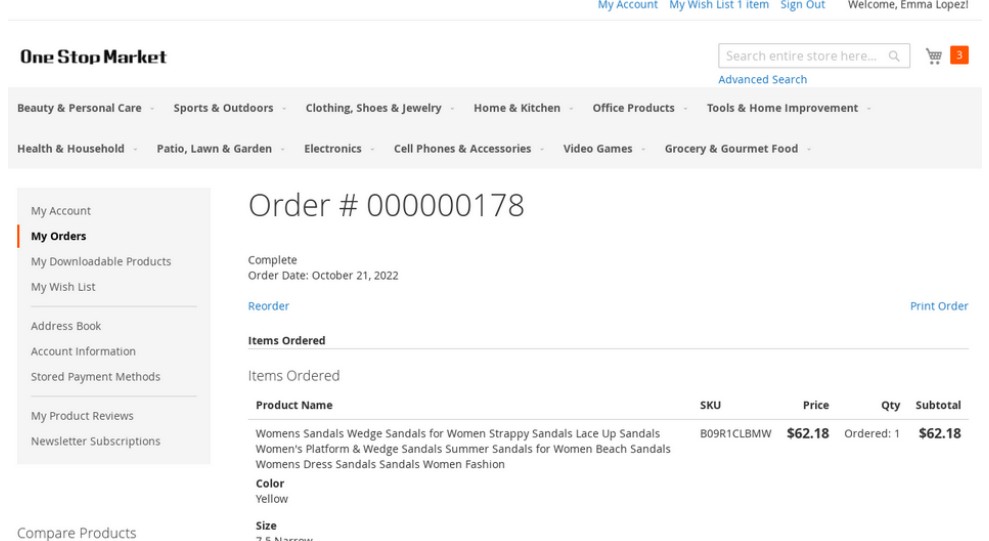

**Analyze and Predict Action**:

Thought: Let's think step-by-step. The current observation shows the order details page for order 000000178. To identify and display the billing address for this order, I need to locate the section that contains the billing address information. This information is typically found in the order details page, often near the shipping address or payment method sections.

Next Action: According to the thoughts, the next action I will perform is "'scroll [down]"'

Target: Locate the section that contains the billing address information for order 000000178.

Summary: The current observation shows the order details page for order 000000178. To identify and display the billing address for this order, I will scroll down the page to locate the section that contains the billing address information. The expected outcome of this action is to find the billing address information for order 000000178

**Action: scroll [down]**

**Figure of Observation (only for visualization, not used in input prompt)**:

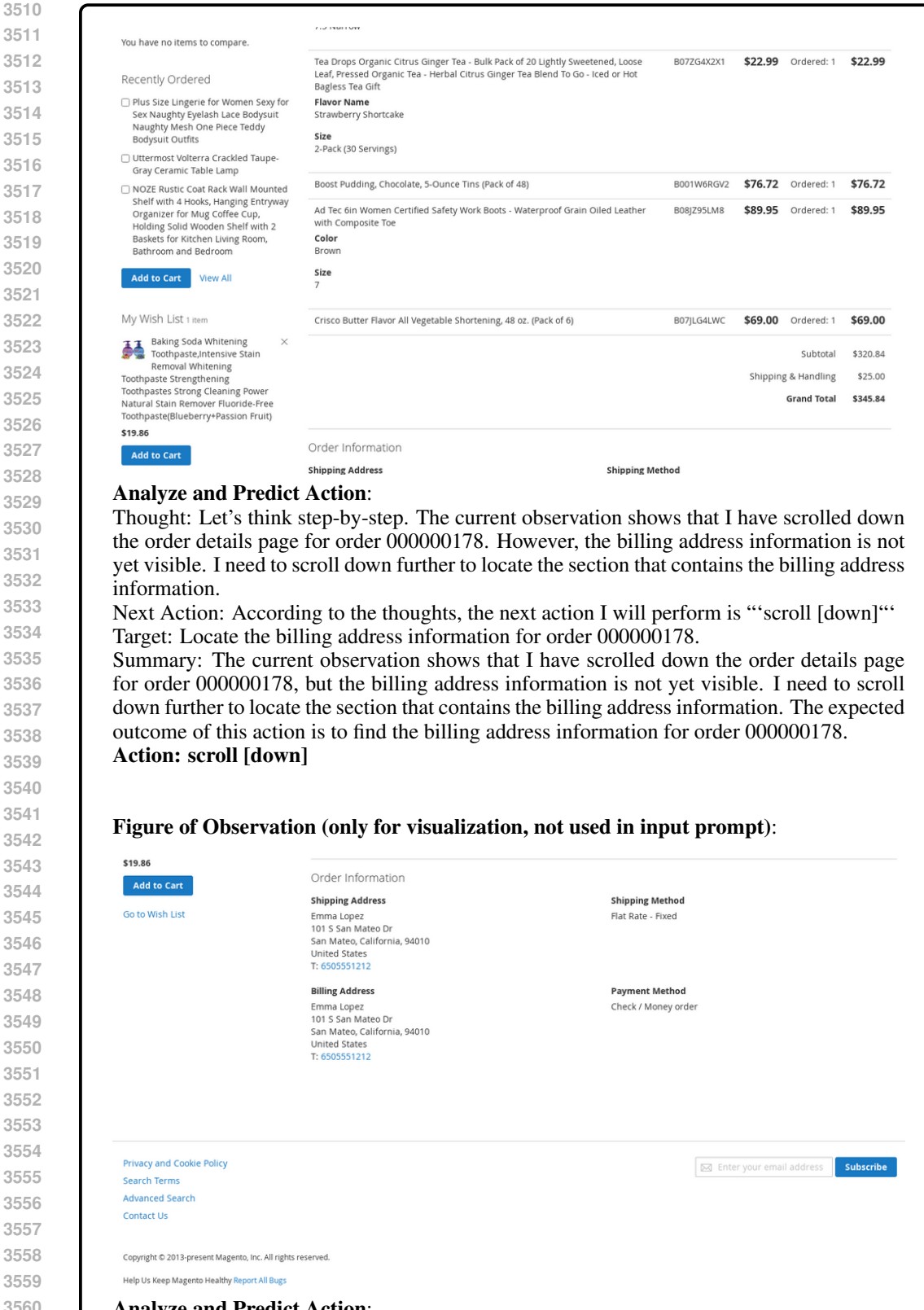

**Analyze and Predict Action**:
Thought: Let's think step-by-step. The current observation shows that I have scrolled down the order details page for order 000000178. However, the billing address information is not yet visible. I need to scroll down further to locate the section that contains the billing address information.
Next Action: According to the thoughts, the next action I will perform is "'scroll [down]"'
Target: Locate the billing address information for order 000000178.
Summary: The current observation shows that I have scrolled down the order details page for order 000000178, but the billing address information is not yet visible. I need to scroll down further to locate the section that contains the billing address information. The expected outcome of this action is to find the billing address information for order 000000178.
**Action: scroll [down]**

**Figure of Observation (only for visualization, not used in input prompt)**:

**Analyze and Predict Action**:
Thought: Let's think step-by-step. The current observation shows that I have successfully located the billing address information for order number "000000178". The billing address

is listed under the "Billing Address" section and includes the following details: - Name: Emma Lopez - Address: 101 S San Mateo Dr - City: San Mateo, California, 94010 - Country: United States - Phone: 6505551212

Next Action: According to the thoughts, the next action I will perform is "'finish_subtask [Emma Lopez, 101 S San Mateo Dr, San Mateo, California, 94010, United States, 6505551212]"'

Target: Confirm that the billing address for order number "000000178" has been identified and displayed.

Summary: The current observation confirms that the billing address for order number "000000178" has been located and includes the following details: Emma Lopez, 101 S San Mateo Dr, San Mateo, California, 94010, United States, 6505551212. Therefore, the subtask of identifying and displaying the billing address for order number "000000178" has been completed.

**Action: finish_subtask [Emma Lopez, 101 S San Mateo Dr, San Mateo, California, 94010, United States, 6505551212]**

**Verification**:
**check_in_webpage("Billing Address")**
1.0 The subgoal target is to determine and display the price range for 'Canon photo printer' products. The summary indicates that the prices have been identified, with a minimum price of $2.56 and a maximum price of $649.99. This information aligns with the subgoal target.
**Verification Pass**

- - - - - - - - - - - - - - - - - - - - - - - - - - - - - - - - - - - - - - - - - - - - - - - - - -

**Answer**:
The billing address for order number 00178 is: **Emma Lopez** 101 S San Mateo Dr San Mateo, California, 94010 United States T: [6505551212](tel:6505551212)

