# OpenReview forum: "VeSX: A Framework Featured by Verification, Self-Correction and In-context Learning for Web Automation Tasks"
_ICLR.cc/2025/Conference — Submitted to ICLR 2025_

### Official Review · Reviewer_MnFm · 2024-10-25

**Soundness:** 2
**Presentation:** 1
**Contribution:** 2
**Rating:** 3
**Confidence:** 4

**Summary:**

The paper presents VeSX, a framework designed to enhance web automation tasks by improving the subtask feasibility of Large Language Models (LLMs). It addresses the challenges of error-prone workflows by introducing three key components: (1) Subgoal-Guided Verification, which ensures that subtasks are completed correctly by generating subgoals during the planning phase and verifying the execution results against those subgoals, (2) Hierarchical Self-Correction, which adds layers of error correction during both the planning and execution phases. If mistakes occur, the model first reflects on its actions, and if needed, replans the task, (3) Exemplar Bank for In-Context Learning, which uses stored examples of previous tasks to help the model learn from experience and improve performance on future tasks.

**Strengths:**

Originality: The paper presents VeSX, a framework that introduces a combination of verification, self-correction, and in-context learning for web automation tasks. The approach is notable for its hierarchical self-correction mechanism, which allows the model to reflect on errors and replan, addressing potential common challenges.

Quality: The idea proposed in this paper is straightforward and clear. The overall structure is clear, despite some minor confusion.

Clarity: Key concepts such as subgoal-guided verification and hierarchical self-correction are explained straightforwardly, and the diagrams effectively support the explanations.

Significance: VeSX addresses a common issue in web automation—handling subtask failures and error correction. Its ability to autonomously verify and correct errors while using in-context learning is a useful enhancement.

**Weaknesses:**

While the paper proposes an interesting framework for web automation, the technical contribution feels somewhat limited. The system is more focused on practical application rather than introducing a novel method or algorithm. Additionally, there is no follow-up evaluation of the entire system under real-world conditions. It would be beneficial to see both quantitative and qualitative analyses of VeSX in real-world usage scenarios to better understand its performance in practical settings. E.g., a statistical evaluation or user study focusing on whether this system truly works for real-world tasks would significantly strengthen the paper. A field study or feedback from real users would also provide practical insights into how the system performs in dynamic, unstructured environments.

The concept of "self-correction" is promising, but the evaluation of this feature is not comprehensive enough. Although the paper includes an ablation study, a more detailed analysis of the self-correction mechanism is necessary to demonstrate its effectiveness. For example, breaking down how self-correction functions in different failure cases or assessing the time and resource costs associated with error correction would provide deeper insights into the feature’s utility.

The paper does not address what happens if the system or specific components fail. While self-correction is included, there is no discussion of how the system handles scenarios where self-correction or verification mechanisms fail. For real-world applications, understanding the system’s resilience and fallback options is crucial. Including an analysis of fail-safe protocols would enhance the system’s reliability and robustness.

**Questions:**

How would the system perform in a real-world scenario with unstructured tasks and environments? Have you considered conducting a user study or statistical analysis to test its practical application and effectiveness?

While you provide an ablation study, could you expand on the evaluation of the self-correction feature? How does it handle various failure cases, and what are the associated costs (in terms of time, resources, etc.) for error correction?

Beyond its application focus, do you see any potential for extending the technical contributions of VeSX? For example, could the subgoal-guided verification or hierarchical self-correction be applied in other domains?

PS: content under Section 2.1 Overview is missing.

---

> ### Author Response · Authors · 2024-11-25
>
> We sincerely thank the reviewer for reading our paper and providing valuable insights.
>
> > W1.1 The system is more  focused on practical application rather than introducing a novel method  or algorithm. Additionally, there is no follow-up evaluation of the  entire system under real-world conditions. It would be beneficial to see  both quantitative and qualitative analyses of VeSX in real-world usage  scenarios to better understand its performance in practical settings.  E.g., a statistical evaluation or user study focusing on whether this  system truly works for real-world tasks would significantly strengthen  the paper. A field study or feedback from real users would also provide  practical insights into how the system performs in dynamic, unstructured  environments.
>
> AW 1.1: We invited ten volunteers and asked them to experience our pipeline. User feedback indicates that simple user needs can be fulfilled in VeSX, but the total execution time is long. We show some cases in Appendix A.1.2 and Appendix C, which are close to users' daily require when surfing on the web.
>
> > W2.1: The concept of "self-correction" is promising, but the evaluation of  this feature is not comprehensive enough. Although the paper includes an  ablation study, a more detailed analysis of the self-correction  mechanism is necessary to demonstrate its effectiveness. For example,  breaking down how self-correction functions in different failure cases  or assessing the time and resource costs associated with error  correction would provide deeper insights into the feature’s utility.
>
> AW2.1: We provide some case studies in Appendix C to analyze the effect of the self-correction mechanism. We also conduct experiments to calculate the computational cost of self-correction in Appendix B.1. The verification and  self-correction module account for 10.7% of input tokens and 16.1% of output tokens.
>
> > W3.1: The paper does not address what happens if the system or specific components fail. While self-correction is included, there is no discussion of how the system handles scenarios where self-correction or verification mechanisms fail. For real-world applications, understanding  the system’s resilience and fallback options is crucial. Including an analysis of fail-safe protocols would enhance the system’s reliability and robustness.
>
> AW3.1: Understanding the robustness of the system is important for practical applications. Currently, we have set a maximum number of actions for the entire process to prevent VeSX from getting stuck in an infinite loop. This setting is taken by many related works. Furthermore, we have some designs for resilience except for the self-correction module:
>
> 1. We set a limit to the count of actions within one subtask. When the limit is reached, the subtask is considered as infeasible and the replanning module will be triggered directly.
>
> 2. We make the reflection module to check whether there is an error in verification to avoid the false negative caused by verification.
>
> 3. The verification module often be constructed with multiple functions to avoid over rely on one function's result.
>
> We make some analysis of each modules to see how often they make mistakes, which can be found in Appendix B.2.
>
> The robustness of the system is a critical yet challenging issue, and we find it difficult to resolve comprehensively in a short period. We hope our experiments can clarify your concerns and provide guidance for further improving the system's stability in future research and engineering efforts.

---

> ### Author Response · Authors · 2024-11-25
>
> > Q1.1: How would the system perform in a real-world scenario with unstructured tasks and environments? Have you considered conducting a user study or statistical analysis to test its practical application and effectiveness?
>
> AQ1.1: See AW1.1.
>
> > Q2.1: While you provide an ablation study, could you expand on the evaluation of the self-correction feature? How does it handle various failure cases, and what are the associated costs (in terms of time, resources, etc.) for error correction?
>
> AQ2.1: See AW2.1 about the evaluation of the self-correction feature.
>
> In terms of computational costs, we supplemented some experiments for analysis. The extra computational cost caused by error correction mainly lies in two aspects: workflow design, and the more action steps.
>
> 1. Workflow Design: From the experiments, the verification, reflection and replanning modules account for approximately 10.7% of the input tokens and 16.1% of the output tokens. This indicates that the three main modules in VeSX workflow, verification, reflection, and replanning, do not introduce significant more computational cost.
>
> 2. More Action Steps: We compared our method with the Sodhi et al. (2024) approach, which uses human-labeled subtasks to guide the LLM in breaking down and solving tasks. Their average action step is 9.1, while our average action step is 13.1. From this perspective, VeSX requires approximately 43.9% more actions compared to human-guided workflows to achieve competitive results.
>
> Due to the significant time and cost involved in reproducing other approaches, we do not fully reproduce all of them. Compared to the sota method with human guidance, VeSX takes about ~50% more computational cost to achieve competitive results totally. It is nearly impossible for other methods to achieve the same accuracy by 1.5x scaling up without other design. The details of this experiments can be found in Appendix B.1.
>
> > Q3.1: Beyond its application focus, do you see any potential for extending the technical contributions of VeSX? For example, could the subgoal-guided verification or hierarchical self-correction be applied in other domains?
>
> AQ3.1: Yes, the verification scheme, the self-correction approach,  and the construction process of the exemplar bank used in VeSX may have the potential to be effective in other tasks. In simple terms, we believe that VeSX will be useful in complex tasks that require multi-step reasoning and execution. We have added a new section, Section 5, to discuss about the current limitations of VeSX and potential directions for future expansion. We will also consider leaving this part for our further studies.
>
> > Q4.1: PS: content under Section 2.1 Overview is missing.
>
> AQ4.1: Thank you for your correction! The missing section 2.1 has now been added.

---

> > ### Comment · Reviewer_MnFm · 2024-11-26
> >
> > I appreciate the author's efforts in addressing my concerns. However, I believe that case studies alone are insufficient to demonstrate the practical impact of this work. Given its application-oriented nature, the limited novelty cannot be adequately compensated without evidence of real-world testing or deployment. Thus, I will keep my current score.

---

### Official Review · Reviewer_7Lym · 2024-10-26

**Soundness:** 2
**Presentation:** 3
**Contribution:** 3
**Rating:** 3
**Confidence:** 4

**Summary:**

This paper presents VeSX, a framework for web automation that integrates verification, self-correction, and in-context learning mechanisms.

**Strengths:**

1. The exemplar bank's approach of breaking down trajectories into smaller, reusable components is innovative and practically valuable for reducing context length while maintaining effectiveness.
2. The ablation studies are comprehensive and help validate the contribution of each component.
3. The design of local reflection and global reflection are interesting.

**Weaknesses:**

1. The literature review on LLM-based agents appears incomplete, missing several relevant recent works
2. About "subgoal-based verification," process supervision is a well-studied research direction[1]. This paper's key difference lies in the hierarchical verification mechanism. However， to prove the effectiveness of hierarchical verification，more comparison experiments and discussions should be made。
3. Although the authors don't use ground-truth labels, their exemplar construction process still utilizes tasks from the target domain. While this doesn't constitute supervision in the traditional sense, it does provide the model with domain-specific information that zero-shot baselines may not have access to, potentially creating an unfair comparison if the baselines are purely zero-shot.

References:

[1] Lightman H, Kosaraju V, Burda Y, et al. Let's verify step by step[J]. arXiv preprint arXiv:2305.20050, 2023.

**Questions:**

See Weaknesses. Further:

1. Could the authors clarify whether the 60 sampled tasks used for creating exemplars overlap with the test set? If so, how do they address potential data leakage concerns?
2. How does the system handle cases where the verification phase produces false positives or false negatives? Is there any analysis of the verification accuracy?
3. How scalable is the exemplar bank approach as the number of tasks and domains increases? Is there a strategy for managing the growing size of the exemplar bank?

---

> ### Author Response · Authors · 2024-11-25
>
> We sincerely thank the reviewer for reading our paper and providing valuable insights.
>
> > W1.1: The literature review on LLM-based agents appears incomplete, missing several relevant recent works
>
> AW1.1: Thank you for your valuable advice. We add a new paragraph to talk about the recent research on LLM-based agents, which can be found in Section 4.1.
>
> > W2.1: About "subgoal-based verification," process supervision is a well-studied research direction[1]. This paper's key difference lies in the hierarchical verification mechanism. However, to prove the effectiveness of hierarchical verification，more comparison experiments and discussions should be made
>
> AW2.1: It is an interesting and valuable question! In fact, process rewards were our initial exploration direction during the research, while hierarchical verification and error correction were later discovered paths. We found that step-by-step verification is difficult to implement in such scenarios and has low error correction efficiency. Previous verification methods were mostly used in mathematical tasks, and there has been little work related to LLM agents that require interaction with the environment. During our exploration, we identified two main reasons:
>
> 1. First, there is inherently a lack of a verification target in such scenarios. In mathematical tasks, the verification target is often whether a particular expression is incorrect, but in the agent’s environment, there is no clear right or wrong verification target. To address this issue, we proposed the idea of subgoal-guided verification and a scheme for generating expectations while predicting actions.
>
> 2. Second, the granularity of step-by-step verification is not suitable for capturing errors in agent tasks. In many cases, the errors are caused by actions taken several steps earlier. Step-by-step verification only ensures current stability and does not involve error correction functionality, which is why we designed hierarchical self-correction.
>
> We did not provide such comparative experiments because there is currently no good solution for step-by-step verification in web automation. In fact, about 56% of our subtasks involve only one action (excluding finish_subtask).
>
> > W3.1: Although the authors don't use ground-truth labels, their exemplar construction process still utilizes tasks from the target domain. While this doesn't constitute supervision in the traditional sense, it does provide the model with domain-specific information that zero-shot baselines may not have access to, potentially creating an unfair comparison if the baselines are purely zero-shot.
>
> AW3.1: All other baselines used in-context-learning methods. Our idea comes from precess of the data collection with a model or behavior policy. So we think the comparison is not unfair.

---

> ### Author Response · Authors · 2024-11-25
>
> > Q1.1: Could the authors clarify whether the 60 sampled tasks used for creating exemplars overlap with the test set? If so, how do they address potential data leakage concerns?
>
> AQ1.1: Yes, the exemplars are sampled from the benchmark and included in later evaluation. But this process does not use the ground-truth labels or any other outer supervision. We let the LLM do reflection by itself and this idea comes from precess of the data collection with a model or behavior policy. So it will not lead to an unfair comparison, though it is more acceptable to use a new set of queries to construct the exemplar bank. We will leave this for our further studies.
>
> > Q2.1: How does the system handle cases where the verification phase produces false positives or false negatives? Is there any analysis of the verification accuracy?
>
> AQ2.1: Good question. VeSX let reflection module to judge whether error happened during verification first. The example shown in Appendix A.1.2 is under such circumstance.
>
> And we supplement the experiments to give more analysis of verification phase. We give an estimation of the sensitivity of verification:
>
> 1. The false positive rate can be estimated as 0.57. This result demonstrates that the verification may not be reliable when classified as a positive sample.
>
> 2. The false negative rate needs to be estimated using the success rate when not verifying and the success rate when verification fails, and can be estimated as 0.20. This result demonstrates that verification rarely classifies positive samples as negative, which does not incur much unnecessary self-correction overhead.
>
> 3.The true negative rate can be estimated as 0.79. This result demonstrates that the verification has high rate to capture the error in the process.
>
> Here, the predicted positive means "verification pass", which is defined as the instance that all verifications in this process pass, while the predicted negative means "verification fail", which is defined as the instance where at least one verification does not pass.
>
> The details of these experiments can be found in Appendix B.3.
>
> > Q3.1: How scalable is the exemplar bank approach as the number of tasks and domains increases? Is there a strategy for managing the growing size of the exemplar bank?
>
> AQ3.1: We supplement some experiments to test the impact of scaling of exemplar bank. We mainly conduct two experiments:
> 1. Scaling of Exemplar Bank: We set four settings: (1) sampling 70% of the whole exemplar bank, (2) using the whole execution exemplar bank but not the planning exemplar, (3) using the whole planning exemplar bank but not the execution exemplar, and (4) using the whole exemplar bank. The results shows that using only 70% of the whole bank causes a obvious decrease of SR comparing to even removing the whole planning exemplars or execution exemplars. It indicates the more diversity of the execution or the planning exemplars will bring more increase and the potential of the scaling of the exemplar bank.
>
> |Number of Planning Exemplars|Number of Execution Exemplars|Success Rate|
> |------|------|------|
> |26|120|0.334|
> |0|171|0.340|
> 35|0|0.373|
> |35|171|0.412|
>
> 2. Scaling of In-Context_Learning Exemplars: The SR improves when the count of ICL exemplars increases, underlying the potential of the scaling of exemplars during the inference process.
>
> |Number of  ICL Exemplars| Success Rate|
> |------|------|
> |3|0.412|
> |5|0.458|
>
> We do not conduct experiments to quantify whether exemplars from other domain contribute to the current domain because the executions of subtasks in different scenarios do not share the same pattern. For example, you can search the items and click to view its information when in shopping sites, but you may enter a certain subreddit to search for some post in forums. The logic to accessing contents in shopping sites and forums vary, indicating the difficulty of scaling across domains.
>
> One approach worth considering to manage the growing exemplars is to remove functionally similar exemplars from the exemplar bank, allowing the remaining exemplars to be more diverse. Currently, the size of the exemplar bank does not require a sophisticated management strategy. We will  consider leaving this point for our future research after a system capable of collecting large amounts of data is implemented.

---

> ### Comment · Reviewer_7Lym · 2024-11-27
>
> Thanks for the authors' response. Regarding Q 1.1, even though VeSX did not use the ground truth from the test set, it still utilized some tricks to formulate better in-context exemplars (Line 307), which is beyond the framework itself. Furthermore, as I already mentioned, the model is actually hacking some domain-specific knowledge in this situation. Given the potential data leakage risk, I decide to maintain my score.

---

### Official Review · Reviewer_H4cu · 2024-10-29

**Soundness:** 3
**Presentation:** 3
**Contribution:** 3
**Rating:** 6
**Confidence:** 4

**Summary:**

The authors present a solution to automating web tasks such as checking on shopping orders. The solution leverages LLMs that break down the task into subtasks, executes those subtasks in the browser, verifies the subgoals are accomplished, can self corrects and replan if necessary, and leverages in context examples retrieved from an exemplar bank created by the authors. Experimental results show the authors' superior approach compared to the literature on WebArena, a popular benchmark in the literature.

**Strengths:**

Summary:
- Solves a relevant problem
- Adopts a solution that is based on the latest technology
- Beats the state of the art with their experimental results on a well-known benchmark from the literature

Details:
The problem of automating web tasks is difficult and very relevant in this age of enterprise productivity. Many tasks are quite repetitive and could benefit from automation but the diversity of browsers and apps and tasks makes it challenging for automated systems.

LLMs have proven beneficial and the paper not only leverages them but also tests GPT-4o which is one of the newest and less costly models compared to others from the literature.

The proposed framework introduces three key components to the LLM pipeline: 1) sub-goal verification, 2) self-correction and 3) exemplar bank. Each of these components are not particularly original but combining them into a single framework and applying this framework to the web task automation leads to state of the art of results.

**Weaknesses:**

Summary:
- Limited experimental results and analysis including missing computational cost analysis, error analysis especially when linked to the various contributed components in their framework
- Typing and grammar mistakes

Details:
The experimental results show that the proposed approach (including individual components) do improve the state of the art on the web arena benchmark. The authors compare to other approaches from the literature and do an ablation study on the components they proposed. However, the experimental analysis is still missing some key results that could help the community understand and evaluate this approach better. Notable, the authors perform multiple LLM calls during their pipeline. Quantifying the computational cost (whether with number of calls per input or some other metrics) would help evaluate the approach and compare to other in the literature. Furthermore, the authors do not analyze what errors benefited more from what components in their pipelines. What types of errors needed replanning, which were addressed with reflection only, why did some of the verifications fail, etc. Finally, the authors perform an end to end evaluation but do not evaluate each component individually on intrinsic metrics; e.g., how often was the reflection component able to correct an error that is within its scope, etc.

**Questions:**

NA

---

> ### Author Response · Authors · 2024-11-25
>
> We sincerely thank the reviewer for reading our paper and providing valuable insights.
>
> > W1.1: Limited experimental results and analysis including missing computational cost analysis...
>
> AW1.1: We supplemented some experiments about computational cost analysis. The extra computational cost mainly comes from three aspects: workflow design, examples for in-context-learning and the more action steps.
>
> 1. Workflow Design: From the experiments, the verification, reflection and replanning modules account for approximately 10.7% of the input tokens and 16.1% of the output tokens. This indicates that the three main modules in VeSX workflow, verification, reflection, and replanning, do not introduce significant more computational cost.
>
> 2. Examples for ICL: From our experiments, the examples for ICL account for 35.3% of the input tokens, which is a big deal. However, considering almost every other methods use ICL technique, the computational cost of these examples are acceptable especially in that our scheme of exemplar bank can provide exemplars effectively and efficiently. Furthermore, we do summarization for the examples' input to save the input tokens while some of other methods use the redundant web context for ICL.
>
> 3. More Action Steps: We compared our method with the Sodhi et al. (2024) approach, which uses human-labeled subtasks to guide the LLM in breaking down and solving tasks. Their average action step is 9.1, while our average action step is 13.1. From this perspective, VeSX requires approximately 43.9% more actions compared to human-guided workflows to achieve competitive results.
>
> Due to the significant time and cost involved in reproducing other approaches, we do not fully reproduce all of them. Compared to the sota method with human guidance, VeSX takes about ~50% more computational cost to achieve competitive results totally. It is nearly impossible for other methods to achieve the same accuracy by 1.5x scaling up without other design. The details of this experiments can be found in Appendix B.1.
>
> > W1.2: Limited experimental results and analysis including missing ..., error analysis especially when linked to the various contributed components in their framework
>
> AW1.2: As suggested by the reviewer, we analyze the performance of each new module of VeSX, which is in Appendix B.3. And we conduct some case studies on the errors occurring in each module, including reflection and replanning, which can be found in Appendix C.
>
> > W2.1: Typing and grammar mistakes
>
> AW2.1: We have carefully reviewed our article again and made revisions to the textual and figure errors. Thank you for the suggestions.

---

> > ### Comment · Reviewer_H4cu · 2024-11-27
> >
> > Thank you for the additional details.

---

### Official Review · Reviewer_zkoG · 2024-10-31

**Soundness:** 2
**Presentation:** 2
**Contribution:** 2
**Rating:** 5
**Confidence:** 4

**Summary:**

The paper presents VeSX, a framework for enhancing large language models (LLMs) in web automation tasks by introducing verification, self-correction, and in-context learning. VeSX aims to tackle the common challenges in web automation workflows, such as subtask infeasibility and data scarcity, by implementing three key components: subgoal-guided verification, which checks the accuracy of each subtask; hierarchical self-correction, allowing the model to reflect and replan when errors occur; and an exemplar bank for in-context learning, storing structured examples that improve decision-making. Evaluated on the WebArena benchmark, VeSX achieved a state-of-the-art success rate of 34% across multiple scenarios without human guidance, demonstrating its potential to improve accuracy and reliability in complex, multi-step web interactions.

**Strengths:**

- The web automation task is interesting and worth exploring.
- The proposed self-reflection approach seems to have great improvement in performance, highlighting its potential to enhance task success and reliability in complex, interactive environments.

**Weaknesses:**

- The novelty is limited. Compared to previous work on web automation, the paper integrates self-reflection and retrieval-augmentation components, both of which have been widely explored. The paper also lacks discussions on relevant works on reflection and retrieval augmentation.
- The writing needs to be improved, especially in explaining the main components and their novelty.
    * Section 2.1 Overview is empty
    * Clearly indicate success rates as percentages by adding the percentage sign (e.g., 34% instead of just 34)
    * It will be better to put short descriptions in the captions for terms in the table (‘Shop’, ‘CMS’, ‘Red’, ‘Git’, ‘Map’).
    * Adding example prompts would provide readers with a practical understanding of the pipeline.
- Figures need to be significantly improved:
    * ‘orders’ rather than ‘oreders’ in the teaser figure.
    * Miss left bracket for ‘click sorted by]’. Is ‘click [sorted by]’ and ‘click [sortby]’ the same operation?
    * the texts frequently touch or cross the boundaries of the icons.
    * Some figures are blurry and difficult to interpret. (e.g. In Figure 2, it is not clear what the four boxes below the environment represent.)
    * The figure captions should be refined to clearly describe each major component.

**Questions:**

How does VeSX distinguish itself from approaches like Tree of Thought (which leverages branching and self-verification of reasoning steps), Reflexion (which incorporates self-reflection mechanisms), and various retrieval-augmented frameworks?

---

> ### Author Response · Authors · 2024-11-25
>
> We sincerely thank the reviewer for reading our paper and providing valuable insights.
>
> > W1.1: The novelty is limited. Compared to previous work on web automation, the paper integrates self-reflection and retrieval-augmentation components, both of which have been widely explored. The paper also lacks discussions on relevant works on reflection and retrieval augmentation.
>
> AW1.1: The self-correction technology has been rarely used in the field of web automation, and there is currently no solution centered around self-correction for web automation. VeSX implements a self-correction approach for web automation tasks with sophisticated design at various stages,  such as subgoal-guided verification and a reflection & replanning error correction hierarchy. We have also designed specific elements within these stages, such as our proposed execution with expectation in the reflection phase.
> The retrieval-augmentation technology has been widely studied across various fields and is also commonly used in web automation. We have made some incremental improvements in this area. We addressed the issue of context length when applying previous retrieval-augmented methods to real-world scale scenarios, and we designed a metadata construction scheme to facilitate retrieval.
>
> > W2.1: Section 2.1 Overview is empty
>
> AW2.1: Thank you for your correction! The missing section 2.1 has now been added.
>
> > W2.2: Clearly indicate success rates as percentages by adding the percentage sign (e.g., 34% instead of just 34)
>
> AW2.2: We compared several presentation options. Including a percentage seems somewhat redundant, while omitting it does not clearly convey that this is a success rate. After weighing our options, we decided to use decimals for clearer representation.
>
> > W2.3: It will be better to put short descriptions in the captions for terms in the table (‘Shop’, ‘CMS’, ‘Red’, ‘Git’, ‘Map’).
>
> AW2.3: In section 3.1, we introduced which scenarios correspond to each  abbreviation. As suggested by the reviewer, we have also added descriptions of the terms in the table header to help readers obtain information more clearly and quickly from the table.
>
> > W2.4: Adding example prompts would provide readers with a practical understanding of the pipeline.
>
> AW2.4: We also have added some examples of each module in the appendix. Please refer to Appendix A.2 section.
>
> > W3:
>
> >    ‘orders’ rather than ‘oreders’ in the teaser figure.
>
> >    Miss left bracket for ‘click sorted by]’. Is ‘click [sorted by]’ and ‘click [sortby]’ the same operation?
>
> >    the texts frequently touch or cross the boundaries of the icons.
>
> >    Some figures are blurry and difficult to interpret. (e.g. In Figure 2, it is not clear what the four boxes below the environment represent.)
>
> >    The figure captions should be refined to clearly describe each major component.
>
> AW3: Thank you for your thorough reading of our article and for pointing out so many formatting errors! We have carefully reviewed our article again and made revisions to the textual and figure errors.
>
> > Q1.1: How does VeSX distinguish itself from approaches like Tree of Thought (which leverages branching and self-verification of reasoning steps), Reflexion (which incorporates self-reflection mechanisms), and various retrieval-augmented frameworks?
>
> AQ1.1: The ToT and related tree search frameworks have been widely used in various agent tasks, including web automation. The approach that leverages branching and self-verification of reasoning steps has been adopted by many agent frameworks and has become a mainstream agent paradigm. Among the baselines we compared in our experiments, some solutions, such as Koh et al. (2024) also used tree search methods, but their performance was not as strong as ours. This is due to VeSX's sophisticated design in how verification and error correction are handled. What we aim to showcase is how to design effective verification and error correction solutions, an area that lacks relevant research in web automation. From this perspective, our work is entirely innovative.
> Regarding the comparison with the reflection and retrieval-augmented frameworks, we addressed this question in AW1.1.

---

> > ### Comment · Reviewer_zkoG · 2024-11-28
> >
> > Thanks for the response. The writing improves after the paper revision. Applying self-correction (a widely studied concept) to a new task such as web automation is a step forward, but it does not demonstrate sufficient novelty. As such, I will raise the score but still do not find it sufficient for acceptance.

---

### Official Review · Reviewer_NqCv · 2024-11-02

**Soundness:** 3
**Presentation:** 2
**Contribution:** 3
**Rating:** 6
**Confidence:** 3

**Summary:**

VeSX is a framework for interactive web automation tasks using LLMs that focuses on improving sub-task feasibility, a common issue for planning based methods that initially break down tasks into multiple steps before execution. To improve sub-goal feasibility, VeSX introduces three components: sub-goal guided verification, which verifies either with the model itself or external methods if the sub-task is feasible. The second is a hierarchical self-correction method that takes place when verification fails during planning as well as during execution. Hierarchical self-correction uses reflection to correct verification errors, and replans if necessary. Lastly, VeSX uses an exemplar bank for in-context learning for both planning and execution. Unlike previous uses of in-context learning, the VeSX exemplar bank does not use full trajectories, instead sampling from existing trajectories to build the examples. For evaluation, VeSX uses 5 scenarios from the WebArena benchmark.

**Strengths:**

- Identifies key weaknesses in current methods for web automation
- Method tries to account for different types of failures through the dual verification system and self-correction
- Notable observations as part of method:
    - A) It is easier to verify then come up verification for different goals
    - B) Having the LLM output expected results as part of reflection
- Exemplar bank: I think this is one of the strongest contributions since it is very different than existing work in particular using parts of trajectories instead of full trajectories.

**Weaknesses:**

- Presentation:
    - I am a bit confused about the overall workflow. It would be helpful to have it written in an algorithm.
    - It would also be helpful to see more examples
- Extra Time and Cost:
    - How much extra time and tokens does it take for this method compared to others (if available for other methods)? If these other methods also had access to more compute, they might also have higher performance.
- Original of exemplars: Are the exemplars produced from questions in the benchmark? Are those questions included in the final results? This could also lead to an unfair comparison.
- One stated advantage of the approach is that human guidance is not needed. Is any human guidance used to design the prompts for the different steps? Is the exemplar bank used as in-context examples for all of the different steps?

**Questions:**

In addition the ones listed in the weaknesses:
- How many total tasks are there for each scenario?
- Out of the 60 sampled tasks for each scenario, how many exemplars were produced?
- After re-planning is done or self-correction, does the process start from the beginning again? Is there a limit to the number of times self-correction or reflection is allowed? Is this the same as in other papers?

---

> ### Author Response · Authors · 2024-11-25
>
> We sincerely thank the reviewer for reading our paper and providing valuable insights.
>
> > W1.1: Presentation: I am a bit confused about the overall workflow. It would be helpful to have it written in an algorithm.
>
> AW1.1: As suggested by the reviewer, we have improved the presentation by adding an algorithm of the whole workflow in the appendix part of our revised version. Please refer to Appendix A.1.1 section.
>
> > W1.2: Presentation: It would also be helpful to see more examples.
>
> AW1.2: We also have added some examples in the appendix. Please refer to Appendix A.1.2 section.
>
> > W2.1: Extra Time and Cost: How much extra time and tokens does it take for this method compared to others (if available for other methods)? If these other methods also had access to more compute, they might also have higher performance.
>
> AW2.1: We supplemented some experiments about computational cost analysis. The extra computational cost mainly comes from three aspects: workflow design, examples for in-context-learning and the more action steps.
>
> 1. Workflow Design: From the experiments, the verification, reflection and replanning modules account for approximately 10.7% of the input tokens and 16.1% of the output tokens. This indicates that the three main modules in VeSX workflow, verification, reflection, and replanning, do not introduce significant more computational cost.
>
> 2. Examples for ICL: From our experiments, the examples for ICL account for 35.3% of the input tokens, which is a big deal. However, considering almost every other methods use ICL technique, the computational cost of these examples are acceptable especially in that our scheme of exemplar bank can provide exemplars effectively and efficiently. Furthermore, we do summarization for the examples' input to save the input tokens while some of other methods use the redundant web context for ICL.
>
> 3. More Action Steps: We compared our method with the Sodhi et al. (2024) approach, which uses human-labeled subtasks to guide the LLM in breaking down and solving tasks. Their average action step is 9.1, while our average action step is 13.1. From this perspective, VeSX requires approximately 43.9% more actions compared to human-guided workflows to achieve competitive results.
>
> Due to the significant time and cost involved in reproducing other approaches, we do not fully reproduce all of them. Compared to the sota method with human guidance, VeSX takes about ~50% more computational cost to achieve competitive results totally. It is nearly impossible for other methods to achieve the same accuracy by 1.5x scaling up without other design. The details of this experiments can be found in Appendix B.1.
>
> > W3.1: Original of exemplars: Are the exemplars produced from questions in the benchmark? Are those questions included in the final results? This could also lead to an unfair comparison.
>
> AW3.1: Yes, the exemplars are sampled from the benchmark and included in later evaluation. But this process does not use the  ground-truth labels or any other outer supervision. We let the LLM do reflection by itself and this idea comes from precess of the data collection with a model or behavior policy. So it will not lead to an unfair comparison, though it is more acceptable to use a new set of queries to construct the exemplar bank. We will leave this for our further studies.
>
> > W4.1: One stated advantage of the approach is that human guidance is not needed. Is any human guidance used to design the prompts for the different steps? Is the exemplar bank used as in-context examples for all of the different steps?
>
> AW4.1: In VeSX, the same prompt is used for different steps in execution (except for retrieving different exemplars from the exemplar bank for in-context-learning). The exemplar bank is same for all steps.

---

> ### Author Response · Authors · 2024-11-25
>
> > Q1.1: How many total tasks are there for each scenario?
>
> AQ1.1: Here we list out the tasks for each scenario:
>
> |Shop|CMS|Git|Red|Map|
> |------|------|------|------|------|
> |153|173|198|110|127|
>
> > Q2.1: Out of the 60 sampled tasks for each scenario, how many exemplars were produced?
>
> AQ2.1: Here we list out the details of the exemplar bank:
>
> | |Shop|CMS|Git|Red|Map|
> |------|------|------|------|------|------|
> |Planning|35|24|30|29|20|
> |Execution|171|123|211|193|87|
>
> We also update this information in Appendix B.2.
>
> > Q3.1: After re-planning is done or self-correction, does the process start from the beginning again? Is there a limit to the number of times self-correction or reflection is allowed? Is this the same as in other papers?
>
> AQ3.1: VeSX will not restart from the beginning but from the errorous subtask. We are not limit the time of self-corrections but set a restrict to the total steps of the executions for the whole process to prevent it stuck in an endless loop. This setting is follwed by other papers.

---

> > ### Comment · Reviewer_NqCv · 2024-11-26
> >
> > Thank you to the authors for answering my questions. I will keep my score at 6 though I think it would have been better if the exemplars did not come from the same dataset as the test set.

---

### Meta-Review · Area_Chair_9oCo · 2024-12-21

**Metareview:**

This paper proposes a agent-based framework for web automation tasks (VeSX), with 3 main new modules: subgoal-guided verification, hierarchical self-correction, and exemplar bank. The goal is to ensure that subtasks are are executable, and be able to correct both subtasks and the plan. Instead of storing entire trajectories as previous work, the exemplar bank in VeSX stores single-action exemplars, as well as planning exemplars. Evaluation on WebArena shows that VeSX (which does not require human feedback) performs comparably to a baseline method that uses human feedback. While reviewers appreciated the introduction of different modules in VeSX and the importance of the application, concerns were raised on potential issues in the execution of the exemplar bank (the usage of the benchmark examples to construct this), computational cost of the framework, and additional analysis (e.g. what happens when modules fail?). There was additional discussion between reviewers during the discussion phase on these concerns. The AC agrees with the concern that there are a lot of design choices in the examples used to form the exemplar bank (while test labels were not used, it is difficult to evaluate how these choices impacted performance). Additionally, the ablation study is only on one of the 5 scenarios, and the effect on the entire dataset is unknown. Additional examples on comparing the use of exemplar bank with in-context learning in VeSX would be helpful, or what if the exemplar bank is formed with 1 scenario held-out? The framing of the paper need to be more clear if all test tasks need to be known ahead of time to form the exemplar bank, vs. being used with in-context learning. The proposed method is interesting, but additional experiments are required to fully understand the contribution of each component. With completed experiments, I think this could be a good submission in the future.

**Additional Comments On Reviewer Discussion:**

Reviewers raised concerns on data leakage with examples from the dataset being also used to construct the exemplar bank, robustness of the system (e.g. what if components fail), computational cost, potentially limited novelty in method design, and presentation issues in the paper. Some concerns were addressed by authors during rebuttal period. However, key concerns remain on novelty and the exemplar bank construction itself, and reviewers discussed whether this is similar to in-context learning used by other works or not. I took a closer look at the discussion of this point, and there seems to be a lot of design choices on constructing this exemplar bank (which is important for performance, and is not clear in the current presentation the impact of these choices). I encourage the authors to either perform more experiments (discussed above, further analyzing these decisions) or re-frame the paper to mention that the test scenarios should be known ahead of time (without labels), which can be used for the construction of an exemplar bank.

---

### Decision · Program_Chairs · 2025-01-22

Reject